# Generator Identification for Linear SDEs with Additive and Multiplicative Noise

**Yuanyuan Wang**
The University of Melbourne
`yuanyuanw2@student.unimelb.edu.au`

**Xi Geng**
The University of Melbourne
`xi.geng@unimelb.edu.au`

**Wei Huang**
The University of Melbourne
`wei.huang@unimelb.edu.au`

**Biwei Huang**
University of California, San Diego
`bih007@ucsd.edu`

**Mingming Gong** *
The University of Melbourne
`mingming.gong@unimelb.edu.au`

## Abstract

In this paper, we present conditions for identifying the **generator** of a linear stochastic differential equation (SDE) from the distribution of its solution process with a given fixed initial state. These identifiability conditions are crucial in causal inference using linear SDEs as they enable the identification of the post-intervention distributions from its observational distribution. Specifically, we derive a sufficient and necessary condition for identifying the generator of linear SDEs with additive noise, as well as a sufficient condition for identifying the generator of linear SDEs with multiplicative noise. We show that the conditions derived for both types of SDEs are generic. Moreover, we offer geometric interpretations of the derived identifiability conditions to enhance their understanding. To validate our theoretical results, we perform a series of simulations, which support and substantiate the established findings.

## 1 Introduction

Stochastic differential equations (SDEs) are a powerful mathematical tool for modelling dynamic systems subject to random fluctuations. These equations are widely used in various scientific disciplines, including finance [11, 30, 40], physics [53, 55, 58], biology [2, 8, 61] and engineering [18, 44, 55]. In recent years, SDEs have garnered growing interest in the machine learning research community. Specifically, they have been used for tasks such as modelling time series data [19, 21, 33] and estimating causal effects [5, 36, 47].

To enhance understanding we first introduce the SDEs of our interest, which are multidimensional linear SDEs with additive and multiplicative noise, respectively. Consider an $m$-dimensional standard Brownian motion defined on a filtered probability space $(\Omega, \mathcal{F}, \mathbb{P}, \{\mathcal{F}_t\})$, denoted by $W := \{W_t = [W_{1,t}, \ldots, W_{m,t}]^\top : 0 \leqslant t < \infty\}$. Let $X_t \in \mathbb{R}^d$ be the state at time $t$ and let $x_0 \in \mathbb{R}^d$ be a constant vector denoting the initial state of the system, we present the forms of the aforementioned two linear SDEs.

---

*Corresponding author.

37th Conference on Neural Information Processing Systems (NeurIPS 2023).

1. Linear SDEs with additive noise.
$$dX_t = AX_t dt + G dW_t, \quad X_0 = x_0, \tag{1}$$
where $0 \leqslant t < \infty$, $A \in \mathbb{R}^{d \times d}$ and $G \in \mathbb{R}^{d \times m}$ are some constant matrices.

2. Linear SDEs with multiplicative noise.
$$dX_t = AX_t dt + \sum_{k=1}^{m} G_k X_t dW_{k,t}, \quad X_0 = x_0, \tag{2}$$
where $0 \leqslant t < \infty$, $A, G_k \in \mathbb{R}^{d \times d}$ for $k = 1, \ldots, m$ are some constant matrices.

Linear SDEs are wildly used in financial modeling for tasks like asset pricing, risk assessment, and portfolio optimization [3, 10, 24]. Where they are used to model the evolution of financial variables, such as stock prices and interest rates. Furthermore, linear SDEs are also used in genomic research, for instance, they are used for modeling the gene expression in the yeast microorganism Saccharomyces Cerevisiae [17]. The identifiability analysis of linear SDEs is essential for reliable causal inference of dynamic systems governed by these equations. For example, in the case of Saccharomyces Cerevisiae, one aims to identify the system such that making reliable causal inference when interventions are introduced. Such interventions may involve deliberate knockout of specific genes to achieve optimal growth of an organism. In this regard, identifiability analysis plays a pivotal role in ensuring reliable predictions concerning the impact of interventions on the system.

Previous studies on identifiability analysis of linear SDEs have primarily focused on Gaussian diffusions, as described by the SDE (1) [6, 16, 23, 28, 35, 42]. These studies are typically based on observations located on one trajectory of the system and thus require restrictive identifiability conditions, such as the ergodicity of the diffusion or other restrictive requirements on the eigenvalues of matrix $A$. However, in practical applications, multiple trajectories of the dynamic system can often be accessed [15, 31, 45, 54]. In particular, these multiple trajectories may start from the same initial state, e.g., in experimental studies where repeated trials or experiments are conducted under the same conditions [9, 12, 26, 29] or when the same experiment is performed on multiple identical units [48]. To this end, this work presents an identifiability analysis for linear SDEs based on the distribution of the observational process with a given fixed initial state. Furthermore, our study is not restricted to Gaussian diffusions (1), but also encompasses linear SDEs with multiplicative noise (2). Importantly, the conditions derived for both types of SDEs are generic, meaning that the set of system parameters that violate the proposed conditions has Lebesgue measure zero.

Traditional identifiability analysis of dynamic systems focuses on deriving conditions under which a unique set of parameters can be obtained from error-free observational data. However, our analysis of dynamic systems that are described by SDEs aims to uncover conditions that would enable a unique generator to be obtained from its observational distribution. Our motivation for identifying generators of SDEs is twofold. Firstly, obtaining a unique set of parameters from the distribution of a stochastic process described by an SDE is generally unfeasible. For example, in the SDE (1), parameter $G$ cannot be uniquely identified since one can only identify $GG^\top$ based on the distribution of its solution process [17, 28]. Secondly, the identifiability of an SDE's generator suffices for reliable causal inferences for this system. Note that, in the context of SDEs, the main task of causal analysis is to identify the post-intervention distributions from the observational distribution. As proposed in [17], for an SDE satisfying specific criteria, the post-intervention distributions are identifiable from the generator of this SDE. Consequently, the intricate task of unraveling causality can be decomposed into two constituent components through the generator. This paper aims to uncover conditions under which the generator of a linear SDE attains identifiability from the observational distribution. By establishing these identifiability conditions, we can effectively address the causality task for linear SDEs.

In this paper, we present a sufficient and necessary identifiability condition for the generator of linear SDEs with additive noise (1), along with a sufficient identifiability condition for the generator of linear SDEs with multiplicative noise (2).

## 2 Background knowledge

In this section, we introduce some background knowledge of linear SDEs. In addition, we provide a concise overview of the causal interpretation of SDEs, which is a critical aspect of understanding the nature and dynamics of these equations. This interpretation also forms a strong basis for the motivation of this research.

## 2.1 Background knowledge of linear SDEs

The solution to the SDE (1) can be explicitly expressed as (cf. [50]):

$$X_t := X(t; x_0, A, G) = e^{At}x_0 + \int_0^t e^{A(t-s)}G dW_s. \tag{3}$$

Note that in the context of our study, the solution stands for the strong solution, refer to [22] for its detailed definition.

In general, obtaining an explicit expression for the solution to the SDE (2) is not feasible. In fact, an explicit solution can be obtained when the matrices $A, G_1, \ldots, G_k$ commute, that is when

$$AG_k = G_k A \quad \text{and} \quad G_k G_l = G_l G_k \tag{4}$$

holds for all $k, l = 1, \ldots, m$ (cf. [25]). However, the conditions described in (4) are too restrictive and impractical. Therefore, this study will focus on the general case of the SDE (2).

We know that both the SDE (1) and the SDE (2) admit unique solutions that manifest as continuous stochastic processes [22]. A $d$-dimensional stochastic process is a collection of $\mathbb{R}^d$-valued random variables, denoted as $X = \{X_t; 0 \leqslant t < \infty\}$ defined on some probability space. When comparing two stochastic processes, $X$ and $\tilde{X}$, that are defined on the same probability space $(\Omega, \mathcal{F}, \mathbb{P})$, various notions of equality may be considered. In this study, we adopt the notion of equality with respect to their distributions, which is a weaker requirement than strict equivalence, see [22] for relevant notions. We now present the definition of the distribution of a stochastic process.

**Definition 2.1.** *Let $X$ be a random variable on a probability space $(\Omega, \mathcal{F}, \mathbb{P})$ with values in a measurable space $(S, \mathcal{B}(S))$, i.e., the function $X : \Omega \rightarrow S$ is $\mathcal{F}/\mathcal{B}(S)$-measurable. Then, the distribution of the random variable $X$ is the probability measure $P^X$ on $(S, \mathcal{B}(S))$ given by*

$$P^X(B) = \mathbb{P}(X \in B) = \mathbb{P}\{\omega \in \Omega : X(\omega) \in B\}, \quad B \in \mathcal{B}(S).$$

*When $X := \{X_t; 0 \leqslant t < \infty\}$ is a continuous stochastic process on $(\Omega, \mathcal{F}, \mathbb{P})$, and $S = C[0, \infty)$, such an $X$ can be regarded as a random variable on $(\Omega, \mathcal{F}, \mathbb{P})$ with values in $(C[0, \infty), \mathcal{B}(C[0, \infty)))$, and $P^X$ is called the distribution of $X$. Here $C[0, \infty)$ stands for the space of all continuous, real-valued functions on $[0, \infty]$.*

It is noteworthy that the distribution of a continuous process can be uniquely determined by its finite-dimensional distributions. Hence, if two stochastic processes, labelled as $X$ and $\tilde{X}$, share identical finite-dimensional distributions, they are regarded as equivalent in distribution, denoted by $X \overset{\mathrm{d}}{=} \tilde{X}$. Relevant concepts and theories regarding this property can be found in [22].

The generator of a stochastic process is typically represented by a differential operator that acts on functions. It provides information about how a function evolves over time in the context of the underlying stochastic process. Mathematically, the generator of a stochastic process $X_t$ can be defined as

$$(\mathcal{L}f)(x) = \lim_{s \to 0} \frac{\mathbb{E}[f(X_{t+s}) - f(X_t)|X_t = x]}{s},$$

where $f$ is a suitably regular function.

In the following, we present the generator of the SDEs under consideration. Obviously, both the SDE (1) and the SDE (2) conform to the general form:

$$dX_t = b(X_t)dt + \sigma(X_t)dW_t, \quad X_0 = x_0. \tag{5}$$

where $b$ and $\sigma$ are locally Lipschitz continuous in the space variable $x$. The generator $\mathcal{L}$ of the SDE (5) can be explicitly computed by utilizing Itô's formula (cf. [50]).

**Proposition 2.1.** *Let $X$ be a stochastic process defined by the SDE (5). The generator $\mathcal{L}$ of $X$ on $C_b^2(\mathbb{R}^d)$ is given by*

$$(\mathcal{L}f)(x) := \sum_{i=1}^d b_i(x)\frac{\partial f(x)}{\partial x_i} + \frac{1}{2}\sum_{i,j=1}^d c_{ij}(x)\frac{\partial^2 f(x)}{\partial x_i \partial x_j} \tag{6}$$

*for $f \in C_b^2(\mathbb{R}^d)$ and $x \in \mathbb{R}^d$, where $c(x) = \sigma(x) \cdot \sigma(x)^\top$ is a $d \times d$ matrix, and $C_b^2(\mathbb{R}^d)$ denotes the space of continuous functions on $\mathbb{R}^d$ that have bounded derivatives up to order two.*

## 2.2 Causal interpretation of SDEs

An important motivation for the identification of the generator of an SDE lies in the desire to infer reliable causality within dynamic models described by SDEs. In this subsection, we aim to provide some necessary background knowledge on the causal interpretation of SDEs. Consider the general SDE framework described as:

$$dX_t = a(X_t)dZ_t\,, \quad X_0 = x_0\,, \tag{7}$$

where $Z$ is a $p$-dimensional semimartingale and $a : \mathbb{R}^d \to \mathbb{R}^{d \times p}$ is a continuous mapping. By writing the SDE (7) in integral form

$$X_t^i = x_0^i + \sum_{j=1}^p \int_0^t a_{ij}(X_s)dZ_s^j\,, \quad i \leqslant d\,. \tag{8}$$

The authors of [17] proposed a mathematical definition of the SDE resulting from an intervention to the SDE (8). In the following, $X^{(-l)}$ denotes the $(d-1)$-dimensional vector that results from the removal of the $l$-th coordinate of $X \in \mathbb{R}^d$.

**Definition 2.2.** *[17, Definition 2.4.] Consider some $l \leqslant d$ and $\zeta : \mathbb{R}^{d-1} \to \mathbb{R}$. The SDE arising from* (8) *under the intervention $X_t^l := \zeta(X_t^{(-l)})$ is the $(d-1)$-dimensional equation*

$$(Y^{(-l)})_t^i = x_0^i + \sum_{j=1}^p \int_0^t b_{ij}(Y_s^{(-l)})dZ_s^j\,, \quad i \neq l\,, \tag{9}$$

*where $b : \mathbb{R}^{d-1} \to \mathbb{R}^{(d-1) \times p}$ is defined by $b_{ij}(y) = a_{ij}(y_1, \ldots, \zeta(y), \ldots, y_d)$ for $i \neq l$ and $j \leqslant p$ and the $\zeta(y)$ is on the $l$-th coordinate.*

Definition 2.2 presents a natural approach to defining how interventions should affect dynamic systems governed by SDEs. We adopt the same notations as used in [17]. Assuming (8) and (9) have unique solutions for all interventions, we refer to (8) as the observational SDE, to its solution as the observational process, to the distribution of its solution as observational distribution, to (9) as the post-intervention SDE, to the solution of (9) as the post-intervention process, and to the distribution of the solution of (9) as the post-intervention distribution. The authors in [17] related Definition 2.2 to mainstream causal concepts by establishing a mathematical connection between SDEs and structural equation models (SEMs). Specifically, the authors showed that under regularity assumptions, the solution to the post-intervention SDE is equal to the limit of a sequence of interventions in SEMs based on the Euler scheme of the observational SDE. Despite the fact that the parameters of the SDEs are generally not identifiable from the observational distribution, the post-intervention distributions can be identified, thus enabling causal inference of the system. To this end, Sokol and Hansen [17] derived a condition under which the generator associated with the observational SDE allows for the identification of the post-intervention distributions. We present the corresponding theory as follows.

**Lemma 2.1.** *[17, Theorem 5.3.] Consider the SDEs*

$$dX_t = a(X_t)dZ_t\,, \quad X_0 = x_0\,, \tag{10}$$

$$d\tilde{X}_t = \tilde{a}(\tilde{X}_t)d\tilde{Z}_t\,, \quad \tilde{X}_0 = \tilde{x}_0\,, \tag{11}$$

*where $Z$ is a $p$-dimensional Lévy process and $\tilde{Z}$ is a $\tilde{p}$-dimensional Lévy process. Assume that* (10) *and* (11) *have the same generator, that $a : \mathbb{R}^d \to \mathbb{R}^{d \times p}$ and $\zeta : \mathbb{R}^{d-1} \to \mathbb{R}$ are Lipschitz and that the initial values have the same distribution. Then the post-intervention distributions of doing $X^l := \zeta(X^{(-l)})$ in* (10) *and doing $\tilde{X}^l := \zeta(\tilde{X}^{(-l)})$ in* (11) *are equal for any choice of $\zeta$ and $l$.*

A main task in the causality research community is to uncover the conditions under which the post-intervention distributions are identifiable from the observational distribution. In the context of dynamic systems modelled in SDEs, similar conditions need to be derived. Lemma 2.1 establishes that, for SDEs with a Lévy process as the driving noise, the post-intervention distributions can be identifiable from the generator. Nevertheless, a gap remains between the observational distribution and the SDE generator's identifiability. This work aims to address this gap by providing conditions under which the generator is identifiable from the observational distribution.

## 3 Main results

In this section, we present some prerequisites first, and then we present the main theoretical results of our study, which include the condition for the identifiability of generator that is associated with the SDE (1) / SDE (2) from the distribution of the corresponding solution process.

### 3.1 Prerequisites

We first show that both the SDE (1) and the SDE (2) satisfy the conditions stated in Lemma 2.1.

**Lemma 3.1.** *Both the SDE (1) and the SDE (2) can be expressed as the form of (10), with $Z$ being a $p$-dimensional Lévy process, and $a : \mathbb{R}^d \to \mathbb{R}^{d \times p}$ being Lipschitz.*

The proof of Lemma 3.1 can be found in Appendix A.1. This lemma suggests that Lemma 2.1 applies to both the SDE (1) and the SDE (2), given that they meet the specified conditions. Therefore, for either SDE, deriving the conditions that allow for the generator to be identifiable from the observational distribution is sufficient. By applying Lemma 2.1, when the intervention function $\zeta$ is Lipschitz, the post-intervention distributions can be identified from the observational distribution under these conditions.

We then address the identifiability condition of the generator $\mathcal{L}$ defined by (6).

**Proposition 3.1.** *Let $\mathcal{L}$ and $\tilde{\mathcal{L}}$ be generators of stochastic processes defined by the form of the SDE (5) on $C_b^2(\mathbb{R}^d)$, where $\mathcal{L}$ is given by (6) and $\tilde{\mathcal{L}}$ is given by the same expression, with $\tilde{b}(x)$ and $\tilde{c}(x)$ substituted for $b(x)$ and $c(x)$. It then holds that the two generators $\mathcal{L} = \tilde{\mathcal{L}}$ if and only if $b(x) = \tilde{b}(x)$ and $c(x) = \tilde{c}(x)$ for all $x \in \mathbb{R}^d$.*

The proof of Proposition 3.1 can be found in Appendix A.2. This proposition states that for stochastic processes defined by the SDE (5), the generator is identifiable from functions associated with its coefficients: $b(x)$ and $c(x) = \sigma(x) \cdot \sigma(x)^\top$.

### 3.2 Conditions for identifying generators of linear SDEs with additive noise

Expressing the SDE (1) in the form given by (5) yields $b(x) = Ax$ and $c(x) = GG^\top$. Therefore, based on Proposition 3.1, we define the identifiability of the generator of the SDE (1) as follows.

**Definition 3.1** (($x_0, A, G$)-identifiability). *For $x_0 \in \mathbb{R}^d$, $A \in \mathbb{R}^{d \times d}$ and $G \in \mathbb{R}^{d \times m}$, the generator of the SDE (1) is said to be identifiable from $x_0$, if for all $\tilde{A} \in \mathbb{R}^{d \times d}$ and all $\tilde{G} \in \mathbb{R}^{d \times m}$, with $(A, GG^\top) \neq (\tilde{A}, \tilde{G}\tilde{G}^\top)$, it holds that $X(\cdot; x_0, A, G)^2 \overset{\mathrm{d}}{\neq} X(\cdot; x_0, \tilde{A}, \tilde{G})$.*

In the following, we begin by introducing two lemmas that serve as the foundation for deriving our main identifiability theorem.

**Lemma 3.2.** *For $x_0 \in \mathbb{R}^d$, $A, \tilde{A} \in \mathbb{R}^{d \times d}$ and $G, \tilde{G} \in \mathbb{R}^{d \times m}$, let $X_t := X(t; x_0, A, G)$, $\tilde{X}_t := X(t; x_0, \tilde{A}, \tilde{G})$, then $X(\cdot; x_0, A, G) \overset{\mathrm{d}}{=} X(\cdot; x_0, \tilde{A}, \tilde{G})$ if and only if the mean $\mathbb{E}[X_t] = \mathbb{E}[\tilde{X}_t]$ and the covariance $\mathbb{E}\{(X_{t+h} - \mathbb{E}[X_{t+h}])(X_t - \mathbb{E}[X_t])^\top\} = \mathbb{E}\{(\tilde{X}_{t+h} - \mathbb{E}[\tilde{X}_{t+h}])(\tilde{X}_t - \mathbb{E}[\tilde{X}_t])^\top\}$ for all $0 \leqslant t < \infty$ and $0 \leqslant h < \infty$.*

The proof of Lemma 3.2 can be found in Appendix A.3. This lemma states that for stochastic processes modelled by the SDE (1), the equality of the distribution of two processes can be deconstructed as the equality of the mean and covariance of the state variables at all time points. Calculation shows

$$
\begin{aligned}
\mathbb{E}[X_t] &= e^{At} x_0 \,, \\
V(t, t+h) &:= \mathbb{E}\{(X_{t+h} - \mathbb{E}[X_{t+h}])(X_t - \mathbb{E}[X_t])^\top\} \\
&= e^{Ah} V(t) \,,
\end{aligned}
\tag{12}
$$

where $V(t) := V(t, t)$. Please refer to the proof A.5 of Theorem 3.4 for the detailed calculations. It can be easily checked that $\mathbb{E}[X_t]$ follows the linear ordinary differential equation (ODE)

$$
\dot{m}(t) = Am(t), \quad m(0) = x_0 \,,
\tag{13}
$$

where $\dot{m}(t)$ denotes the first derivative of function $m(t)$ with respect to time $t$. Similarly, each column of the covariance $V(t, t+h)$ also follows the linear ODE (13) but with a different initial state: the corresponding column of $V(t)$. This observation allows us to leverage not only the characteristics of the SDE (1), but also the established theories [43, 57] on identifiability analysis for the ODE (13), to derive the identifiability conditions for the generator of the SDE (1).

---

$^2 X(\cdot; x_0, A, G) = \{X(t; x_0, A, G) : 0 \leqslant t < \infty\}$

We adopt the same setting as in [43], discussing the case where $A$ has distinct eigenvalues. Because random matrix theory suggests that almost every $A \in \mathbb{R}^{d \times d}$ has $d$ distinct eigenvalues with respect to the Lebesgue measure on $\mathbb{R}^{d \times d}$ [43]. And the Jordan decomposition of such a matrix $A$ follows a straightforward form which is helpful for deriving the geometric interpretation of the proposed identifiability condition. The Jordan decomposition can be expressed as $A = Q \Lambda Q^{-1}$, where

$$\Lambda = \begin{bmatrix} J_1 & & \\ & \ddots & \\ & & J_K \end{bmatrix}, \text{with } J_k = \begin{cases} \lambda_k, & \text{if } k = 1, \ldots, K_1, \\ \begin{bmatrix} a_k & -b_k \\ b_k & a_k \end{bmatrix}, & \text{if } k = K_1 + 1, \ldots, K. \end{cases}$$

$$Q = [Q_1 | \ldots | Q_K] = [v_1 | \ldots | v_d],$$

$$Q_k = \begin{cases} v_k, & \text{if } k = 1, \ldots, K_1, \\ [v_{2k-K_1-1} | v_{2k-K_1}], & \text{if } k = K_1 + 1, \ldots, K, \end{cases}$$

where $\lambda_k$ is a real eigenvalue of $A$ and $v_k$ is the corresponding eigenvector of $\lambda_k$, for $k = 1, \ldots, K_1$. For $k = K_1 + 1, \ldots, K$, $[v_{2k-K_1-1} | v_{2k-K_1}]$ are the corresponding "eigenvectors" of complex eigenvalues $a_k \pm b_k i$. Inspired by [43, Definition 2.3., Lemma 2.3.], we establish the following Lemma.

**Lemma 3.3.** *Assuming $A \in \mathbb{R}^{d \times d}$ has $d$ distinct eigenvalues, with Jordan decomposition $A = Q\Lambda Q^{-1}$. Let $\gamma_j \in \mathbb{R}^d$ and $\tilde{\gamma}_j := Q^{-1}\gamma_j \in \mathbb{R}^d$ for all $j = 1, \ldots, n$ with $n \geqslant 2$. We define*

$$w_{j,k} := \begin{cases} \tilde{\gamma}_{j,k} \in \mathbb{R}^1, & \text{for } k = 1, \ldots, K_1, \\ (\tilde{\gamma}_{j,2k-K_1-1}, \tilde{\gamma}_{j,2k-K_1})^\top \in \mathbb{R}^2, & \text{for } k = K_1 + 1, \ldots, K, \end{cases}$$

*where $\tilde{\gamma}_{j,k}$ denotes the $k$-th entry of $\tilde{\gamma}_j$. $\text{rank}([\gamma_1|A\gamma_1|\ldots|A^{d-1}\gamma_1|\ldots|\gamma_n|A\gamma_n|\ldots|A^{d-1}\gamma_n]) < d$ if and only if there exists $k \in \{1, \ldots, K\}$, such that $|w_{j,k}| = 0$ for all $j = 1, \ldots, n$, where $|w_{j,k}|$ is the absolute value of $w_{j,k}$ for $k = 1, \ldots, K_1$, and the Euclidean norm of $w_{j,k}$ for $k = K_1+1, \ldots, K$.*

The proof of Lemma 3.3 can be found in Appendix A.4. From a geometric perspective, $\gamma_j$ can be decomposed into a linear combination of $Q_k$'s

$$\gamma_j = Q\tilde{\gamma}_j = \sum_{k=1}^{K} Q_k w_{j,k}.$$

Let $L_k := \text{span}(Q_k)$. According to [43, Theorem 2.2], each $L_k$ is an $A$-invariant subspace of $\mathbb{R}^d$. Recall that a space $L$ is called $A$-invariant, if for all $\gamma \in L$, $A\gamma \in L$. We say $L$ is a proper subspace of $\mathbb{R}^d$ if $L \subset \mathbb{R}^d$ and $L \neq \mathbb{R}^d$. If $|w_{j,k}| = 0$ (i.e., $w_{j,k} = 0$ in $\mathbb{R}^1$ or $\mathbb{R}^2$), then $\gamma_j$ does not contain any information from $L_k$. In this case, $\gamma_j$ is contained in an $A$-invariant proper subspace of $\mathbb{R}^d$ that excludes $L_k$, denoted as $L_{-k}$. It is worth emphasizing that $L_{-k} \subset \mathbb{R}^d$ is indeed a **proper** subspace of $\mathbb{R}^d$. This further implies that the trajectory of the ODE (13) generated from initial state $\gamma_j$ is confined to $L_{-k}$ [57, Lemma 3.2]. Lemma 3.3 indicates that if $\text{rank}([\gamma_1|A\gamma_1|\ldots|A^{d-1}\gamma_1|\ldots|\gamma_n|A\gamma_n|\ldots|A^{d-1}\gamma_n]) < d$ then all $\gamma_j$ for $j = 1, \ldots, n$ are confined to an $A$-invariant proper subspace of $\mathbb{R}^d$, denoted as $L$. Therefore, all trajectories of the ODE (13) generated from initial states $\gamma_j$ are also confined to $L$. Furthermore, based on the identifiability conditions proposed in [57], the ODE (13) is not identifiable from observational data collected in these trajectories. This lemma provides an approach to interpreting our identifiability conditions from a geometric perspective.

Now we are ready to present our main theorem.

**Theorem 3.4.** *Let $x_0 \in \mathbb{R}^d$ be fixed. Assuming that the matrix $A$ in the SDE (1) has $d$ distinct eigenvalues. The generator of the SDE (1) is identifiable from $x_0$ if and only if*

$$\text{rank}([x_0|Ax_0|\ldots|A^{d-1}x_0|H_{\cdot 1}|AH_{\cdot 1}|\ldots|A^{d-1}H_{\cdot 1}|\ldots|H_{\cdot d}|AH_{\cdot d}|\ldots|A^{d-1}H_{\cdot d}]) = d, \quad (14)$$

*where $H := GG^T$, and $H_{\cdot j}$ stands for the $j$-th column vector of matrix $H$, for all $j = 1, \cdots, d$.*

The proof of Theorem 3.4 can be found in Appendix A.5. The condition in Theorem 3.4 is both sufficient and necessary when the matrix $A$ has distinct eigenvalues. It is worth noting that almost every $A \in \mathbb{R}^{d \times d}$ has $d$ distinct eigenvalues concerning the Lebesgue measure on $\mathbb{R}^{d \times d}$. Hence, this condition is both sufficient and necessary for almost every $A$ in $\mathbb{R}^{d \times d}$. However, in cases where $A$ has repetitive eigenvalues, this condition is solely sufficient and not necessary.

**Remark.** The identifiability condition stated in Theorem 3.4 is generic, that is, let

$$S := \{(x_0, A, G) \in \mathbb{R}^{d+d^2+dm} : \text{condition (14) is violated}\},$$

$S$ has Lebesgue measure zero in $\mathbb{R}^{d+d^2+dm}$. Refer to Appendix B.2 for the detailed proof.

From the geometric perspective, suppose matrix $A$ has distinct eigenvalues, the generator of the SDE (1) is identifiable from $x_0$ when not all of the vectors: $x_0, H_{.1}, \ldots, H_{.d}$ are confined to an $A$-invariant **proper** subspace of $\mathbb{R}^d$. A key finding is that when all the vectors $H_{.j}, j = 1, \ldots, d$ are confined to an $A$-invariant proper subspace $L$ of $\mathbb{R}^d$, each column of the covariance matrix $V(t)$ in Equation (12) is also confined to $L$, for all $0 \leqslant t < \infty$. Thus, the identifiability of the generator of the SDE (1) can be fully determined by $x_0$ and the system parameters $(A, GG^\top)$. Further details can be found in the proof A.5 of Theorem 3.4.

By rearranging the matrix in (14), the identifiability condition can also be expressed as

$$\text{rank}([x_0|Ax_0|\ldots|A^{d-1}x_0|GG^\top|AGG^\top|\ldots|A^{d-1}GG^\top]) = d. \tag{15}$$

Based on the identifiability condition (15), we derive the following corollary.

**Corollary 3.4.1.** *Let $x_0 \in \mathbb{R}^d$ be fixed. If $\text{rank}([G|AG|\ldots|A^{d-1}G]) = d$, then the generator of the SDE (1) is identifiable from $x_0$.*

The proof of Corollary 3.4.1 can be found in Appendix A.6. This corollary indicates that the generator of the SDE (1) is identifiable from **any** initial state $x_0 \in \mathbb{R}^d$ when the pair $[A, G]$ is controllable $(\text{rank}([G|AG|\ldots|A^{d-1}G]) = d)$. Notably, this identifiability condition is stricter than that proposed in Theorem 3.4, as it does not use the information of $x_0$.

## 3.3 Conditions for identifying generators of linear SDEs with multiplicative noise

Expressing the SDE (2) in the form given by (5) yields $b(x) = Ax$ and $\sigma(x) = [G_1 x|\ldots|G_m x] \in \mathbb{R}^{d \times m}$, thus, $c(x) = \sigma(x)\sigma(x)^\top = \sum_{k=1}^m G_k x x^\top G_k^\top$. Let $X(t; x_0, A, \{G_k\}_{k=1}^m)$ denote the solution to the SDE (2), then based on Proposition 3.1, we define the identifiability of the generator of the SDE (2) as follows.

**Definition 3.2** $((x_0, A, \{G_k\}_{k=1}^m)$-identifiability). *For $x_0 \in \mathbb{R}^d, A, G_k \in \mathbb{R}^{d \times d}$ for all $k = 1, \ldots, m$, the generator of the SDE (2) is said to be identifiable from $x_0$, if for all $\tilde{A}, \tilde{G}_k \in \mathbb{R}^{d \times d}$, there exists an $x \in \mathbb{R}^d$, such that $(A, \sum_{k=1}^m G_k x x^\top G_k^\top) \neq (\tilde{A}, \sum_{k=1}^m \tilde{G}_k x x^\top \tilde{G}_k^\top)$, it holds that $X(\cdot; x_0, A, \{G_k\}_{k=1}^m) \overset{\mathrm{d}}{\neq} X(\cdot; x_0, \tilde{A}, \{\tilde{G}_k\}_{k=1}^m)$.*

Based on Definition 3.2, we present the identifiability condition for the generator of the SDE (2).

**Theorem 3.5.** *Let $x_0 \in \mathbb{R}^d$ be fixed. The generator of the SDE (2) is identifiable from $x_0$ if the following conditions are satisfied:*

   A1  $\text{rank}([x_0|Ax_0|\ldots|A^{d-1}x_0]) = d$,

   A2  $\text{rank}([v|\mathcal{A}v|\ldots|\mathcal{A}^{(d^2+d-2)/2}v]) = (d^2 + d)/2$,

*where $\mathcal{A} = A \oplus A + \sum_{k=1}^m G_k \otimes G_k \in \mathbb{R}^{d^2 \times d^2}$, $\oplus$ denotes Kronecker sum and $\otimes$ denotes Kronecker product, $v$ is a $d^2$-dimensional vector defined by $v := \text{vec}(x_0 x_0^\top)$, where $\text{vec}(M)$ denotes the vectorization of matrix $M$.*

The proof of Theorem 3.5 can be found in Appendix A.7. This condition is only sufficient but not necessary. Specifically, condition A1 guarantees that matrix $A$ is identifiable, and once $A$ is identifiable, condition A2 ensures that the identifiability of $\sum_{k=1}^m G_k x x^\top G_k^\top$ holds for all $x \in \mathbb{R}^d$.

**Remark.** The identifiability condition stated in Theorem 3.5 is generic, that is, let

$$S := \{(x_0, A, \{G_k\}_{k=1}^m) \in \mathbb{R}^{d+(m+1)d^2} : \text{either condition A1 or A2 in Theorem 3.5 is violated}\},$$

$S$ has Lebesgue measure zero in $\mathbb{R}^{d+(m+1)d^2}$. This signifies that the conditions are satisfied for most of the combinations of $x_0$, $A$ and $G_k$'s, except for those that lie in a set of Lebesgue measure zero. The corresponding proposition and detailed proof can be found in Appendix B.1.

Since obtaining an explicit solution for the SDE (2) is generally infeasible, we resort to utilizing the first- and second-order moments of this SDE to derive the identifiability conditions. Let $m(t) := \mathbb{E}[X_t]$ and $P(t) := \mathbb{E}[X_t X_t^\top]$, it is known that these moments satisfy ODE systems. Specifically, $m(t)$ satisfies the ODE (13), while $P(t)$ satisfies the following ODE (cf. [56]):

$$\dot{P}(t) = AP(t) + P(t)A^\top + \sum_{k=1}^{m} G_k P(t) G_k^\top, \quad P(0) = x_0 x_0^\top. \tag{16}$$

An important trick to deal with the ODE (16) is to vectorize $P(t)$, then it can be expressed as:

$$\text{vec}(\dot{P}(t)) = \mathcal{A}\text{vec}(P(t)), \quad \text{vec}(P(0)) = v, \tag{17}$$

where $\mathcal{A}$ and $v$ are defined in Theorem 3.5. In fact, the ODE (17) follows the same mathematical structure as that of the ODE (13), which is known as homogeneous linear ODEs. Thus, in addition to the inherent properties of the SDE (2), we also employ some existing identifiability theories for homogeneous linear ODEs to establish the identifiability condition for the generator of the SDE (2).

From the geometric perspective, condition A1 indicates that the initial state $x_0$ is not confined to an $A$-invariant **proper** subspace of $\mathbb{R}^d$ [57, Lemma 3.1.]. And condition A2 implies that the vectorization of $x_0 x_0^\top$ is not confined to an $\mathcal{A}$-invariant **proper** subspace of $W$, with $W \subset \mathbb{R}^{d^2}$, and $\dim(W) = (d^2 + d)/2$, where $\dim(W)$ denotes the dimension of the subspace $W$, that is the number of vectors in any basis for $W$. In particular, one can construct a basis for $W$ as follows:

$$\{\text{vec}(E_{11}), \text{vec}(E_{21}), \text{vec}(E_{22}), \ldots, \text{vec}(E_{dd})\},$$

where $E_{ij}$ denotes a $d \times d$ matrix whose $ij$-th and $ji$-th elements are 1, and all other elements are 0, for all $i, j = 1, \ldots, d$ and $i \geqslant j$. Refer to the proof A.7 of Theorem 3.5 for more details.

## 4    Simulations and examples

In order to assess the validity of the identifiability conditions established in Section 3, we present the results of simulations. Specifically, we consider SDEs with system parameters that either satisfy or violate the proposed identifiability conditions. We then apply the maximum likelihood estimation (MLE) method to estimate the system parameters from discrete observations sampled from the corresponding SDE. The accuracy of the resulting parameter estimates serves as an indicator of the validity of the proposed identifiability conditions.

**Simulations.** We conduct five sets of simulations, which include one identifiable case and one unidentifiable case for the SDE (1), and one identifiable case and two unidentifiable cases with either condition A1 or A2 in Theorem 3.5 unsatisfied for the SDE (2). We set both the system dimension, $d$, and the Brownian motion dimension, $m$, to 2. Details on the true underlying system parameters for the SDEs can be found in Appendix C. We simulate observations from the true SDEs for each of the five cases under investigation. Specifically, the simulations are carried out for different numbers of trajectories ($N$), with 50 equally-spaced observations sampled on each trajectory from the time interval $[0, 1]$. We employ the Euler-Maruyama (EM) method [34], a widely used numerical scheme for simulating SDEs, to generate the observations.

**Estimation.** We use MLE [38, 50] to estimate the system parameters. The MLE method requires knowledge of the transition probability density function (pdf) that governs the evolution of the system. For the specific case of the SDE (1), the transition density follows a Gaussian distribution, which can be computed analytically based on the system's drift and diffusion coefficients (cf. [50]). To compute the covariance, we employ the commonly used matrix fraction decomposition method [4, 49, 50]. However, in general, the transition pdf of the SDE (2) cannot be obtained analytically due to the lack of a closed-form solution. To address this issue, we implement the Euler-Maruyama approach [32, 34], which has been shown to be effective in approximating the transition pdf of SDEs.

**Metric.** We adopt the commonly used metric, mean squared error (MSE), to assess the accuracy of the parameter estimates. To ensure reliable estimation outcomes, we perform 100 independent random replications for each configuration and report the mean and variance of their MSEs.

**Results analysis.** Table 1 and Table 2 present the simulation results for the SDE (1) and the SDE (2), respectively. In Table 1, the simulation results demonstrate that in the identifiable case, as the number of trajectories $N$ increases, the MSE for both $A$ and $GG^\top$ decreases and approaches zero. However, in the unidentifiable case, where the identifiable condition (14) stated in Theorem 3.4 is not

Table 1: Simulation results of the SDE (1)

| $N$ | Identifiable | | Unidentifiable | |
| --- | --- | --- | --- | --- |
| | MSE-$A$ | MSE-$GG^\top$ | MSE-$A$ | MSE-$GG^\top$ |
| 5 | $0.0117 \pm 0.0115$ | $5.28\text{E-}05 \pm 4.39\text{E-}05$ | $3.66 \pm 0.10$ | $0.05 \pm 0.03$ |
| 10 | $0.0063 \pm 0.0061$ | $2.39\text{E-}05 \pm 1.82\text{E-}05$ | $3.88 \pm 0.06$ | $0.64 \pm 0.59$ |
| 20 | $0.0029 \pm 0.0027$ | $1.87\text{E-}05 \pm 1.51\text{E-}05$ | $3.70 \pm 0.06$ | $0.09 \pm 0.07$ |
| 50 | $0.0013 \pm 0.0010$ | $8.00\text{E-}06 \pm 5.68\text{E-}06$ | $3.76 \pm 0.07$ | $0.11 \pm 0.08$ |
| 100 | $0.0007 \pm 0.0004$ | $4.34\text{E-}06 \pm 2.70\text{E-}06$ | $3.66 \pm 0.02$ | $2.09 \pm 1.98$ |

satisfied, the MSE for both $A$ and $GG^\top$ remains high regardless of the number of trajectories. These findings provide strong empirical evidence supporting the validity of the identifiability condition proposed in Theorem 3.4. The simulation results presented in Table 2 show that in the identifiable case, the MSE for both $A$ and $Gsx$ decreases and approaches zero with the increase of the number of trajectories $N$. Here, $Gsx := \sum_{k=1}^{m} G_k xx^\top G_k^\top$, where $x$ is a randomly generated vector from $\mathbb{R}^2$ (in these simulations, $x = [1.33, 0.72]^\top$). Interestingly, even in unidentifiable case 1, the MSE for both $A$ and $Gsx$ decreases with an increasing number of trajectories $N$, indicating that the generator of the SDE utilized in this particular case is still identifiable, although a larger number of trajectories is required compared to the identifiable case to achieve the same level of accuracy. This result is reasonable, because it aligns with our understanding that condition A1 is only sufficient but not necessary for identifying $A$, as the lack of an explicit solution for the SDE (2) results in condition A1 not incorporating any information from $G_k$'s. The identifiability condition derived for the SDE (1) in Theorem 3.4 leverages the information of $G$, similarly, if information regarding $G_k$'s is available, a weaker condition for identifying $A$ could be obtained. For illustration, in Appendix E, we present such a condition assuming the SDE (2) has a closed-form solution. In the case of unidentifiable case 2, the MSE for $A$ decreases with an increasing number of trajectories $N$; however, the MSE for $Gsx$ remains high, indicating that $A$ is identifiable, while $Gsx$ is not, albeit requiring more trajectories compared to the identifiable case to achieve the same level of accuracy of $A$ (since the $Gsx$ is far away from its true underlying value). This finding is consistent with the derived identifiability condition, as condition A1 is sufficient to identify $A$, whereas condition A2 governs the identifiability of $Gsx$. Worth noting that in cases where neither condition A1 nor condition A2 is satisfied, the estimated parameters barely deviate from their initial values, implying poor estimation of both $A$ and $Gsx$. These results indicate the validity of the identifiability condition stated in Theorem 3.5.

Table 2: Simulation results of the SDE (2)

| $N$ | Identifiable | | Unidentifiable | | | |
| --- | --- | --- | --- | --- | --- | --- |
| | | | case1: A1-False, A2-True | | case2: A1-True, A2-False | |
| | MSE-$A$ | MSE-$Gsx$ | MSE-$A$ | MSE-$Gsx$ | MSE-$A$ | MSE-$Gsx$ |
| 10 | $0.069 \pm 0.061$ | $0.3647 \pm 0.3579$ | $0.509 \pm 0.499$ | $0.194 \pm 0.140$ | $2.562 \pm 2.522$ | $9763 \pm 8077$ |
| 20 | $0.047 \pm 0.045$ | $0.1769 \pm 0.1694$ | $0.195 \pm 0.180$ | $0.088 \pm 0.058$ | $0.967 \pm 0.904$ | $8353 \pm 6839$ |
| 50 | $0.018 \pm 0.018$ | $0.1703 \pm 0.1621$ | $0.132 \pm 0.131$ | $0.081 \pm 0.045$ | $0.423 \pm 0.410$ | $4779 \pm 4032$ |
| 100 | $0.006 \pm 0.006$ | $0.0015 \pm 0.0012$ | $0.065 \pm 0.065$ | $0.068 \pm 0.036$ | $0.207 \pm 0.198$ | $3569 \pm 3150$ |
| 500 | $0.001 \pm 0.001$ | $0.0004 \pm 0.0001$ | $0.008 \pm 0.008$ | $0.059 \pm 0.004$ | $0.046 \pm 0.046$ | $4490 \pm 3991$ |

**Illustrative instances of causal inference for linear SDEs (with interventions).** To illustrate how our proposed identifiability conditions can guarantee reliable causal inference for linear SDEs, we present examples corresponding to both the SDE (1) and the SDE (2). In these examples, we show that under our proposed identifiability conditions, the post-intervention distributions are identifiable from their corresponding observational distributions. Please refer to Appendix D.1 and D.2 for the details of the examples.

## 5  Related work

Most current studies on the identifiability analysis of SDEs are based on the Gaussian diffusion processes that conform to the form described in the SDE (1). In particular, the authors of [27, 28, 42]

have conducted research on the identifiability or asymptotic properties of parameter estimators of Gaussian diffusions in view of continuous observations of one trajectory, and have highlighted the need for the diffusion to be ergodic. A considerable amount of effort has also been directed towards the identifiability analysis of Gaussian diffusions, relying on the exact discrete models of the SDEs [6, 16, 23, 35, 41]. Typically, these studies involve transferring the continuous-time system described in the SDE (1) to a discrete-time model such as a vector autoregressive model, based on equally-spaced observations sampled from one trajectory, and then attempting to determine conditions under which $(A, GG^\top)$ is identifiable from the parameters of the corresponding exact discrete models. These conditions often have requirements on eigenvalues of $A$ among other conditions, such as requiring the eigenvalues to have only negative real parts, or the eigenvalues to be strictly real. Due to the limitation of the available observations (continuous or discrete observations located on one trajectory of the SDE system), the identifiability conditions proposed in these works are restrictive.

Causal modelling theories have been well-developed based on directed acyclic graphs (DAGs), which do not explicitly incorporate a time component [39]. In recent years, similar concepts of causality have been developed for dynamic systems operating in both discrete and continuous time. Discrete-time models, such as autoregressive processes, can be readily accommodated within the DAG-based framework [13, 14]. On the other hand, differential equations offer a natural framework for understanding causality in dynamic systems within the context of continuous-time processes [1, 52]. Consequently, considerable effort has been devoted to establishing a theoretical connection between causality and differential equations. In the deterministic case, Mooij et al. [37] and Rubenstein et al. [46] have established a mathematical link between ODEs and structural causal models (SCMs). Wang et al. [60] have proposed a method to infer the causal structure of linear ODEs. Turning to the stochastic case, Boogers and Mooij have built a bridge from random differential equations (RDEs) to SCMs [7], while Hansen and Sokol have proposed a causal interpretation of SDEs by establishing a connection between SDEs and SEMs [17].

# 6 Conclusion and discussion

In this paper, we present an investigation into the identifiability of the generators of linear SDEs under additive and multiplicative noise. Specifically, we derive the conditions that are fully built on system parameters and the initial state $x_0$, which enables the identification of a linear SDE's generator from the distribution of its solution process with a given fixed initial state. We establish that, under the proposed conditions, the post-intervention distribution is identifiable from the corresponding observational distribution for any Lipschitz intervention $\zeta$.

The main limitation of our work is that the practical verification of these identifiability conditions poses a challenge, as the true underlying system parameters are typically unavailable in real-world applications. Nevertheless, our study contributes to the understanding of the intrinsic structure of linear SDEs. By offering valuable insights into the identifiability aspects, our findings empower researchers and practitioners to employ models that satisfy the proposed conditions (e.g., through constrained parameter estimation) to learn real-world data while ensuring identifiability. We believe the paramount significance of this work lies in providing a systematic and rigorous causal interpretation of linear SDEs, which facilitates reliable causal inference for dynamic systems governed by such equations. It is worth noting that in our simulations, we employed the MLE method to estimate the system parameters. This necessitates the calculation of the transition pdf from one state to the successive state at each discrete temporal increment. Consequently, as the state dimension and Brownian motion dimension increase, the computational time is inevitably significantly increased, rendering the process quite time-consuming. To expedite parameter estimation for scenarios involving high dimensions, alternative estimation approaches are required. The development of a more efficient parameter estimation approach remains an important task in the realm of SDEs, representing a promising direction for our future research. We claim that this work does not present any foreseeable negative social impact.

# Acknowledgements

YW was supported by the Australian Government Research Training Program (RTP) Scholarship from the University of Melbourne. XG was supported by ARC DE210101352. MG was supported by ARC DE210101624.

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

# Appendix for "Generator Identification for Linear SDEs with Additive and Multiplicative Noise"

## A Detailed proofs

### A.1 Proof of Lemma 3.1

*Proof.* We start by presenting the mathematical definition of a Lévy process. (cf. [51])

**Definition A.1.** *A stochastic process $X := \{X_t : 0 \leqslant t < \infty\}$ is said to be a Lévy process if it satisfies the following properties:*

1. *$X_0 = 0$ almost surely;*

2. *Independence of increments: For any $0 \leqslant t_1 < t_2 < \ldots < t_n < \infty$, $X_{t_2} - X_{t_1}$, $X_{t_3} - X_{t_2}$, $\ldots$, $X_{t_n} - X_{t_{n-1}}$ are independent;*

3. *Stationary increments: For any $s < t$, $X_t - X_s$ is equal in distribution to $X_{t-s}$;*

4. *Continuity in probability: For any $\varepsilon > 0$ and $0 \leqslant t < \infty$ it holds that $\lim_{h \to 0} \mathbb{P}(|X_{t+h} - X_t| > \varepsilon) = 0$.*

In the following, we first show that the SDE (1) can be expressed as the form of (10), with $Z$ being a $p$-dimensional Lévy process and $a : \mathbb{R}^d \to \mathbb{R}^{d \times p}$ being Lipschitz. The first equation in the SDE (1) can be rearranged as

$$
\begin{aligned}
dX_t &= AX_t dt + G dW_t \\
&= [AX_t \quad G] \begin{bmatrix} dt \\ dW_t \end{bmatrix} \\
&= a(X_t) dZ_t \,,
\end{aligned}
\tag{18}
$$

with

$$
a(X_t) = [AX_t \quad G] \in \mathbb{R}^{d \times (m+1)} \,,
$$

and

$$
dZ_t = \begin{bmatrix} dt \\ dW_t \end{bmatrix} = \underbrace{\begin{bmatrix} 1 \\ 0_{m \times 1} \end{bmatrix}}_{r} dt + \underbrace{\begin{bmatrix} 0_{1 \times 1} & 0_{1 \times m} \\ 0_{m \times 1} & I_{m \times m} \end{bmatrix}}_{E} \begin{bmatrix} dW_{0,t} \\ dW_t \end{bmatrix} \,,
\tag{19}
$$

where $0_{i \times j}$ denotes an $i \times j$ zero matrix, let $\tilde{W} := \{\tilde{W}_t : 0 \leqslant t < \infty\}$ with $\tilde{W}_t = [W_{0,t}, W_{1,t}, \ldots, W_{m,t}]^\top$ denote a $(m+1)$-dimensional standard Brownian motion, then one can find a process $Z := \{Z_t : 0 \leqslant t < \infty\}$ with

$$
\begin{aligned}
Z_t &= rt + E\tilde{W}_t \,, \\
Z_0 &= 0 \,,
\end{aligned}
\tag{20}
$$

satisfying $dZ_t$ described in Equation (19). Then we will show that the process $Z$ described in (20) is a Lévy process, that is, it satisfies the four properties stated in Definition A.1.

**Property 1:** The first property is readily checked since $Z_0 = 0$.

**Property 2:** For any $0 \leqslant t_1 < t_2 < t_3 < \infty$,

$$
\begin{aligned}
Z_{t_2} - Z_{t_1} &= r(t_2 - t_1) + E(\tilde{W}_{t_2} - \tilde{W}_{t_1}) \\
&= \begin{bmatrix} t_2 - t_1 \\ 0_{m \times 1} \end{bmatrix} + \begin{bmatrix} 0 \\ W_{t_2} - W_{t_1} \end{bmatrix} \,.
\end{aligned}
$$

Similarly,

$$Z_{t_3} - Z_{t_2} = \begin{bmatrix} t_3 - t_2 \\ 0_{m \times 1} \end{bmatrix} + \begin{bmatrix} 0 \\ W_{t_3} - W_{t_2} \end{bmatrix}.$$

Since $W_{t_2} - W_{t_1}$ and $W_{t_3} - W_{t_2}$ are independent, $Z_{t_3} - Z_{t_2}$ and $Z_{t_2} - Z_{t_1}$ are independent.

**Property 3:** when $s < t$,

$$Z_t - Z_s = \begin{bmatrix} t - s \\ 0_{m \times 1} \end{bmatrix} + \begin{bmatrix} 0 \\ W_t - W_s \end{bmatrix}$$

$$\sim \mathcal{N} \left( \begin{bmatrix} t - s \\ 0_{m \times 1} \end{bmatrix}, \begin{bmatrix} 0_{1 \times 1} & 0_{1 \times m} \\ 0_{m \times 1} & (t - s) I_{m \times m} \end{bmatrix} \right).$$

And

$$Z_{t-s} = r(t - s) + E\tilde{W}_{t-s}$$

$$= \begin{bmatrix} t - s \\ 0_{m \times 1} \end{bmatrix} + \begin{bmatrix} 0 \\ W_{t-s} \end{bmatrix}$$

$$\sim \mathcal{N} \left( \begin{bmatrix} t - s \\ 0_{m \times 1} \end{bmatrix}, \begin{bmatrix} 0_{1 \times 1} & 0_{1 \times m} \\ 0_{m \times 1} & (t - s) I_{m \times m} \end{bmatrix} \right).$$

Therefore, property 3 is checked.

**Property 4:** Obviously, process $Z$ described in (20) is continuous with probability one at $t$ for all $0 \leqslant t < \infty$, therefore, $Z$ has continuity in probability.

Now that we have shown that process $Z$ is a $p$-dimensional Lévy process with $p = m + 1$. Then we will show that $a(X_t) = [AX_t \quad G]$ is Lipschitz.

$$\| a(X_t) - a(X_s) \|_F = \| [A(X_t - X_s) \quad 0] \|_F$$
$$= \| A(X_t - X_s) \|_2$$
$$\leqslant \| A \|_F \| X_t - X_s \|_2$$

where $\| M \|_F$ denotes the Frobenius norm of matrix $M$ and $\| v \|_2$ denotes the Euclidean norm of vector $v$. Now it is readily checked that function $a : \mathbb{R}^d \to \mathbb{R}^{d \times p}$ is Lipschitz.

Similarly, we will show that the SDE (2) can also be expressed as the form of (10), with $Z$ being a $p$-dimensional Lévy process and $a : \mathbb{R}^d \to \mathbb{R}^{d \times p}$ being Lipschitz. Let us rearrange the first equation in the SDE (2):

$$dX_t = AX_t dt + \sum_{k=1}^{m} G_k X_t dW_{k,t}$$

$$= [AX_t \quad G_1 X_t \quad \dots \quad G_m X_t] \begin{bmatrix} dt \\ dW_{1,t} \\ \vdots \\ dW_{m,t} \end{bmatrix}$$

$$= a(X_t) dZ_t.$$

Since the $dZ_t$ here has the same form as that of the SDE (1), we use the same process $Z$ described in Equation (20), which has been shown to be a Lévy process.

As for the function $a(X_t)$,

$$\| a(X_t) - a(X_s) \|_F = \| [A(X_t - X_s) \quad G_1(X_t - X_s) \quad \dots \quad G_m(X_t - X_s)] \|_F$$

$$\leqslant \| A(X_t - X_s) \|_2 + \sum_{k=1}^{m} \| G_k(X_t - X_s) \|_2$$

$$\leqslant \| A \|_F \| X_t - X_s \|_2 + \sum_{k=1}^{m} \| G_k \|_F \| X_t - X_s \|_2$$

$$= \left( \| A \|_F + \sum_{k=1}^{m} \| G_k \|_F \right) \| X_t - X_s \|_2,$$

it is readily checked that function $a : \mathbb{R}^d \to \mathbb{R}^{d \times p}$ is Lipschitz. $\qquad\square$

## A.2 Proof of Proposition 3.1

*Proof.* For the backward direction, when $b(x) = \tilde{b}(x)$ and $c(x) = \tilde{c}(x)$ for all $x \in \mathbb{R}^d$, it is obviously that $(\mathcal{L}f)(x) = (\tilde{\mathcal{L}}f)(x)$ for all $f \in C_b^2(\mathbb{R}^d)$ and $x \in \mathbb{R}^d$, that is $\mathcal{L} = \tilde{\mathcal{L}}$.

For the forward direction, since $(\mathcal{L}f)(x_1) = (\tilde{\mathcal{L}}f)(x_1)$ for all $f \in C_b^2(\mathbb{R}^d)$ and $x_1 \in \mathbb{R}^d$.

We first set
$$f(x) = x_p \,,$$
where $x_p$ denotes the $p$-th component of variable $x$. It is readily checked that
$$b_p(x_1) = \tilde{b}_p(x_1) \,,$$
for all $x_1 \in \mathbb{R}^d$ and $p = 1, \ldots, d$. As a result,
$$b(x) = \tilde{b}(x) \,, \quad \text{for all } x \in \mathbb{R}^d \,.$$

Then we set
$$f(x) = (x_p - x_{1p})(x_q - x_{1q}) \,,$$
where $x_{1p}$ denotes the $p$-th component of $x_1$. It is readily checked that
$$c_{pq}(x_1) = \tilde{c}_{pq}(x_1) \,,$$
for all $x_1 \in \mathbb{R}^d$ and $p, q = 1, \ldots, d$. Consequently,
$$c(x) = \tilde{c}(x) \,, \quad \text{for all } x \in \mathbb{R}^d \,.$$
$\qquad\square$

## A.3 Proof of Lemma 3.2

*Proof.* For the forward direction, since
$$X(\cdot; x_0, A, G) \stackrel{\mathrm{d}}{=} X(\cdot; x_0, \tilde{A}, \tilde{G}) \,,$$
one has
$$\mathbb{E}[X_t] = \mathbb{E}[\tilde{X}_t] \,, \quad \forall 0 \leqslant t < \infty \,.$$
Thus,
$$(X_t - \mathbb{E}[X_t])_{0 \leqslant t < \infty} \stackrel{\mathrm{d}}{=} (\tilde{X}_t - \mathbb{E}[\tilde{X}_t])_{0 \leqslant t < \infty} \,,$$
in particular, one has
$$\mathbb{E}\{(X_{t+h} - \mathbb{E}[X_{t+h}])(X_t - \mathbb{E}[X_t])^\top\} = \mathbb{E}\{(\tilde{X}_{t+h} - \mathbb{E}[\tilde{X}_{t+h}])(\tilde{X}_t - \mathbb{E}[\tilde{X}_t])^\top\} \quad \text{for all } 0 \leqslant t, h < \infty \,.$$

For the backward direction, we know that the solution of the SDE (1) is a Gaussian process. The distribution of a Gaussian process can be fully determined by its mean and covariance functions. Therefore, the two processes have the same distribution when the mean and covariance are the same for both processes for all $0 \leqslant t, h < \infty$. $\qquad\square$

## A.4 Proof of Lemma 3.3

*Proof.* For the forward direction, since
$$\mathrm{rank}([\gamma_1 | A\gamma_1 | \ldots | A^{d-1}\gamma_1 | \ldots | \gamma_n | A\gamma_n | \ldots | A^{d-1}\gamma_n]) < d \,,$$
then for all $l = [l_1, \ldots, l_n]^\top \in \mathbb{R}^n$,
$$\mathrm{rank}([\beta | A\beta | \ldots | A^{d-1}\beta]) < d \,,$$
where $\beta := l_1\gamma_1 + \ldots + l_n\gamma_n$. Consequently, the corresponding ODE system
$$\begin{aligned} \dot{x}(t) &= Ax(t) \,, \\ x(0) &= \beta \,, \end{aligned} \tag{21}$$

is not identifiable from $\beta$ by [57, Theorem 2.5.], where $\dot{x}(t)$ denotes the first derivative of $x(t)$ with respect to time $t$.

Let
$$\tilde{\beta} := Q^{-1}\beta \in \mathbb{R}^d\,,$$
and
$$w_k := \begin{cases} \tilde{\beta}_k \in \mathbb{R}^1\,, & \text{for } k = 1, \ldots, K_1\,, \\ (\tilde{\beta}_{2k-K_1-1}, \tilde{\beta}_{2k-K_1})^\top \in \mathbb{R}^2\,, & \text{for } k = K_1 + 1, \ldots, K\,. \end{cases}$$

Simple calculation shows that
$$\begin{aligned} \tilde{\beta} &= Q^{-1}\beta \\ &= Q^{-1}(l_1\gamma_1 + \ldots + l_n\gamma_n) \\ &= l_1\tilde{\gamma}_1 + \ldots + l_n\tilde{\gamma}_n\,, \end{aligned}$$

therefore, one has
$$w_k = l_1 w_{1,k} + \ldots + l_n w_{n,k}\,, \quad \text{for all } k \in \{1, \ldots, K\}\,. \tag{22}$$

By [43, Theorem 2.4], we know that for any $l \in \mathbb{R}^n$, there always exists $k \in \{1, \ldots, K\}$ such that $w_k = 0$ ($\in \mathbb{R}^1$ or $\mathbb{R}^2$) since the ODE (21) is not identifiable from initial state $\beta$. Next, we will show that this result is satisfied only when there exists a $k$ such that $w_{j,k} = 0$ ($\in \mathbb{R}^1$ or $\mathbb{R}^2$) for all $j = 1, \ldots, n$. Let us rearrange the Equation (22) as
$$\begin{bmatrix} w_{1,1} & \cdots & w_{n,1} \\ \vdots & \ddots & \vdots \\ w_{1,K} & \cdots & w_{n,K} \end{bmatrix} \begin{bmatrix} l_1 \\ \vdots \\ l_n \end{bmatrix} = \begin{bmatrix} w_1 \\ \vdots \\ w_K \end{bmatrix}\,,$$

assume for any $k \in \{1, \ldots, K\}$, $[w_{1,k}, \ldots, w_{n,k}]^\top \neq 0$, then there always exists a $l \in \mathbb{R}^n$ such that $w_k \neq 0$ for all $k = \{1, \ldots, K\}$. The reason is that under this circumstance, for any $k \in \{1, \ldots, K\}$, the set of $l$'s such that $w_k = 0$ has Lebesgue measure zero in $\mathbb{R}^n$. Therefore, the set of $l$'s such that there exists a $k$ such that $w_k = 0$ has Lebesgue measure zero in $\mathbb{R}^n$. This result creates a contradiction. Thus, there must exist a $k$, such that $[w_{1,k}, \ldots, w_{n,k}]^\top = 0$, that is $|w_{j,k}| = 0$ for all $j = 1, \ldots, n$.

For the backward direction, there exists $k$ such that $|w_{j,k}| = 0$ for all $j = 1, \ldots, n$, that is $w_{j,k} = 0$ ($\in \mathbb{R}^1$ or $\mathbb{R}^2$) for all $j = 1, \ldots, n$. Simple calculation shows that
$$\gamma_j = Q\tilde{\gamma}_j = \sum_{p=1}^{k-1} Q_p w_{j,p} + \sum_{p=k+1}^{K} Q_p w_{j,p}\,,$$
and
$$\begin{aligned} A^q\gamma_j &= Q\Lambda^q Q^{-1}\gamma_j \\ &= Q\Lambda^q\tilde{\gamma}_j \\ &= \sum_{p=1}^{k-1} Q_p J_p^q w_{j,p} + \sum_{p=k+1}^{K} Q_p J_p^q w_{j,p}\,, \end{aligned}$$

recall that
$$J_k = \begin{cases} \lambda_k\,, & \text{if } k = 1, \ldots, K_1\,, \\ \begin{bmatrix} a_k & -b_k \\ b_k & a_k \end{bmatrix}\,, & \text{if } k = K_1 + 1, \ldots, K\,. \end{cases}$$

Then matrix
$$\begin{aligned} M :&= [\gamma_1 | A\gamma_1 | \ldots | A^{d-1}\gamma_1 | \ldots | \gamma_n | A\gamma_n | \ldots | A^{d-1}\gamma_n] \\ &= Q_{-k}C\,, \end{aligned}$$
where
$$Q_{-k} = [Q_1 | \ldots | Q_{k-1} | Q_{k+1} | \ldots | Q_K]\,,$$

and matrix $C$ denotes:

$$
\begin{bmatrix}
w_{1,1} & J_1 w_{1,1} & \cdots & J_1^{d-1} w_{1,1} & \cdots & w_{n,1} & J_1 w_{n,1} & \cdots & J_1^{d-1} w_{n,1} \\
\vdots & \vdots & \ddots & \vdots & \ddots & \vdots & \vdots & \ddots & \vdots \\
w_{1,k-1} & J_{k-1} w_{1,k-1} & \cdots & J_{k-1}^{d-1} w_{1,k-1} & \cdots & w_{n,k-1} & J_{k-1} w_{n,k-1} & \cdots & J_{k-1}^{d-1} w_{n,k-1} \\
w_{1,k+1} & J_{k+1} w_{1,k+1} & \cdots & J_{k+1}^{d-1} w_{1,k+1} & \cdots & w_{n,k+1} & J_{k+1} w_{n,k+1} & \cdots & J_{k+1}^{d-1} w_{n,k+1} \\
\vdots & \vdots & \ddots & \vdots & \ddots & \vdots & \vdots & \ddots & \vdots \\
w_{1,K} & J_K w_{1,K} & \cdots & J_K^{d-1} w_{1,K} & \cdots & w_{n,K} & J_K w_{n,K} & \cdots & J_K^{d-1} w_{n,K}
\end{bmatrix}
$$

We know that

$$
\operatorname{rank}(M) = \operatorname{rank}(Q_{-k} C) \leqslant \min(\operatorname{rank}(Q_{-k}), \operatorname{rank}(C)).
$$

When $k \in \{1, \ldots, K_1\}$, $Q_{-k} \in \mathbb{R}^{d \times (d-1)}$, and $\operatorname{rank}(Q_{-k}) = d - 1$, while when $k \in \{K_1 + 1, \ldots, K\}$, $Q_{-k} \in \mathbb{R}^{d \times (d-2)}$, and $\operatorname{rank}(Q_{-k}) = d - 2$. In both cases, $\operatorname{rank}(Q_{-k}) < d$, thus $\operatorname{rank}(M) < d$. $\qquad\square$

### A.5 Proof of Theorem 3.4

*Proof.* Let $\tilde{A} \in \mathbb{R}^{d \times d}$ and $\tilde{G} \in \mathbb{R}^{d \times m}$, such that $X(\cdot; x_0, A, G) \stackrel{\mathrm{d}}{=} X(\cdot; x_0, \tilde{A}, \tilde{G})$, we denote as $X \stackrel{\mathrm{d}}{=} \tilde{X}$. For simplicity of notation, in the following, we denote $A_1 := A$, $A_2 := \tilde{A}$, $G_1 := G$ and $G_2 := \tilde{G}$, and denote $X \stackrel{\mathrm{d}}{=} \tilde{X}$ as $X^1 \stackrel{\mathrm{d}}{=} X^2$.

**Sufficiency**. We will show that under the identifiability condition (14), one has $(A_1, G_1 G_1^\top) = (A_2, G_2 G_2^\top)$.

We first show that $H_1 = H_2$ ($H_i := G_i G_i^T$). Indeed, since $X^1, X^2$ have the same distribution, one has

$$
\mathbb{E}[f(X_t^1)] = \mathbb{E}[f(X_t^2)] \tag{23}
$$

for all $0 \leqslant t < \infty$ and $f \in C^\infty(\mathbb{R}^d)$. By differentiating (23) at $t = 0$, one finds that

$$
(\mathcal{L}_1 f)(x_0) = (\mathcal{L}_2 f)(x_0), \tag{24}
$$

where $\mathcal{L}_i$ is the generator of $X^i$ ($i = 1, 2$). Based on the Proposition 2.1,

$$
(\mathcal{L}_i f)(x_0) = \sum_{k=1}^d \sum_{l=1}^d (A_i)_{kl} x_{0l} \frac{\partial f}{\partial x_k}(x_0) + \frac{1}{2} \sum_{k,l=1}^d (H_i)_{kl} \frac{\partial^2 f}{\partial x_k \partial x_l}(x_0),
$$

where $(M)_{kl}$ denotes the $kl$-entry of matrix $M$, and $x_{0l}$ is the $l$-th component of $x_0$. Since (24) is true for all $f$, by taking

$$
f(x) = (x_p - x_{0p})(x_q - x_{0q}),
$$

it is readily checked that

$$
(H_1)_{pq} = (H_2)_{pq},
$$

for all $p, q = 1, \ldots, d$. As a result, $H_1 = H_2$. Let us call this matrix $H$.

Next, we show that $A_1 = A_2$. We first show the relationship between $A_i$ and $x_0$, and then show the relationship between $A_i$ and $H$. To this end, one first recalls that

$$
X_t^i = e^{A_i t} x_0 + \int_0^t e^{A_i(t-s)} G_i dW_s.
$$

Set $m_i(t) := \mathbb{E}[X_t^i]$, we know that $m_i(t)$ satisfies the ODE

$$
\begin{aligned}
\dot{m}_i(t) &= A_i m_i(t), \quad \forall 0 \leqslant t < \infty, \\
m_i(0) &= x_0,
\end{aligned} \tag{25}
$$

where $\dot{f}(t)$ denotes the first derivative of function $f(t)$ with respect to time $t$.

Simple calculation shows that

$$
m_i(t) = e^{A_i t} x_0.
$$

Since $X^1 \overset{\mathrm{d}}{=} X^2$, one has

$$\mathbb{E}[X^1_t] = \mathbb{E}[X^2_t]$$

for all $0 \leqslant t < \infty$. That is

$$e^{A_1 t} x_0 = e^{A_2 t} x_0, \quad \forall 0 \leqslant t < \infty.$$

Taking $k$-th derivative of $e^{A_i t} x_0$ with respect to $t$, one finds that

$$\frac{d^k}{dt^k}\Big|_{t=0} e^{A_i t} x_0 = A_i^k x_0,$$

for all $k = 1, 2, \ldots$. Consequently,

$$A_1^k x_0 = A_2^k x_0.$$

Let us denote this vector $A^k x_0$. Obviously, one gets

$$A_1 A^{k-1} x_0 = A_2 A^{k-1} x_0 \quad \text{for all } k = 1, 2, \ldots. \tag{26}$$

In the following, we show the relationship between $A_i$ and $H$. Let us denote

$$Y^i_t := \int_0^t e^{A_i(t-s)} G_i dW_s = X^i_t - \mathbb{E}[X^i_t]$$

and

$$V_i(t, t+h) := \mathbb{E}[Y^i_{t+h} \cdot (Y^i_t)^T].$$

Simple calculation shows that

$$\begin{aligned} V_i(t, t+h) &= e^{A_i h} \int_0^t e^{A_i(t-s)} H e^{A_i^\top (t-s)} ds \\ &= e^{A_i h} V_i(t), \end{aligned} \tag{27}$$

where $V_i(t) := V_i(t, t)$.

Since $X^1 \overset{\mathrm{d}}{=} X^2$, by Lemma 3.2, one has

$$V_1(t, t+h) = V_2(t, t+h), \quad \forall 0 \leqslant t, h < \infty.$$

To obtain information about $A_i$, let us fix $t$ for now and take $k$-th derivative of (27) with respect to $h$. One finds that

$$\frac{d^k}{dh^k}\Big|_{h=0} V_i(t, t+h) = A_i^k V_i(t), \tag{28}$$

for all $k = 1, 2, \ldots$.

On the other hand, the function $V_i(t)$ satisfies the ODE [56]

$$\begin{aligned} \dot{V}_i(t) &= A_i V_i(t) + V_i(t) A_i^\top + H, \quad 0 \leqslant t < \infty, \\ V_i(0) &= 0. \end{aligned}$$

In particular,

$$\dot{V}_i(0) = A_i V_i(0) + V_i(0) A_i + H = H.$$

By differentiating (28) at $t = 0$, it follows that

$$\frac{d}{dt}\Big|_{t=0} \frac{d^k}{dh^k}\Big|_{h=0} V_i(t, t+h) = A_i^k H,$$

for all $k = 1, 2, \ldots$. Consequently,

$$A_1^k H = A_2^k H.$$

Let us denote this matrix $A^k H$. Obviously, by rearranging this matrix, one gets

$$A_1 A^{k-1} H = A_2 A^{k-1} H \quad \text{for all } k = 1, 2, \ldots. \tag{29}$$

Recall our identifiability condition is that $\mathrm{rank}(M) = d$ with

$$M := [x_0 | A x_0 | \ldots | A^{d-1} x_0 | H_{\cdot 1} | A H_{\cdot 1} | \ldots | A^{d-1} H_{\cdot 1} | \ldots | H_{\cdot d} | A H_{\cdot d} | \ldots | A^{d-1} H_{\cdot d}].$$

If we denote the $j$-th column in $M$ as $M_{\cdot j}$, one gets

$$A_1 M_{\cdot j} = A_2 M_{\cdot j} \,,$$

for all $j = 1, \ldots, d + d^2$ by Equations (26) and (29).

This means one can find a full-rank matrix $B \in \mathbb{R}^{d \times d}$ by horizontally stacking $d$ linearly independent columns of matrix $M$, such that $A_1 B = A_2 B$. Since $B$ is invertible, one thus concludes that $A_1 = A_2$. Hence, the sufficiency of the condition is proved.

**Necessity**. In the following, we will show that when $A$ has distinct eigenvalues. The condition (14) stated in Theorem 3.4 is also necessary. Specifically, we will show that when the identifiability condition (14) is not satisfied, one can always find a $\tilde{A}$ with $(A, GG^\top) \neq (\tilde{A}, GG^\top)$ such that $X \overset{\mathrm{d}}{=} \tilde{X}$. Recall that for simplicity of notation, we denote $A_1 := A$, $A_2 := \tilde{A}$, and denote $X \overset{\mathrm{d}}{=} \tilde{X}$ as $X^1 \overset{\mathrm{d}}{=} X^2$, where process $X^i = \{X_t^i : 0 \leqslant t < \infty\}$, and $X_t^i = X(t; x_0, A_i, G)$ following the form described in the solution process (3). In the following, we may use both $A$ and $A_1$ interchangeably according to the context.

By Lemma 3.2, to guarantee $X^1 \overset{\mathrm{d}}{=} X^2$ one only needs to show that

$$\mathbb{E}[X_t^1] = \mathbb{E}[X_t^2], \quad \forall 0 \leqslant t < \infty \,,$$
$$V_1(t, t+h) = V_2(t, t+h), \quad \forall 0 \leqslant t, h < \infty \,.$$

That is,

$$e^{A_1 t} x_0 = e^{A_2 t} x_0, \quad \forall 0 \leqslant t < \infty \,,$$
$$e^{A_1 h} V(t) = e^{A_2 h} V(t), \quad \forall 0 \leqslant t, h < \infty \,, \tag{30}$$
$$V_1(t) = V_2(t), \quad \forall 0 \leqslant t < \infty \,,$$

where $V(t) := V_1(t) = V_2(t)$.

Recall that $H = GG^\top$. For simplicity of notation, abusing notation a bit, we denote $H_{\cdot 0} := x_0$. Let

$$\tilde{H}_{\cdot j} := Q^{-1} H_{\cdot j}, \quad \text{for all } j = 0, \ldots, d \,,$$

and

$$w_{j,k} := \begin{cases} \tilde{H}_{\cdot j, k} \in \mathbb{R}^1, & \text{for } k = 1, \ldots, K_1 \,, \\ (\tilde{H}_{\cdot j, 2k - K_1 - 1}, \tilde{H}_{\cdot j, 2k - K_1})^\top \in \mathbb{R}^2, & \text{for } k = K_1 + 1, \ldots, K \,. \end{cases}$$

When the identifiability condition (14) is not satisfied, that is

$$\mathrm{rank}([H_{\cdot 0}|AH_{\cdot 0}|\ldots|A^{d-1}H_{\cdot 0}|H_{\cdot 1}|AH_{\cdot 1}|\ldots|A^{d-1}H_{\cdot 1}|\ldots|H_{\cdot d}|AH_{\cdot d}|\ldots|A^{d-1}H_{\cdot d}]) < d \,,$$

by Lemma 3.3, there exists $k$ such that $|w_{j,k}| = 0$, i.e., $w_{j,k} = 0$ ($\in \mathbb{R}^1$ or $\mathbb{R}^2$), for all $j = 0, \ldots, d$. Recall that

$$V(t) = V_1(t) = \int_0^t e^{A(t-s)} H e^{A^\top (t-s)} ds$$
$$= \int_0^t Q e^{\Lambda(t-s)} Q^{-1} Q [\tilde{H}_{\cdot 1}|\ldots|\tilde{H}_{\cdot d}] e^{A^\top (t-s)} ds$$
$$= Q \int_0^t e^{\Lambda(t-s)} [\tilde{H}_{\cdot 1}|\ldots|\tilde{H}_{\cdot d}] e^{A^\top (t-s)} ds \tag{31}$$
$$:= Q \int_0^t W e^{A^\top (t-s)} ds \,,$$

where

$$W = e^{\Lambda(t-s)} [\tilde{H}_{\cdot 1}|\ldots|\tilde{H}_{\cdot d}] \,,$$

and some calculation shows that

$$W = \begin{bmatrix} e^{J_1 (t-s)} w_{1,1} & \cdots & e^{J_1 (t-s)} w_{d,1} \\ \vdots & \ddots & \vdots \\ e^{J_K (t-s)} w_{1,K} & \cdots & e^{J_K (t-s)} w_{d,K} \end{bmatrix} \,,$$

recall that

$$J_k = \begin{cases} \lambda_k, & \text{if } k = 1, \ldots, K_1, \\ \begin{bmatrix} a_k & -b_k \\ b_k & a_k \end{bmatrix}, & \text{if } k = K_1 + 1, \ldots, K. \end{cases}$$

Since $w_{j,k} = 0 \ (\in \mathbb{R}^1 \text{ or } \mathbb{R}^2)$, for all $j = 0, 1, \ldots, d$, then if $k \in \{1, \ldots, K_1\}$, the $k$-th row of $W$

$$W_{k\cdot} = 0\,;$$

and if $k \in \{K_1 + 1, \ldots, K\}$, then the $(2k - K_1 - 1)$-th and the $(2k - K_1)$-th rows

$$W_{(2k-K_1-1)\cdot} = W_{(2k-K_1)\cdot} = 0\,,$$

where $W_{k\cdot}$ denotes the $k$-th row vector of matrix $W$.

If we denote

$$\tilde{V}(t)_{\cdot j} := Q^{-1} V(t)_{\cdot j}, \quad \text{for all } j = 1, \ldots, d\,,$$

and

$$w(t)_{j,k} := \begin{cases} \tilde{V}(t)_{\cdot j,k} \in \mathbb{R}^1, & \text{for } k = 1, \ldots, K_1, \\ (\tilde{V}(t)_{\cdot j,2k-K_1-1}, \tilde{V}(t)_{\cdot j,2k-K_1})^\top \in \mathbb{R}^2, & \text{for } k = K_1 + 1, \ldots, K. \end{cases}$$

Then by multiplying $Q^{-1}$ in both sides of Equation (31), one obtains that

$$w(t)_{j,k} = 0 \ (\in \mathbb{R}^1 \text{ or } \mathbb{R}^2)$$

for all $j = 1, \ldots, d$ and all $0 \leqslant t < \infty$. This indicates that when all the vectors $H_{\cdot j}$ for $j = 1, \ldots, d$ are confined to an $A$-invariant proper subspace of $\mathbb{R}^d$, denotes as $L$, then each column of the covariance matrix $V(t)$ in Equation (30) is also confined to $L$, for all $0 \leqslant t < \infty$. Therefore, under condition (14), $x_0$, $H_{\cdot j}$ (for all $j = 1, \ldots, d$) and each column of the covariance matrix $V(t)$ (for all $0 \leqslant t < \infty$) are confined to an $A$-invariant proper subspace of $\mathbb{R}^d$. Thus, a matrix $A_2$ exists, with $A_2 \neq A_1$ such that the first two equations in Equation (30) are satisfied.

In particular, by [43, Theorem 2.5], when $k \in \{1, \ldots, K_1\}$, there exists matrix $D \in \mathbb{R}^{d \times d}$, with the $kk$-th element $D_{kk} = c \neq 0$ and all the other elements of $D$ are zeros. Let

$$A_2 = A_1 + QDQ^{-1} \neq A_1\,,$$

then $A_1$ and $A_2$ satisfy the first two equations in Equation (30). Then we will show that such a $A_2$ also satisfy the third equation in Equation (30).

Some calculation shows that

$$\begin{aligned} V_1(t) &= \int_0^t e^{A_1(t-s)} H e^{A_1^\top(t-s)} ds \\ &= \int_0^t Q e^{\Lambda(t-s)} Q^{-1} H (Q^T)^{-1} e^{\Lambda(t-s)} Q^T ds \\ &:= \int_0^t Q e^{\Lambda(t-s)} P_1 e^{\Lambda(t-s)} Q^T ds\,, \end{aligned} \tag{32}$$

where $P_1 := Q^{-1} H (Q^T)^{-1}$. And

$$\begin{aligned} V_2(t) &= \int_0^t e^{A_2(t-s)} H e^{A_2^\top(t-s)} ds \\ &= \int_0^t e^{(A_1+QDQ^{-1})(t-s)} H e^{(A_1+QDQ^{-1})^\top(t-s)} ds \\ &= \int_0^t Q e^{\Lambda(t-s)} e^{D(t-s)} Q^{-1} H (Q^T)^{-1} e^{D(t-s)} e^{\Lambda(t-s)} Q^T ds \\ &:= \int_0^t Q e^{\Lambda(t-s)} P_2 e^{\Lambda(t-s)} Q^T ds\,, \end{aligned} \tag{33}$$

where $P_2 := e^{D(t-s)}Q^{-1}H(Q^T)^{-1}e^{D(t-s)}$. If one can show that $P_1 = P_2$, then it is readily checked that $V_1(t) = V_2(t)$ for all $0 \leqslant t < \infty$. Recall that

$$Q^{-1}H = \tilde{H},$$

where $\tilde{H} = [\tilde{H}_{\cdot 1}|\ldots|\tilde{H}_{\cdot d}]$. And when condition (14) is not satisfied, the $k$-th row of $\tilde{H}$:

$$\tilde{H}_{k\cdot} = 0.$$

Since

$$P_1 = Q^{-1}H(Q^T)^{-1} = \tilde{H}(Q^T)^{-1},$$

therefore, the $k$-th row of $P_1$:

$$(P_1)_{k\cdot} = 0.$$

Simple calculation shows that matrix $P_1$ is symmetric, thus, the $k$-th column of $P_1$:

$$(P_1)_{\cdot k} = 0.$$

It is easy to obtain that $e^{D(t-s)}$ is a diagonal matrix expressed as

$$e^{D(t-s)} = \begin{bmatrix} 1 & & & & \\ & \ddots & & & \\ & & e^{c(t-s)} & & \\ & & & \ddots & \\ & & & & 1 \end{bmatrix}$$

where $e^{c(t-s)}$ is the $kk$-th entry. Then, simple calculation shows that

$$P_2 = e^{D(t-s)}Q^{-1}H(Q^T)^{-1}e^{D(t-s)} = e^{D(t-s)}P_1 e^{D(t-s)} = P_1.$$

Therefore, one obtains that

$$V_1(t) = V_2(t), \quad \forall 0 \leqslant t < \infty.$$

Hence, when $k \in \{1, \ldots, K_1\}$, we find a $A_2$, with $A_2 \neq A_1$ such that Equation (30) is satisfied.

When $k \in \{K_1 + 1, \ldots, K\}$, there exists matrix $D' \in \mathbb{R}^{d \times d}$, with

$$\begin{bmatrix} D'_{2k-K_1-1,2k-K_1-1} & D'_{2k-K_1-1,2k-K_1} \\ D'_{2k-K_1,2k-K_1-1} & D'_{2k-K_1,2k-K_1} \end{bmatrix} := \begin{bmatrix} c_1 & c_2 \\ c_3 & c_4 \end{bmatrix},$$

where $M_{i,j}$ denotes the $ij$-th entry of matrix $M$, $c = [c_1, c_2, c_3, c_4]^\top \neq 0$, and all the other elements of $D'$ are zeros. Let

$$A_2 = A_1 + QD'Q^{-1} \neq A_1,$$

then $A_1$ and $A_2$ satisfy the first two equations in Equation (30). Similar to the case where $k \in \{1, \ldots, K_1\}$, one can also show that such a $A_2$ also satisfies the third equation in Equation (30).

Therefore, assuming $A$ has distinct eigenvalues, then when the identifiability condition (14) is not satisfied, one can always find a $A_2$ with $(A_1, GG^\top) \neq (A_2, GG^\top)$ such that Equation (30) is satisfied, i.e., $X^1 \overset{d}{=} X^2$. Hence, the necessity of the condition is proved. $\square$

### A.6 Proof of Corollary 3.4.1

*Proof.* There are two ways to prove this corollary, we will present both of them in the following.

**Way1.** By [62, Lemma 2.2],

$$\text{span}([G|AG|\ldots|A^{d-1}G]) = \text{span}([GG^\top|AGG^\top|\ldots|A^{d-1}GG^\top]),$$

where $\text{span}(M)$ denotes the linear span of the columns of the matrix $M$. therefore, when

$$\text{rank}([G|AG|\ldots|A^{d-1}G]) = d,$$

then

$$\text{span}([G|AG|\ldots|A^{d-1}G]) = \mathbb{R}^d,$$

thus,

$$\text{span}([GG^\top | AGG^\top | \ldots | A^{d-1}GG^\top]) = \mathbb{R}^d \,.$$

Therefore,

$$\text{rank}([GG^\top | AGG^\top | \ldots | A^{d-1}GG^\top]) = d \,,$$

since the rank of a matrix is the dimension of its span. Then by Theorem 3.4, the generator of the SDE (1) is identifiable from $x_0$.

**Way2.** Let $\tilde{A} \in \mathbb{R}^{d \times d}$ and $\tilde{G} \in \mathbb{R}^{d \times m}$, such that $X(\cdot; x_0, A, G) \overset{\text{d}}{=} X(\cdot; x_0, \tilde{A}, \tilde{G})$, we denote as $X \overset{\text{d}}{=} \tilde{X}$, we will show that under our identifiability condition $(A, GG^\top) = (\tilde{A}, \tilde{G}\tilde{G}^\top)$. By applying the same notations used in the proof of Theorem 3.4, in the following, we denote $A_1 := A$, $A_2 := \tilde{A}$, $G_1 := G$ and $G_2 := \tilde{G}$.

In the proof of Theorem 3.4, we have shown that $G_1 G_1^\top = G_2 G_2^\top$, thus, we only need to show that under the condition stated in this corollary, $A_1 = A_2$. According to the proof of Theorem 3.4, for all $0 \leqslant t < \infty$, we have

$$V_1(t) = V_2(t) \,,$$
$$A_1 V_1(t) = A_2 V_2(t) \,.$$

Let $V(t) := V_i(t) (i = 1, 2)$, one gets

$$A_1 V(t) = A_2 V(t) \,, \quad \forall 0 \leqslant t < \infty \,.$$

Therefore, if there exists a $0 \leqslant t < \infty$, such that $V(t)$ is nonsingular, then one can conclude that $A_1 = A_2$.

By [20, Theorem 3.2], the covariance $V(t)$ is nonsingular for all $t > 0$, if and only if

$$\text{rank}([G|AG| \ldots | A^{d-1}G]) = d \,,$$

that is the pair $[A, G]$ is controllable. Therefore, under the condition stated in this corollary, $A_1 = A_2$, thus the generator of the SDE (1) is identifiable from $x_0$. $\qquad\square$

## A.7    Proof of Theorem 3.5

*Proof.* Let $\tilde{A}, \tilde{G}_k \in \mathbb{R}^{d \times d}$ for all $k = 1, \ldots, m$, such that $X(\cdot; x_0, A, \{G_k\}_{k=1}^m) \overset{\text{d}}{=} X(\cdot; x_0, \tilde{A}, \{\tilde{G}_k\}_{k=1}^m)$, we denote as $X \overset{\text{d}}{=} \tilde{X}$, we will show that under our identifiability condition, for all $x \in \mathbb{R}^d$, $(A, \sum_{k=1}^m G_k xx^\top G_k^\top) = (\tilde{A}, \sum_{k=1}^m \tilde{G}_k xx^\top \tilde{G}_k^\top)$. For simplicity of notation, in the following, we denote $A_1 := A$, $A_2 := \tilde{A}$, $G_{1,k} := G_k$ and $G_{2,k} := \tilde{G}_k$, and denote $X \overset{\text{d}}{=} \tilde{X}$ as $X^1 \overset{\text{d}}{=} X^2$.

We first show that $A_1 = A_2$. Set $m_i(t) := \mathbb{E}[X_t^i]$, we know that $m_i(t)$ satisfies the ODE

$$\begin{aligned} \dot{m}_i(t) &= A_i m_i(t) \,, \quad \forall 0 \leqslant t < \infty \,, \\ m_i(0) &= x_0 \,, \end{aligned} \tag{34}$$

where $\dot{f}(t)$ denotes the first derivative of function $f(t)$ with respect to time $t$.

Simple calculation shows that

$$m_i(t) = e^{A_i t} x_0 \,.$$

Since $X^1 \overset{\text{d}}{=} X^2$, one has

$$\mathbb{E}[X_t^1] = \mathbb{E}[X_t^2]$$

for all $0 \leqslant t < \infty$. That is

$$e^{A_1 t} x_0 = e^{A_2 t} x_0 \,, \quad \forall 0 \leqslant t < \infty \,.$$

Taking $j$-th derivative of $e^{A_i t} x_0$ with respect to $t$, one finds that

$$\frac{d^j}{dt^j}\Big|_{t=0} e^{A_i t} x_0 = A_i^j x_0 \,,$$

for all $j = 1, 2, \ldots$. Consequently,

$$A_1^j x_0 = A_2^j x_0 \,.$$

Let us denote this vector $A^j x_0$. Obviously, one gets

$$A_1 A^{j-1} x_0 = A_2 A^{j-1} x_0 \quad \text{for all } j = 1, 2, \ldots . \tag{35}$$

By condition A1, it is readily checked that $A_1 = A_2$ from Equation (35).

In the following, we show that under condition A2, for all $x \in \mathbb{R}^d$,

$$\sum_{k=1}^m G_{1,k} x x^\top G_{1,k}^\top = \sum_{k=1}^m G_{2,k} x x^\top G_{2,k}^\top \,.$$

We know the function $P_i(t) := \mathbb{E}[X_t^i (X_t^i)^\top]$ satisfies the ODE

$$\dot{P}_i(t) = A_i P_i(t) + P_i(t) A_i^\top + \sum_{k=1}^m G_{i,k} P_i(t) G_{i,k}^\top \,, \quad \forall 0 \leqslant t < \infty \,,$$

$$P_i(0) = x_0 x_0^\top \,. \tag{36}$$

Since $X^1 \overset{\mathrm{d}}{=} X^2$,

$$P_1(t) = P_2(t) \,, \quad \forall 0 \leqslant t < \infty \,,$$

let us call it $P(t)$. By differentiating $P_i(t)$ one also gets that

$$\dot{P}_1(t) = \dot{P}_2(t) \,, \quad \forall 0 \leqslant t < \infty \,.$$

Since we have shown that $A_1 = A_2$ under condition A1, from Equation (36) one observes that

$$\sum_{k=1}^m G_{1,k} P(t) G_{1,k}^\top = \sum_{k=1}^m G_{2,k} P(t) G_{2,k}^\top \,, \quad \forall 0 \leqslant t < \infty \,. \tag{37}$$

By vectorizing $P(t)$, some calculation shows that the ODE (36) can be expressed as

$$\mathrm{vec}(\dot{P}(t)) = \mathcal{A} \mathrm{vec}(P(t)) \,,$$

$$\mathrm{vec}(P(0)) = \mathrm{vec}(x_0 x_0^\top) \,, \tag{38}$$

with an explicit solution

$$\mathrm{vec}(P(t)) = e^{\mathcal{A}t} \mathrm{vec}(x_0 x_0^\top) \,,$$

where $\mathcal{A} = A \oplus A + \sum_{k=1}^m G_k \otimes G_k \in \mathbb{R}^{d^2 \times d^2}$, and $\mathrm{vec}(M)$ denotes the vector by stacking the columns of matrix $M$ vertically.

By definition, $P(t) \in \mathbb{R}^{d \times d}$ is symmetric, thus $\mathrm{vec}(P(t))$ for all $0 \leqslant t < \infty$ is confined to a proper subspace of $\mathbb{R}^{d^2}$, let us denote this proper subspace $W$, simple calculation shows that

$$\dim(W) = (d^2 + d)/2 \,,$$

where $\dim(W)$ denotes the dimension of the subspace $W$, that is the number of vectors in any basis for $W$. In particular, one can find a basis of $W$ denoting as

$$\{\mathrm{vec}(E_{11}), \mathrm{vec}(E_{21}), \mathrm{vec}(E_{22}), \ldots, \mathrm{vec}(E_{dd})\} \,,$$

where $E_{ij}$ stands for a $d \times d$ matrix, with the $ij$-th and $ji$-th elements are 1 and all other elements are 0, for all $i, j = 1, \ldots, d$ and $i \geqslant j$.

Suppose there exists $t_i$'s, for $i = 1, \ldots, (d^2 + d)/2$, such that $\mathrm{vec}(P(t_1)), \ldots, \mathrm{vec}(P(t_{(d^2+d)/2}))$ are linearly independent, then for all $x \in \mathbb{R}^d$,

$$\mathrm{vec}(x x^\top) = l_1 \mathrm{vec}(P(t_1)) + \ldots + l_{(d^2+d)/2} \mathrm{vec}(P(t_{(d^2+d)/2})) \,,$$

that is

$$x x^\top = l_1 P(t_1) + \ldots + l_{(d^2+d)/2} P(t_{(d^2+d)/2}) \,,$$

where $l := \{l_1, \ldots, l_{(d^2+d)/2}\} \in \mathbb{R}^{(d^2+d)/2}$. According to Equation (37), it is readily checked that for all $x \in \mathbb{R}^d$,

$$\sum_{k=1}^{m} G_{1,k} x x^\top G_{1,k}^\top = \sum_{k=1}^{m} G_{2,k} x x^\top G_{2,k}^\top.$$

By [57, Lemma 6.1], there exists $(d^2 + d)/2$ $t_i$'s such that $\text{vec}(P(t_1)), \ldots, \text{vec}(P(t_{(d^2+d)/2}))$ are linearly independent, if and only if the orbit of $\text{vec}(P(t))$ (i.e., the trajectory of ODE (38) started from initial state $v$), denoting as $\gamma(\mathcal{A}, v)$ with $v = \text{vec}(x_0 x_0^\top)$, is not confined to a proper subspace of $W$.

Next, we show that under condition A2, orbit $\gamma(\mathcal{A}, v)$ is not confined to a proper subspace of $W$.

Assume orbit $\gamma(\mathcal{A}, v)$ is confined to a proper subspace of $W$. Then there exists $w \neq 0 \in W$ such that

$$w^\top e^{\mathcal{A}t} v = 0, \quad \forall 0 \leqslant t < \infty.$$

By taking $j$-th derivative with respect to $t$, we have

$$w^T \mathcal{A}^j e^{\mathcal{A}t} v = 0, \quad \forall 0 \leqslant t < \infty, j = 0, \ldots, (d^2 + d - 2)/2.$$

In particular, for $t = 0$,

$$w^T \mathcal{A}^j v = 0, \quad \text{for } j = 0, \ldots, (d^2 + d - 2)/2.$$

Therefore,

$$w^T [v|\mathcal{A}v| \ldots |\mathcal{A}^{(d^2+d-2)/2} v] = 0. \tag{39}$$

Since $w \in W$, $w \in \text{span}\{\text{vec}(E_{11}), \text{vec}(E_{21}), \ldots, \text{vec}(E_{dd})\}$, set $\text{vec}(\overline{w}) := w$, then $\overline{w}$ is a $d \times d$ symmetric matrix. Since $P(t)$ is symmetric for all $0 \leqslant t < \infty$, according to Equation (36), simple calculation shows that the $j$-th derivative of $P(t)$ is also symmetric for all $0 \leqslant t < \infty$, for $j = 0, 1, \ldots$. Recall that

$$\text{vec}(P(t)) = e^{\mathcal{A}t} v,$$
$$\text{vec}(P^{(j)}(t)) = \mathcal{A}^j e^{\mathcal{A}t} v,$$

where $P^{(j)}(t)$ denotes the $j$-th derivative of $P(t)$ with respect to $t$. In particular, when $t = 0$, one has

$$\text{vec}(P(0)) = v,$$
$$\text{vec}(P^{(j)}(0)) = \mathcal{A}^j v,$$

then if we denote matrix $\overline{\mathcal{A}^j v}$ by setting $\text{vec}(\overline{\mathcal{A}^j v}) := \mathcal{A}^j v$, matrices $\overline{\mathcal{A}^j v}$ are symmetric for all $j = 0, 1, \ldots$.

Therefore, we can say that there are only $(d^2 + d)/2$ distinct elements in each of the vectors: $w, v, \mathcal{A}v, \ldots, \mathcal{A}^{(d^2+d-2)/2} v$ in Equation (39). Moreover, since these vectors all correspond to $d \times d$ symmetric matrices, the repetitive elements in each vector appear in the same positions in each vector. Hence, we can focus on checking those distinct elements in each vector, that is Equation (39) can be expressed as

$$\underline{w}^T [\underline{v}|\underline{\mathcal{A}v}| \ldots |\underline{\mathcal{A}^{(d^2+d-2)/2} v}] = 0, \tag{40}$$

where $\underline{w} \in \mathbb{R}^{(d^2+d)/2}$ denotes as

$$\underline{w} := [\overline{w}_{11}, \sqrt{2}\overline{w}_{21}, \ldots, \sqrt{2}\overline{w}_{d1}, \overline{w}_{22}, \ldots, \sqrt{2}\overline{w}_{d2}, \ldots, \overline{w}_{dd}]^\top,$$

where $\overline{w}_{ij}$ denotes the $ij$-th element of $\overline{w}$, with $i, j = 1, \ldots, d$ and $i \geqslant j$. When $i \neq j$, the element is multiplied by a $\sqrt{2}$, that is, $\underline{w}$ only keeps the distinctive elements in $w$, and for each of the repetitive element, we multiply $\sqrt{2}$. We define $\underline{v}, \underline{\mathcal{A}v}, \ldots$ using the same way.

Under condition A2, matrix

$$[\underline{v}|\underline{\mathcal{A}v}| \ldots |\underline{\mathcal{A}^{(d^2+d-2)/2} v}] \in \mathbb{R}^{\frac{(d^2+d)}{2} \times \frac{(d^2+d)}{2}}$$

is easily to be checked to be invertible, then $\underline{w} = 0$, thus $w = 0$. This contradicts to $w \neq 0$, therefore, under condition A2, orbit $\gamma(\mathcal{A}, v)$ is not confined to a proper subspace of $W$. Hence, the theorem is proved. $\quad\square$

# B  Genericity of the derived identifiability conditions

## B.1  The identifiability condition stated in Theorem 3.5 is generic

We will show that the identifiability condition stated in Theorem 3.5 is generic. Specifically, we will show that for the set of $(x_0, A, \{G_k\}_{k=1}^m) \in \mathbb{R}^{d+(m+1)d^2}$ such that either condition A1 or A2 stated in Theorem 3.5 is violated, has Lebesgue measure zero in $\mathbb{R}^{d+(m+1)d^2}$. In the following, we first present a lemma we will use to prove our main proposition.

**Lemma B.1.** *Let $p : \mathbb{R}^n \to \mathbb{R}$ be a non-zero polynomial function. Let $Z := \{x \in \mathbb{R}^n : p(x) = 0\}$. Then $Z$ has Lebesgue measure zero in $\mathbb{R}^n$.*

*Proof.* When $n = 1$, suppose the degree of $x$ is $k \geqslant 1$, then by the fundamental theorem of algebra, there are at most $k$ $x$'s such that $x \in Z$. Therefore, Z has Lebesgue measure zero, since a finite set has measure zero in $\mathbb{R}$.

Suppose the lemma is established for polynomials in $n - 1$ variables. Let $p$ be a non-zero polynomial in $n$ variables, say of degree $k \geqslant 1$ in $x_n$, then we can write

$$p(\boldsymbol{x}, x_n) = \sum_{j=0}^{k} p_j(\boldsymbol{x}) x_n^j,$$

where $\boldsymbol{x} = \{x_1, \ldots, x_{n-1}\}$ and $p_0, \ldots, p_k$ are polynomials in the $n - 1$ variables $\{x_1, \ldots, x_{n-1}\}$, and there exists $j \in \{0, \ldots, k\}$ such that $p_j$ is a non-zero polynomial since $p$ is a non-zero polynomial. Then we can denote $Z$ as

$$Z = \{(\boldsymbol{x}, x_n) : p(\boldsymbol{x}, x_n) = 0\}.$$

Suppose $(\boldsymbol{x}, x_n) \in Z$, then there are two possibilities:

case 1  $p_0(\boldsymbol{x}) = \ldots = p_k(\boldsymbol{x}) = 0$.

case 2  there exists $i \in \{0, \ldots, k\}$ such that $p_i(\boldsymbol{x}) \neq 0$.

Let

$$A := \{(\boldsymbol{x}, x_n) \in Z : \text{ case 1 is satisfied}\},$$
$$B := \{(\boldsymbol{x}, x_n) \in Z : \text{ case 2 is satisfied}\},$$

then $Z = A \cup B$.

For case 1, recall that there exists $j \in \{0, \ldots, k\}$ such that $p_j$ is a non-zero polynomial, let

$$A_j := \{\boldsymbol{x} \in \mathbb{R}^{n-1} : p_j(\boldsymbol{x}) = 0\},$$

then by the induction hypothesis, $A_j$ has Lebesgue measure zero in $\mathbb{R}^{n-1}$. Therefore, $A_j \times \mathbb{R}$ has Lebesgue measure zero in $\mathbb{R}^n$. Since $A \subseteq A_j \times \mathbb{R}$, $A$ has Lebesgue measure zero in $\mathbb{R}^n$.

For case 2, let $\lambda^n$ be Lebesgue measure on $\mathbb{R}^n$, then

$$
\begin{aligned}
\lambda^n(B) &= \int_{\mathbb{R}^n} \mathbb{1}_B(\boldsymbol{x}, x_n) d\lambda^n \\
&= \int_{\mathbb{R}^n} \mathbb{1}_B(\boldsymbol{x}, x_n) d\boldsymbol{x} dx_n \\
&= \int_{\mathbb{R}^{n-1}} \left( \int_{\mathbb{R}} \mathbb{1}_B(\boldsymbol{x}, x_n) dx_n \right) d\boldsymbol{x},
\end{aligned}
\tag{41}
$$

where

$$\mathbb{1}_B(\boldsymbol{x}, x_n) = \begin{cases} 1, & \text{if } (\boldsymbol{x}, x_n) \in B, \\ 0, & \text{if } (\boldsymbol{x}, x_n) \notin B. \end{cases}$$

The inner integral in Equation (41) is equal to zero, since for a fixed $\boldsymbol{x}$, there are finitely many (indeed, at most $k$) $x_n$'s such that $p(\boldsymbol{x}, x_n) = 0$ under the condition of case 2. Thus, $\lambda^n(B) = 0$, that is, $B$ has Lebesgue measure zero in $\mathbb{R}^n$. Then it is readily checked that $Z$ has Lebesgue measure zero.  $\square$

Now we are ready to present the main proposition.

**Proposition B.1.** *Let*

$$S := \{(x_0, A, \{G_k\}_{k=1}^m) \in \mathbb{R}^{d+(m+1)d^2} : \text{either condition A1 or A2 in Theorem 3.5 is violated}\},$$

*then $S$ has Lebesgue measure zero in $\mathbb{R}^{d+(m+1)d^2}$.*

*Proof.* Let

$$S_A := \{(x_0, A) \in \mathbb{R}^{d+d^2} : \text{condition A1 is violated}\},$$

we first show that $S_A$ has Lebesgue measure zero in $\mathbb{R}^{d+d^2}$. Suppose $(x_0, A) \in S_A$, then $(x_0, A)$ satisfies

$$\text{rank}([x_0|Ax_0|\dots|A^{d-1}x_0]) < d,$$

that is the set of vectors $\{x_0, Ax_0, \dots, A^{d-1}x_0\}$ are linearly dependent, this means that

$$\det([x_0|Ax_0|\dots|A^{d-1}x_0]) = 0. \tag{42}$$

It is a simple matter of algebra that the left side of (42) can be expressed as some universal polynomial of the entries of $x_0$ and entries of $A$, denotes $p(x_0, A) = p(x_{01}, \dots, x_{0d}, a_{11}, a_{12}, \dots, a_{dd})$, where $x_{0j}$ denotes the $j$-th entry of $x_0$ and $a_{ij}$ denotes the $ij$-th entry of $A$. Therefore, one concludes that

$$p(x_0, A) = p(x_{01}, \dots, x_{0d}, a_{11}, a_{12}, \dots, a_{dd}) = 0.$$

Thus, $S_A$ can be expressed as

$$S_A = \{(x_0, A) \in \mathbb{R}^{d+d^2} : p(x_0, A) = 0\}.$$

Some calculation shows that

$$p(x_0, A) = \sum_{i_1, \dots, i_d = 1}^{d} x_{0i_1} \dots x_{0i_d} \det([(A^0)_{\cdot i_1}|(A^1)_{\cdot i_2}|\dots|(A^{d-1})_{\cdot i_d}]), \tag{43}$$

where $(M)_{\cdot j}$ denotes the $j$-th column vector of matrix $M$. Obviously, $p(x_0, A)$ is a non-zero polynomial function of entries of $x_0$ and entries of $A$, therefore, by Lemma B.1, $S_A$ has Lebesgue measure zero in $\mathbb{R}^{d+d^2}$. Let

$$S_1 := \{(x_0, A, \{G_k\}_{k=1}^m) \in \mathbb{R}^{d+(m+1)d^2} : \text{condition A1 is violated}\},$$

then it is readily checked that $S_1$ has Lebesgue measure zero in $\mathbb{R}^{d+(m+1)d^2}$.

Let

$$S_2 := \{(x_0, A, \{G_k\}_{k=1}^m) \in \mathbb{R}^{d+(m+1)d^2} : \text{condition A2 is violated}\},$$

we then show that $S_2$ has Lebesgue measure zero in $\mathbb{R}^{d+(m+1)d^2}$. Suppose $(x_0, A, \{G_k\}_{k=1}^m) \in S_2$, then $(x_0, A, \{G_k\}_{k=1}^m)$ satisfies

$$\text{rank}([v|\mathcal{A}v|\dots|\mathcal{A}^{(d^2+d-2)/2}v]) < (d^2 + d)/2,$$

recall that $\mathcal{A} = A \oplus A + \sum_{k=1}^m G_k \otimes G_k \in \mathbb{R}^{d^2 \times d^2}$ and $v = \text{vec}(x_0 x_0^\top) \in \mathbb{R}^{d^2}$. According to the proof A.7 of Theorem 3.5, we obtain that the set of vectors $\{v, \mathcal{A}v, \dots, \mathcal{A}^{(d^2+d-2)/2}\}$ are linearly dependent. Because all of these vectors are transferred from vectorizing $d \times d$ symmetric matrices, thus each of these vectors has only $(d^2 + d)/2$ distinct elements and the repetitive elements appear in the same positions in all vectors. Hence, abuse notation a little bit, we can focus on checking those distinct elements in each vector, that is

$$\{\underline{v}|\underline{\mathcal{A}v}|\dots|\underline{\mathcal{A}^{(d^2+d-2)/2}v}\}, \tag{44}$$

where $\underline{v} \in \mathbb{R}^{(d^2+d)/2}$ denotes the vector of deleting the repetitive elements of $v$. Since the set of vectors $\{v, \mathcal{A}v, \dots, \mathcal{A}^{(d^2+d-2)/2}\}$ are linearly dependent, the set of vectors $\{\underline{v}|\underline{\mathcal{A}v}|\dots|\underline{\mathcal{A}^{(d^2+d-2)/2}v}\}$ are linearly dependent, that is

$$\det([\underline{v}|\underline{\mathcal{A}v}|\dots|\underline{\mathcal{A}^{(d^2+d-2)/2}v}]) = 0. \tag{45}$$

Each entry of $v$ can be written as a non-zero polynomial function of entries of $x_0$ since $v = \text{vec}(x_0 x_0^\top)$. Each entry of $\mathcal{A}$ can be written as a non-zero polynomial function of entries of $A$ and $G_k$ with $k = 1, \ldots, m$, since $\mathcal{A} = A \oplus A + \sum_{k=1}^m G_k \otimes G_k \in \mathbb{R}^{d^2 \times d^2}$. Hence, the left side of Equation (45) can be expressed as some universal polynomial of the entries of $x_0$, $A$ and $G_k$'s, denotes

$$p(x_0, A, \{G_k\}_{k=1}^m) = p(x_{01}, \ldots, x_{0d}, a_{11}, \ldots, a_{dd}, G_{1,11}, \ldots, G_{1,dd}, \ldots, G_{m,11}, \ldots, G_{m,dd}),$$

where $G_{k,ij}$ denotes the $ij$-th entry of matrix $G_k$. Therefore, one concludes that

$$p(x_0, A, \{G_k\}_{k=1}^m) = 0.$$

Thus, $S_2$ can be expressed as

$$S_2 := \{(x_0, A, \{G_k\}_{k=1}^m) \in \mathbb{R}^{d+(m+1)d^2} : p(x_0, A, \{G_k\}_{k=1}^m) = 0\}.$$

Similar to the calculation of $p(x_0, A)$ in Equation (43), $p(x_0, A, \{G_k\}_{k=1}^m)$ can be expressed as a non-zero polynomial function of entries of $v$ and $\mathcal{A}$, thus it can also be expressed as a non-zero polynomial function of entries of $x_0$, $A$ and $G_k$'s. Therefore, by Lemma B.1, $S_2$ has Lebesgue measure zero in $\mathbb{R}^{d+(m+1)d^2}$.

We know that $S \subseteq S_1 \cup S_2$, let $\lambda$ be Lebesgue measure on $\mathbb{R}^{d+(m+1)d^2}$, then one has

$$\lambda(S) \leqslant \lambda(S_1) + \lambda(S_2) = 0.$$

Thus $S$ has Lebesgue measure zero in $\mathbb{R}^{d+(m+1)d^2}$. $\qquad\square$

## B.2 The identifiability condition stated in Theorem 3.4 is generic

We will show that the identifiability condition stated in Theorem 3.4 is generic.

**Proposition B.2.** *Let*

$$S := \{(x_0, A, G) \in \mathbb{R}^{d+d^2+dm} : condition \text{ (14) } in \text{ Theorem 3.4 is violated}\},$$

*then $S$ has Lebesgue measure zero in $\mathbb{R}^{d+d^2+dm}$.*

*Proof.* Suppose $(x_0, A, G) \in S$, then $(x_0, A, G)$ satisfies

$$\text{rank}([x_0|Ax_0|\ldots|A^{d-1}x_0|H_{\cdot 1}|AH_{\cdot 1}|\ldots|A^{d-1}H_{\cdot 1}|\ldots|H_{\cdot d}|AH_{\cdot d}|\ldots|A^{d-1}H_{\cdot d}]) < d,$$

recall that $H := GG^T$, and $H_{\cdot j}$ stands for the $j$-th column vector of matrix $H$, for all $j = 1, \cdots, d$. Let

$$S' := \{(x_0, A, G) \in \mathbb{R}^{d+d^2+dm} : \text{rank}([x_0|Ax_0|\ldots|A^{d-1}x_0]) < d\},$$

one observes that $S \subseteq S'$. According to the proof of Proposition B.1, it is readily checked that $S'$ has Lebesgue measure zero in $\mathbb{R}^{d+d^2+dm}$. Thus, $S$ has Lebesgue measure zero in $\mathbb{R}^{d+d^2+dm}$. $\qquad\square$

# C    Simulation settings

We present the true underlying system parameters along with the initial states of the SDEs employed in the simulation experiments. We randomly generate the true system parameters that satisfy or violate the corresponding identifiability conditions.

For the SDE (1):

1. identifiable case: satisfy condition (14) stated in Theorem 3.4:

$$x_0^{\text{id}} = \begin{bmatrix} 1.87 \\ -0.98 \end{bmatrix}, \quad A^{\text{id}} = \begin{bmatrix} 1.76 & -0.1 \\ 0.98 & 0 \end{bmatrix}, \quad G^{\text{id}} = \begin{bmatrix} -0.11 & -0.14 \\ -0.29 & -0.22 \end{bmatrix};$$

2. unidentifiable case: violate condition (14):

$$x_0^{\text{un}} = \begin{bmatrix} 1 \\ -1 \end{bmatrix}, \quad A^{\text{un}} = \begin{bmatrix} 1 & 2 \\ 1 & 0 \end{bmatrix}, \quad G^{\text{un}} = \begin{bmatrix} 0.11 & 0.22 \\ -0.11 & -0.22 \end{bmatrix}.$$

For the SDE (2):

1. identifiable case: satisfy both A1 and A2 stated in Theorem 3.5:

$$x_0^{\text{id}} = \begin{bmatrix} 1.87 \\ -0.98 \end{bmatrix}, \quad A^{\text{id}} = \begin{bmatrix} 1.76 & -0.1 \\ 0.98 & 0 \end{bmatrix}, \quad G_1^{\text{id}} = \begin{bmatrix} -0.11 & -0.14 \\ -0.29 & -0.22 \end{bmatrix}, \quad G_2^{\text{id}} = \begin{bmatrix} -0.17 & 0.59 \\ 0.81 & 0.18 \end{bmatrix};$$

2. unidentifiable case1: violate A1 satisfy A2:

$$x_0^{\text{un-A1}} = \begin{bmatrix} 1 \\ 1 \end{bmatrix}, \quad A^{\text{un-A1}} = \begin{bmatrix} 2 & 1 \\ 3 & 0 \end{bmatrix}, \quad G_1^{\text{un-A1}} = \begin{bmatrix} -0.11 & -0.14 \\ -0.29 & -0.22 \end{bmatrix}, \quad G_2^{\text{un-A1}} = \begin{bmatrix} -0.17 & 0.59 \\ 0.81 & 0.18 \end{bmatrix};$$

3. unidentifiable case2: satisfy A1 violate A2:

$$x_0^{\text{un-A2}} = \begin{bmatrix} 1 \\ -1 \end{bmatrix}, \quad A^{\text{un-A2}} = \begin{bmatrix} 1 & -2 \\ -1 & 0 \end{bmatrix}, \quad G_1^{\text{un-A2}} = \begin{bmatrix} -0.3 & 0.4 \\ -0.7 & 0.2 \end{bmatrix}, \quad G_2^{\text{un-A2}} = \begin{bmatrix} 0.8 & 0.2 \\ -0.2 & -0.4 \end{bmatrix}.$$

We have discussed in Section 4 that we use MLE method to estimate the system parameters from discrete observations sampled from the corresponding SDEs. Specifically, the negative log-likelihood function was minimized using the 'scipy.optimize.minimize' library in Python.

For all of our experiments, we initialized the parameter values as the true parameters plus 2. In the case of the SDE (1), we utilized the 'trust-constr' method with the hyper-parameter 'gtol'= 1e-3 and 'xtol'= 1e-3. On the other hand, for the SDE (2), we applied the 'BFGS' method and set the hyper-parameter 'gtol'= 1e-2. The selection of the optimization method and the corresponding hyper-parameters was determined through a series of experiments aimed at identifying the most suitable configuration.

# D Examples for reliable causal inference for linear SDEs

## D.1 Example for reliable causal inference for the SDE (1)

This example is inspired by [17, Example 5.4]. Recall that the SDE (1) is defined as

$$dX_t = AX_t dt + G dW_t, \quad X_0 = x_0,$$

where $0 \leqslant t < \infty$, $A \in \mathbb{R}^{d \times d}$ and $G \in \mathbb{R}^{d \times m}$ are constant matrices, $W$ is an $m$-dimensional standard Brownian motion. Let $X(t; x_0, A, G)$ denote the solution to the SDE (1). Let $\tilde{A} \in \mathbb{R}^{d \times d}$ and $\tilde{G} \in \mathbb{R}^{d \times m}$ define the following SDE:

$$d\tilde{X}_t = \tilde{A}\tilde{X}_t dt + \tilde{G} dW_t, \quad \tilde{X}_0 = x_0,$$

such that

$$X(\cdot; x_0, A, G) \stackrel{\mathrm{d}}{=} \tilde{X}(\cdot; x_0, \tilde{A}, \tilde{G}).$$

Then under our proposed identifiability condition stated in Theorem 3.4, we have shown that the generator of the SDE (1) is identifiable, i.e., $(A, GG^\top) = (\tilde{A}, \tilde{G}\tilde{G}^\top)$. Till now, we have shown that under our proposed identifiability conditions, the observational distribution $\xrightarrow{\text{identity}}$ the generator of the observational SDE. Then we will show that the post-intervention distribution is also identifiable. For notational simplicity, we consider intervention on the first coordinate, making the intervention $X_t^1 = \xi$ and $\tilde{X}_t^1 = \xi$ for $0 \leqslant t < \infty$. It will suffice to show equality of the distributions of the non-intervened coordinates (i.e., $X_t^{(-1)}$ and $\tilde{X}_t^{(-1)}$, note the superscripts do not denote reciprocals, but denote the $(d-1)$-coordinates without the first coordinate). Express the matrices of $A$ and $G$ in blocks

$$A = \begin{bmatrix} A_{11} & A_{12} \\ A_{21} & A_{22} \end{bmatrix}, \quad G = \begin{bmatrix} G_1 \\ G_2 \end{bmatrix},$$

where $A_{11} \in \mathbb{R}^{1 \times 1}$, $A_{22} \in \mathbb{R}^{(d-1) \times (d-1)}$, $G_1 \in \mathbb{R}^{1 \times m}$ and $G_2 \in \mathbb{R}^{(d-1) \times m}$. Also, consider corresponding expressions of matrices $\tilde{A}$ and $\tilde{G}$. By making intervention $X_t^1 = \xi$, one obtains the post-intervention process of the first SDE satisfies:

$$dX_t^{(-1)} = (A_{21}\xi + A_{22}X_t^{(-1)})dt + G_2 dW_t, \quad X_0^{(-1)} = x_0^{(-1)},$$

which is a multivariate Ornstein-Uhlenbeck process, according to [59, Corollary 1], this process is a Gaussian process, assuming $A_{22}$ is invertible, then the mean vector can be described as

$$E[X_t^{(-1)}] = e^{A_{22}t}x_0^{(-1)} - (I - e^{A_{22}t})A_{22}^{-1}A_{21}\xi,$$

and based on [59, Theorem 2], the cross-covariance can be described as

$$V(X_{t+h}^{(-1)}, X_t^{(-1)}) := \mathbb{E}\{(X_{t+h}^{(-1)} - \mathbb{E}[X_{t+h}^{(-1)}])(X_t^{(-1)} - \mathbb{E}[X_t^{(-1)}])^\top\}$$
$$= \int_0^t e^{A_{22}(t+h-s)}G_2 G_2^\top e^{A_{22}^\top(t-s)}ds.$$

Similarly, one can obtain that the mean vector and cross-covariance of the distribution of the post-intervention process of the second SDE by making intervention $\tilde{X}_t^1 = \xi$ satisfy:

$$E[\tilde{X}_t^{(-1)}] = e^{\tilde{A}_{22}t}x_0^{(-1)} - (I - e^{\tilde{A}_{22}t})\tilde{A}_{22}^{-1}\tilde{A}_{21}\xi,$$

and

$$V(\tilde{X}_{t+h}^{(-1)}, \tilde{X}_t^{(-1)}) := \mathbb{E}\{(\tilde{X}_{t+h}^{(-1)} - \mathbb{E}[\tilde{X}_{t+h}^{(-1)}])(\tilde{X}_t^{(-1)} - \mathbb{E}[\tilde{X}_t^{(-1)}])^\top\}$$
$$= \int_0^t e^{\tilde{A}_{22}(t+h-s)}\tilde{G}_2 \tilde{G}_2^\top e^{\tilde{A}_{22}^\top(t-s)}ds \qquad .$$

Then we will show that $E[X_t^{(-1)}] = E[\tilde{X}_t^{(-1)}]$, and $V(X_{t+h}^{(-1)}, X_t^{(-1)}) = V(\tilde{X}_{t+h}^{(-1)}, \tilde{X}_t^{(-1)})$ for all $0 \leqslant t, h < \infty$. Recall that we have shown $(A, GG^\top) = (\tilde{A}, \tilde{G}\tilde{G}^\top)$, thus, $A_{22} = \tilde{A}_{22}$ and $A_{21} = \tilde{A}_{21}$, then it is readily checked that $E[X_t^{(-1)}] = E[\tilde{X}_t^{(-1)}]$ for all $0 \leqslant t < \infty$.

Since

$$GG^\top = \begin{bmatrix} G_1 G_1^\top & G_1 G_2^\top \\ G_2 G_1^\top & G_2 G_2^\top \end{bmatrix} = \tilde{G}\tilde{G}^\top,$$

thus, $G_2 G_2^\top = \tilde{G}_2 \tilde{G}_2^\top$, then it is readily checked that $V(X_{t+h}^{(-1)}, X_t^{(-1)}) = V(\tilde{X}_{t+h}^{(-1)}, \tilde{X}_t^{(-1)})$ for all $0 \leqslant t, h < \infty$. Since both of these two post-intervention processes are Gaussian processes, according to Lemma 3.2, the distributions of these two post-intervention processes are the same. That is, the post-intervention distribution is identifiable.

### D.2    Example for reliable causal inference for the SDE (2)

Recall that the SDE (2) is defined as

$$dX_t = AX_t dt + \sum_{k=1}^m G_k X_t dW_{k,t}, \quad X_0 = x_0,$$

where $0 \leqslant t < \infty$, $A, G_k \in \mathbb{R}^{d \times d}$ for $k = 1, \ldots, m$ are some constant matrices, $W := \{W_t = [W_{1,t}, \ldots, W_{m,t}]^\top : 0 \leqslant t < \infty\}$ is an $m$-dimensional standard Brownian motion. Let $X(t; x_0, A, \{G_k\}_{k=1}^m)$ denote the solution to the SDE (2). Let $\tilde{A}, \tilde{G}_k \in \mathbb{R}^{d \times d}$ for $k = 1, \ldots, m$ define the following SDE:

$$d\tilde{X}_t = \tilde{A}\tilde{X}_t dt + \sum_{k=1}^m \tilde{G}_k \tilde{X}_t dW_{k,t}, \quad \tilde{X}_0 = x_0,$$

such that

$$X(\cdot; x_0, A, \{G_k\}_{k=1}^m) \stackrel{\mathrm{d}}{=} \tilde{X}(\cdot; x_0, \tilde{A}, \{\tilde{G}_k\}_{k=1}^m).$$

Then under our proposed identifiability condition stated in Theorem 3.5, we have shown that the generator of the SDE (2) is identifiable, i.e., $(A, \sum_{k=1}^m G_k xx^\top G_k^\top) = (\tilde{A}, \sum_{k=1}^m \tilde{G}_k xx^\top \tilde{G}_k^\top)$ for all $x \in \mathbb{R}^d$. Till now, we have shown that under our proposed identifiability conditions, the observational distribution $\xrightarrow{\text{identity}}$ the generator of the observational SDE. Then we aim to show that the post-intervention distribution is also identifiable. For notational simplicity, we consider intervention on the first coordinate, making the intervention $X_t^1 = \xi$ and $\tilde{X}_t^1 = \xi$ for $0 \leqslant t < \infty$. It will suffice to show equality of the distributions of the non-intervened coordinates (i.e., $X_t^{(-1)}$ and $\tilde{X}_t^{(-1)}$). Express the matrices of $A$ and $G_k$ for $k = 1, \ldots, m$ in blocks

$$A = \begin{bmatrix} A_{11} & A_{12} \\ A_{21} & A_{22} \end{bmatrix}, \quad G_k = \begin{bmatrix} G_{k,11} & G_{k,12} \\ G_{k,21} & G_{k,22} \end{bmatrix},$$

where $A_{11}, G_{k,11} \in \mathbb{R}^{1 \times 1}$, $A_{22}, G_{k,22} \in \mathbb{R}^{(d-1) \times (d-1)}$. Also consider corresponding expressions of matrices $\tilde{A}$ and $\tilde{G}_k$ for $k = 1, \ldots, m$. By making intervention $X_t^1 = \xi$, one obtains the post-intervention process of the first SDE satisfies:

$$dX_t^{(-1)} = (A_{21}\xi + A_{22}X_t^{(-1)})dt + \sum_{k=1}^m (G_{k,21}\xi + G_{k,22}X_t^{(-1)})dW_{k,t}, \quad X_0^{(-1)} = x_0^{(-1)}.$$

Since this post-intervention process is not a Gaussian process, one cannot explicitly show that the post-intervention distribution is identifiable. Instead, we check the surrogate of the post-intervention distribution, that is the first- and second-order moments of the post-intervention process $X_t^{(-1)}$. Which denote as $m(t)^{(-1)} = \mathbb{E}[X_t^{(-1)}]$ and $P(t)^{(-1)} = \mathbb{E}[X_t^{(-1)}(X_t^{(-1)})^\top]$ respectively. Then $m(t)^{(-1)}$ and $P(t)^{(-1)}$ satisfy the following ODE systems:

$$\frac{dm(t)^{(-1)}}{dt} = A_{21}\xi + A_{22}m(t)^{(-1)}, \quad m(0)^{-1} = x_0^{(-1)},$$

and

$$\begin{aligned}
\frac{dP(t)^{(-1)}}{dt} &= m(t)^{(-1)}\xi^\top A_{21}^\top + A_{21}\xi(m(t)^{(-1)})^\top + P(t)^{(-1)}A_{22}^\top + A_{22}P(t)^{(-1)} \\
&\quad + \sum_{k=1}^m (G_{k,21}\xi\xi^\top G_{k,21}^\top + G_{k,22}m(t)^{(-1)}\xi^\top G_{k,21}^\top + G_{k,21}\xi(m(t)^{(-1)})^\top G_{k,22}^\top \\
&\quad + G_{k,22}P(t)^{(-1)}G_{k,22}^\top), \quad P(0)^{(-1)} = x_0^{(-1)}(x_0^{(-1)})^\top.
\end{aligned} \tag{46}$$

Similarly, one can obtain the ODE systems describing the $\tilde{m}(t)^{(-1)}$ and $\tilde{P}(t)^{(-1)}$. Then we will show that $m(t)^{(-1)} = \tilde{m}(t)^{(-1)}$ and $P(t)^{(-1)} = \tilde{P}(t)^{(-1)}$ for all $0 \leqslant t < \infty$. Recall that we have

shown $A = \tilde{A}$, thus $A_{21} = \tilde{A}_{21}$ and $A_{22} = \tilde{A}_{22}$, then it is readily checked that $m(t)^{(-1)} = \tilde{m}(t)^{(-1)}$ for all $0 \leqslant t < \infty$.

In the proof of Theorem 3.5, we have shown that

$$\sum_{k=1}^{m} G_k P(t) G_k^\top = \sum_{k=1}^{m} \tilde{G}_k P(t) \tilde{G}_k^\top$$

for all $0 \leqslant t < \infty$, where $P(t) = \mathbb{E}[X_t X_t^\top]$. Simple calculation shows that

$$\sum_{k=1}^{m} G_k P(t) G_k^\top = \sum_{k=1}^{m} \left( \begin{bmatrix} G_{k,11} & G_{k,12} \\ G_{k,21} & G_{k,22} \end{bmatrix} \begin{bmatrix} \xi\xi^\top & \xi(m(t)^{(-1)})^\top \\ m(t)^{(-1)}\xi^\top & P(t)^{(-1)} \end{bmatrix} \begin{bmatrix} G_{k,11}^\top & G_{k,21}^\top \\ G_{k,12}^\top & G_{k,22}^\top \end{bmatrix} \right),$$

Then one can get that the $(2,2)$-th block entry of the matrix $\sum_{k=1}^{m} G_k P(t) G_k^\top$ is the same as the $\sum_{k=1}^{m}(\ldots)$ part in the ODE corresponds to $P(t)^{(-1)}$ (i.e., Equation (46)), since $\sum_{k=1}^{m} G_k P(t) G_k^\top = \sum_{k=1}^{m} \tilde{G}_k P(t) \tilde{G}_k^\top$, then the $\sum_{k=1}^{m}(\ldots)$ part in the ODEs correspond to both $P(t)^{(-1)}$ and $\tilde{P}(t)^{(-1)}$ are the same. Thus, it is readily checked that $P(t)^{(-1)} = \tilde{P}(t)^{(-1)}$ for all $0 \leqslant t < \infty$.

Though we cannot explicitly show that the post-intervention distribution is identifiable, showing that the first- and second-order moments of the post-intervention process is identifiable can indicate the identification of the post-intervention distribution to a considerable extent.

# E   Conditions for identifying the generator of a linear SDE with multiplicative noise when its explicit solution is available

**Proposition E.1.** *Let $x_0 \in \mathbb{R}^d$ be fixed. The generator of the SDE* (2) *is identifiable from $x_0$ if the following conditions are satisfied:*

C1  $\mathrm{rank}([x_0|Ax_0|\dots|A^{d-1}x_0|H_{.1}|AH_{.1}|\dots|A^{d-1}H_{.1}|\dots|H_{.d}|AH_{.d}|\dots|A^{d-1}H_{.d}]) = d$,

C2  $\mathrm{rank}([v|\mathcal{A}v|\dots|\mathcal{A}^{(d^2+d-2)/2}v]) = (d^2+d)/2$,

C3  $AG_k = G_k A$ and $G_k G_l = G_l G_k$ for all $k,l = 1,\dots,m$.

*where $H := \sum_{k=1}^m G_k x_0 x_0^\top G_k^\top$, and $H_{.j}$ stands for the $j$-th column vector of matrix $H$, for all $j = 1,\cdots,d$. And $\mathcal{A} = A \oplus A + \sum_{k=1}^m G_k \otimes G_k \in \mathbb{R}^{d^2 \times d^2}$, $\oplus$ denotes Kronecker sum and $\otimes$ denotes Kronecker product, $v$ is a $d^2$-dimensional vector defined by $v := \mathrm{vec}(x_0 x_0^\top)$, where $\mathrm{vec}(M)$ denotes the vectorization of matrix $M$.*

*Proof.* Let $\tilde{A}, \tilde{G}_k \in \mathbb{R}^{d \times d}$ and $\tilde{A}\tilde{G}_k = \tilde{G}_k \tilde{A}$, $\tilde{G}_k \tilde{G}_l = \tilde{G}_l \tilde{G}_k$ for all $k,l = 1,\dots,m$, such that $X(\cdot; x_0, A, \{G_k\}_{k=1}^m) \stackrel{\mathrm{d}}{=} X(\cdot; x_0, \tilde{A}, \{\tilde{G}_k\}_{k=1}^m)$, we denote as $X \stackrel{\mathrm{d}}{=} \tilde{X}$, we will show that under our identifiability condition, for all $x \in \mathbb{R}^d$, $(A, \sum_{k=1}^m G_k x x^\top G_k^\top) = (\tilde{A}, \sum_{k=1}^m \tilde{G}_k x x^\top \tilde{G}_k^\top)$. By applying the same notations used in the proof of Theorem 3.5, in the following, we denote $A_1 := A$, $A_2 := \tilde{A}$, $G_{1,k} := G_k$ and $G_{2,k} := \tilde{G}_k$, and denote $X \stackrel{\mathrm{d}}{=} \tilde{X}$ as $X^1 \stackrel{\mathrm{d}}{=} X^2$.

We first show that $H_1 = H_2$ ($H_i := \sum_{k=1}^m G_{i,k} x_0 x_0^\top G_{i,k}^\top$). Indeed, since $X^1, X^2$ have the same distribution, one has

$$\mathbb{E}[f(X_t^1)] = \mathbb{E}[f(X_t^2)] \tag{47}$$

for all $0 \leqslant t < \infty$ and $f \in C^\infty(\mathbb{R}^d)$. By differentiating (47) at $t = 0$, one finds that

$$(\mathcal{L}_1 f)(x_0) = (\mathcal{L}_2 f)(x_0), \tag{48}$$

where $\mathcal{L}_i$ is the generator of $X^i$ ($i = 1, 2$). Based on the Proposition 2.1,

$$(\mathcal{L}_i f)(x_0) = \sum_{k=1}^d \sum_{l=1}^d (A_i)_{kl} x_{0l} \frac{\partial f}{\partial x_k}(x_0) + \frac{1}{2} \sum_{k,l=1}^d (H_i)_{kl} \frac{\partial^2 f}{\partial x_k \partial x_l}(x_0),$$

where $(M)_{kl}$ denotes the $kl$-entry of matrix $M$, and $x_{0l}$ is the $l$-th component of $x_0$. Since (48) is true for all $f$, by taking

$$f(x) = (x_p - x_{0p})(x_q - x_{0q}),$$

it is readily checked that

$$(H_1)_{pq} = (H_2)_{pq},$$

for all $p, q = 1, \dots, d$. As a result, $H_1 = H_2$. Let us call this matrix $H$. That is

$$H := H_1 = \sum_{k=1}^m G_{1,k} x_0 x_0^\top G_{1,k}^\top = \sum_{k=1}^m G_{2,k} x_0 x_0^\top G_{2,k}^\top = H_2.$$

In the proof of Theorem 3.5, we have shown that

$$A_1 A^{j-1} x_0 = A_2 A^{j-1} x_0 \quad \text{for all } j = 1, 2, \dots, \tag{49}$$

next, we will derive the relationship between $A_i$ and $H$. Under condition C3, the SDE system (2) has an explicit solution (cf. [25]):

$$X_t := X(t; x_0, A, \{G_k\}_{k=1}^m) = \exp\left\{ \left( A - \frac{1}{2} \sum_{k=1}^m G_k^2 \right) t + \sum_{k=1}^m G_k W_{k,t} \right\} x_0, \tag{50}$$

then the covariance of $X_t$, $P(t, t+h) = \mathbb{E}[X_t X_{t+h}^\top]$ can be calculated as

$$
\begin{aligned}
&\mathbb{E}[X_t X_{t+h}^\top] \\
&= \mathbb{E}\left[\exp\left\{\left(A - \frac{1}{2}\sum_{k=1}^m G_k^2\right)t + \sum_{k=1}^m G_k W_{k,t}\right\} x_0 x_0^\top \exp\left\{\left(A^\top - \frac{1}{2}\sum_{k=1}^m (G_k^2)^\top\right)(t+h)\right.\right. \\
&\quad + \left.\left. \sum_{k=1}^m G_k^\top W_{k,t+h}\right\}\right] \\
&= e^{At}e^{-\frac{1}{2}\sum_{k=1}^m G_k^2 t}\mathbb{E}\left[e^{\sum_{k=1}^m G_k W_{k,t}} x_0 x_0^T e^{\sum_{k=1}^m G_k^\top W_{k,t+h}}\right] e^{-\frac{1}{2}\sum_{k=1}^m (G_k^2)^\top (t+h)} e^{A^\top (t+h)},
\end{aligned}
\tag{51}
$$

where

$$
\begin{aligned}
&\mathbb{E}\left[e^{\sum_{k=1}^m G_k W_{k,t}} x_0 x_0^T e^{\sum_{k=1}^m G_k^\top W_{k,t+h}}\right] \\
&= \mathbb{E}\left[e^{\sum_{k=1}^m G_k W_{k,t}} x_0 x_0^T e^{\sum_{k=1}^m G_k^\top W_{k,t}} e^{-\sum_{k=1}^m G_k^\top W_{k,t}} e^{\sum_{k=1}^m G_k^\top W_{k,t+h}}\right] \\
&= \mathbb{E}\left[e^{\sum_{k=1}^m G_k W_{k,t}} x_0 x_0^T e^{\sum_{k=1}^m G_k^\top W_{k,t}} e^{\sum_{k=1}^m G_k^\top (W_{k,t+h}-W_{k,t})}\right] \\
&= \mathbb{E}\left[e^{\sum_{k=1}^m G_k W_{k,t}} x_0 x_0^T e^{\sum_{k=1}^m G_k^\top W_{k,t}}\right]\mathbb{E}\left[e^{\sum_{k=1}^m G_k^\top (W_{k,t+h}-W_{k,t})}\right],
\end{aligned}
\tag{52}
$$

because the Brownian motion $W_{k,t}$ has independent increments.

It is known that, for $Z \sim \mathcal{N}(0,1)$, we have that the $j$th moment is

$$
\mathbb{E}(Z^j) = \left\{\begin{array}{ll} 0, & j \text{ is odd}, \\ 2^{-j/2}\frac{j!}{(j/2)!}, & j \text{ is even}. \end{array}\right.
$$

Since $W_{k,t} \sim \mathcal{N}(0,t)$, we have

$$
\begin{aligned}
\mathbb{E}[e^{G_k W_{k,t}}] &= \mathbb{E}\left[\sum_{j=0}^\infty \frac{(G_k)^j (W_{k,t})^j}{j!}\right] \\
&= \sum_{j=0}^\infty \frac{(G_k)^j \mathbb{E}[(W_{k,t})^j]}{j!} \\
&= \sum_{j=0,2,4\ldots}^\infty \frac{(G_k)^j (t/2)^{j/2}}{(j/2)!} \\
&= \sum_{i=0}^\infty \frac{(G_k^2 t/2)^i}{i!} \\
&= e^{G_k^2 t/2}.
\end{aligned}
$$

Similarly, we have

$$
\mathbb{E}[e^{G_k^\top (W_{k,t+h}-W_{k,t})}] = e^{(G_k^\top)^2 h/2}.
$$

Simple calculation shows that

$$
\mathbb{E}\left[e^{\sum_{k=1}^m G_k^\top (W_{k,t+h}-W_{k,t})}\right] = e^{\sum_{k=1}^m (G_k^\top)^2 h/2}.
\tag{53}
$$

By combining Equations (51), (52) and (53), one readily obtains that

$$
P(t, t+h) = P(t,t)e^{A^\top h},
\tag{54}
$$

we denote $P(t) := P(t,t)$. Set $P_i(t, t+h) = \mathbb{E}[X_t^i (X_{t+h}^i)^\top]$, since $X^1 \overset{d}{=} X^2$, it follows that

$$
P_1(t, t+h) = \mathbb{E}[X_t^1 (X_{t+h}^1)^\top] = \mathbb{E}[X_t^2 (X_{t+h}^2)^\top] = P_2(t, t+h) \quad \forall t, h \geqslant 0.
$$

To obtain information about $A$, let us fix $t$ for now and take $j$-th derivative of (54) with respect to $h$. One finds that

$$
\frac{d^j}{dh^j}\bigg|_{h=0} P(t, t+h) = P(t)(A^\top)^j,
\tag{55}
$$

for all $j = 1, 2, \ldots$. It is readily checked that

$$P_1(t)(A_1^\top)^j = P_2(t)(A_2^\top)^j \quad \forall 0 \leqslant t < \infty. \tag{56}$$

We know the function $P_i(t)$ satisfies the ODE

$$\dot{P}_i(t) = A_i P_i(t) + P_i(t)A_i^\top + \sum_{k=1}^m G_{i,k} P_i(t) G_{i,k}^\top, \quad \forall 0 \leqslant t < \infty,$$

$$P_i(0) = x_0 x_0^\top. \tag{57}$$

In particular,

$$\dot{P}_i(0) = A_i x_0 x_0^\top + x_0 x_0^\top A_i^\top + \sum_{k=1}^m G_{i,k} x_0 x_0^\top G_{i,k}^\top.$$

By differentiating (56) with respect to $t$ at $t = 0$, it follows that

$$A_1 x_0 x_0^\top (A_1^\top)^j + x_0 x_0^\top (A_1^\top)^{j+1} + \left( \sum_{k=1}^m G_{1,k} x_0 x_0^\top G_{1,k}^\top \right)(A_1^\top)^j$$

$$= A_2 x_0 x_0^\top (A_2^\top)^j + x_0 x_0^\top (A_2^\top)^{j+1} + \left( \sum_{k=1}^m G_{2,k} x_0 x_0^\top G_{2,k}^\top \right)(A_2^\top)^j.$$

Since we have known that $A_1^j x_0 = A_2^j x_0$ for all $j = 1, 2, \ldots$, it is readily checked that

$$\left( \sum_{k=1}^m G_{1,k} x_0 x_0^\top G_{1,k}^\top \right)(A_1^\top)^j = \left( \sum_{k=1}^m G_{2,k} x_0 x_0^\top G_{2,k}^\top \right)(A_2^\top)^j,$$

that is $A_1^j H = A_2^j H$ for all $j = 1, 2, \ldots$. Let us denote this matrix $A^j H$. Obviously, by rearranging this matrix, one gets

$$A_1 A^{j-1} H = A_2 A^{j-1} H \quad \text{for all } j = 1, 2, \ldots$$

Therefore, under condition C1, that is $\mathrm{rank}(M) = d$ with

$$M := [x_0 | A x_0 | \ldots | A^{d-1} x_0 | H_{\cdot 1} | A H_{\cdot 1} | \ldots | A^{d-1} H_{\cdot 1} | \ldots | H_{\cdot d} | A H_{\cdot d} | \ldots | A^{d-1} H_{\cdot d}]. \tag{58}$$

if we denote the $j$-th column in $M$ as $M_{\cdot j}$, one gets $A_1 M_{\cdot j} = A_2 M_{\cdot j}$ for all $j = 1, \ldots, d + d^2$ by equations (49) and (58).

This means one can find a full-rank matrix $B \in \mathbb{R}^{d \times d}$ by horizontally stacking $d$ linearly independent columns from matrix $M$, such that $A_1 B = A_2 B$. Since $B$ is invertible, one thus concludes that $A_1 = A_2$.

In the proof of Theorem 3.5, we have shown that when $A_1 = A_2$, under condition C2, for all $x \in \mathbb{R}^d$,

$$\sum_{k=1}^m G_{1,k} x x^\top G_{1,k}^\top = \sum_{k=1}^m G_{2,k} x x^\top G_{2,k}^\top.$$

Thus the proposition is proved. $\qquad \square$

It is noteworthy that Proposition E.1 is established on the explicit solution assumption of the SDE (2), which requires both sets of vectors $\{A, \{G_k\}_{k=1}^m\}$ and $\{\tilde{A}, \{\tilde{G}_k\}_{k=1}^m\}$ to satisfy condition C3. As aforementioned, condition C3 is very restrictive and impractical, rendering the identifiability condition derived in this proposition unsatisfactory. Nonetheless, this condition is presented to illustrate that condition C1 is more relaxed compared to condition A1 stated in Theorem 3.5 when identifying $A$ with the incorporation of $G_k$'s information.

