# OpenReview forum: "Generator Identification for Linear SDEs with Additive and Multiplicative Noise"
_NeurIPS.cc/2023/Conference — NeurIPS 2023 poster_

### Official Review · Reviewer_Huoq · 2023-07-03

**Soundness:** 3 good
**Presentation:** 2 fair
**Contribution:** 3 good
**Rating:** 7
**Confidence:** 1

**Summary:**

This paper presents conditions for identifying the generator of a linear stochastic differential equation (SDE) with additive and multiplicative noise. The authors derive sufficient conditions for identifying the generator of both types of SDEs and offer geometric interpretations.


**Strengths:**

I think this paper is theoretically sound.

**Weaknesses:**

1. It would be great if there are practical examples for motivating readers. It’s hard to evaluate the papers’ practical importance because there are no practical scenarios and simulations.
2. In the introduction, the link between causal inference and the identiability of the SDE is weak. Is the goal to identify the “generator” of the post-interventional stochastic process when only observational process is available?

**Questions:**

1. Could you please give some examples or practical scenarios of the proposed method
2. Could you please provide DAGs or any graphs that describing the data generating process? I want to see graphical settings that enables identification of post-interventional processes.

**Limitations:**

1. I believe that the paper's writing lacks motivation because it lacks examples or practical scenarios to describe practical importance.
2. Without a graphical description of the data generating processes, it's difficult to assess how this result is related to causality.

---

> ### Author Rebuttal · Authors · 2023-08-10
>
> Thank you for your helpful comments.  We address your comments point by point as follows. We have also updated our paper according to your comments, and we believe that the quality has been significantly improved thanks to your insightful comments.
>
> **Answers to weaknesses:**
>
> >1. **W:** practical examples
>
> **A:** Thank you for your comment. We have made necessary adjustments to our manuscript, incorporating practical and illustrative examples that effectively demonstrate the importance of our proposed identifiability conditions.
> - We have added practical scenarios where the identifiability analysis of linear SDEs is important in Introduction section.
> - In Section 4, we have modified the title to "Simulations and examples", accompanied by an additional subsection titled "Illustrative instances of causal inference for linear SDEs". Within this subsection, we presented two examples for both SDE (1) and SDE (3), respectively, to show how our proposed identifiability conditions can ensure reliable causal inference for SDE models.
>
> >2. **W:** link between causal inference and the identifiability of the SDE
>
> **A:** Thank you for your comment. The goal of this paper is to uncover the conditions under which the "generator" of the observational SDE is identifiable from the law of the observational process. The main motivation behind this pursuit lies in causal inference. Specifically, within the domain of dynamic systems modelled in SDEs, the primary goal of causal inference is to uncover the conditions under which "the law of the observational process $\xrightarrow{\text{identify}}$ the laws of post-intervention processes". This intricate task can be addressed by examining the generator of the observational SDE, thereby establishing a connection between "the law of the observational process $\xrightarrow{\text{identify}}$ the generator of the observational SDE $\xrightarrow{\text{identify}}$ the laws of post-intervention processes". The latter aspect of this connection has been established in reference [16], while the former part represents the gap that we address in this paper. In the introduction section, we have made necessary adjustments to enhance the clarity of this connection. The detailed interpretation of this connection, involving complex mathematical formulations and explanations, is presented in Section 2.
>
> **Answers to questions:**
>
> >1. **Q:** examples/practical scenarios
>
> **A:** Thank you for your question. We would like to clarify that an important application of our derived identifiability conditions is to ensure reliable causal inferences for dynamic models governed by linear SDEs. Specifically, for linear SDEs under our proposed identifiability conditions, laws of the post-intervention processes are identifiable from the law of the observational process. In other words, our derived conditions ensure the identification of intervention effects on linear SDE systems using observational data. Note that in cases where the SDE system lacks identifiability, observational data would only suffice for predictive tasks, rendering it unsuitable for establishing reliable causal inferences. Thus, when dealing with real-world phenomena governed by linear SDEs, the application of our proposed identifiability conditions guarantees reliable causal inferences within the system.
>
> As an illustration, linear SDEs are used for modeling the gene expression in the yeast microorganism Saccharomyces Cerevisiae [16], where one aims to identify the system such that making reliable causal inferences when interventions are made. For example, what genes to knock out to achieve optimal growth of an organism. By applying our proposed identifiability conditions ensures reliable predictions of intervention effects on the system.
>
> >2. **Q:** DAGs
>
> **A:** Thank you for your question. In the classic DAG-based framework, the DAG directly represents the causal structure of a system. However, in the context of SDEs, We do not directly provide such a graph representation of causality. This distinction arises because, for SDE models, it is possible to identify laws of the post-intervention processes from the law of the observational process, even when the **signature (graph) cannot be identified** from the law of the observational process. Please refer to [Example 5.5, 16] for the illustration of this statement. This statement also indicates that, when formulating a successful causality theory for SDEs, the relevant concept to consider is **the laws of post-intervention processes**, rather than the signatures, since the former is identifiable from the law of the observational process while the latter is not. This contrasts with the DAG-based case.
>
> **Answers to limitations:**
>
> >1. **L:**  lacks motivation and examples
>
> **A:** Thank you for your comment. We would like to take this opportunity to provide further clarification regarding the motivation behind our work. A great motivation for our work is to uncover identifiability conditions of linear SDEs to ensure reliable causal inferences for dynamic models governed by these systems. Please refer to our response to your first question, where we presented practical scenarios and an illustrative example to show where our proposed conditions are important. In addition, please refer to our response to the first weakness, where we summarized the adjustments we have made to our manuscript, to incorporate practical and illustrative examples.
>
>
> >2. **L:** graphical description
>
> **A:** Thank you for your comment. Please refer to our responses provided for question 2. We would like to reiterate that under our proposed identifiability conditions, the laws of the post-intervention processes are identifiable from the law of the observational process. This capability essentially allows for the identification of causal effects resulting from interventions within the system, thereby establishing a foundation for making reliable causal inferences for SDE models.

---

> > ### Author Response · Authors · 2023-08-13
> >
> > Thank you for your valuable comments. Due to the restriction of a "6000 characters limit" per rebuttal, we encountered limitations in our ability to expound comprehensively upon the DAG problem. Thus, we add essential elucidations within this comment. Specifically, we show an example to illustrate that  "for SDE models, the laws of the post-intervention processes are identifiable from the law of the observational process even in scenarios where the signature is not identifiable from the law of the observational process."
> >
> >  Firstly, we would like to introduce the notion of a "signature", which represents the graph associated with an SDE. Consider the general SDE framework described as:
> >     \begin{equation*}
> >         dX_t = a(X_t)dZ_t,\ \ \
> >         X_0 = x_0,
> >     \end{equation*}
> >     where $Z$ is a $p$-dimensional semimartingale and $a: \mathbb{R}^d \rightarrow \mathbb{R}^{d \times p}$ is a continuous mapping. The signature of the SDE corresponds to the graph denoted as $S$, with a vertex set $\{1, \ldots, d\}$ and an edge from vertex $i$ to vertex $j$ if it holds that there is $k$ such that the mapping $a_{jk}$ is not independent of the $i$-th coordinate [16].
> >     However, in contrast to the classic DAG-based framework, the DAG directly represents the causal structure of a system. In the context of SDEs, We do not directly provide such a graph representation of causality. This distinction arises because, for SDE models, it is possible to identify laws of the post-intervention processes from the law of the observational process, even when the signature cannot be identified from the law of the observational process. To illustrate this intriguing observation, we offer an example [Example 5.5, 16].
> >     Consider the mapping $a: \mathbb{R}^2 \rightarrow \mathbb{R}^{2\times 2}$ defined by
> >     \begin{equation*}
> >         a(x) = \begin{bmatrix}
> >             x_1 & 0 \\\\
> >             x_2^2/\sqrt{x_1^2 + x_2^2} & -x_1x_2/\sqrt{x_1^2 + x_2^2}
> >         \end{bmatrix},
> >     \end{equation*}
> >     whenever $x$ is not zero, and $a(0) = 0$. Consider another mapping $\tilde{a}: \mathbb{R}^2 \rightarrow \mathbb{R}^{2\times 2}$ defined by
> >     \begin{equation*}
> >         \tilde{a}(x) = \begin{bmatrix}
> >             x_1^2/\sqrt{x_1^2 + x_2^2} & x_1x_2/\sqrt{x_1^2 + x_2^2} \\\\
> >             0 & x_2
> >         \end{bmatrix},
> >     \end{equation*}
> >     whenever $x$ is not zero, and $\tilde{a}(0) = 0$.
> >     Define two SDEs:
> >     \begin{equation}
> >         dX_t = a(X_t)dW_t, \ \ \ (1)
> >     \end{equation}
> >     \begin{equation}
> >         dX_t = \tilde{a}(X_t)dW_t. \ \ \ (2)
> >     \end{equation}
> >     Since the first row of $a$ depends only on the first coordinate $x_1$, while the second row depends on both coordinates $x_1$ and $x_2$, SDE (1) corresponds to the following signature:
> >     \begin{equation*}
> >         \hookrightarrow 1 \rightarrow 2 \hookleftarrow,
> >     \end{equation*}
> >     where $\hookrightarrow 1$ denotes there is an edge from $1\rightarrow 1$, and $2 \hookleftarrow$ denotes there is an edge from $2\rightarrow 2$.
> >     On the other hand, the first row of $\tilde{a}$ depends on both coordinates $x_1$ and $x_2$, while the second row of $\tilde{a}$ depends only on the second coordinate $x_2$, SDE (2) corresponds to the following signature:
> >     \begin{equation*}
> >         \hookrightarrow 1 \leftarrow 2 \hookleftarrow.
> >     \end{equation*}
> >     Now that we have shown that SDE (1) and SDE (2) correspond to different signatures. In the following, we will show that laws of the solution (observational) processes of SDE (1) and SDE (2) are the same. Furthermore, it will be established that the laws of the post-intervention processes of the two SDEs are also the same.
> >     Simple calculation shows that
> >     \begin{equation*}
> >         a(x)a(x)^{\top} = \tilde{a}(x)\tilde{a}(x)^{\top} =\begin{bmatrix}
> >             x_1^2 & x_1x_2^2/\sqrt{x_1^2+x_2^2}\\\\
> >             x_1x_2^2/\sqrt{x_1^2+x_2^2} & x_2^2
> >         \end{bmatrix},
> >     \end{equation*}
> >     whenever $x\neq 0$. Since $a(x)a(x)^{\top} = \tilde{a}(x)\tilde{a}(x)^{\top}$, the generators of SDE (1) and SDE (2) are the same. Consequently, the laws of the solution processes of SDE (1) and SDE (2) (i.e., the observational processes) are the same. Thus, we have explicitly constructed two SDEs with the same laws of the solution processes but with different signatures. Consider the intervention $X^2 := \xi$ in SDE (1) yields the post-intervention SDE where the first coordinate satisfies
> >     \begin{equation}
> >         dX_t^1 = X_t^1 dW_t^1, \ \ \ (3)
> >     \end{equation}
> >     while the intervention $X^2 := \xi$ in SDE (2) yields the post-intervention SDE where the first coordinate satisfies
> >     \begin{equation}
> >         dX_t^1 = \cfrac{(X_t^1)^2}{\sqrt{(X_t^1)^2 + \xi^2}}dW_t^1 + \cfrac{X_t^1\xi}{\sqrt{(X_t^1)^2 + \xi^2}}dW_t^2. \ \ \ (4)
> >     \end{equation}
> >
> > **The rest of this example is presented in the following comment**

---

> > > ### Author Response · Authors · 2023-08-13
> > >
> > > **following the previous comment**
> > >
> > > The solutions to both SDE (3) and SDE (4) are It\^o diffusion processes, then according to Proposition 2.1, the generator of SDE (3) on $C_b^2(\mathbb{R})$ is given by
> > >     \begin{equation}
> > >         (\mathcal{L}f)(x) = \cfrac{x^2}{2}\cfrac{d^2f(x)}{dx^2},
> > >     \end{equation}
> > >     and the generator of SDE (4) on $C_b^2(\mathbb{R})$ is given by
> > >     \begin{equation}
> > >     \begin{split}
> > >         (\mathcal{L}f)(x) &=  \cfrac{1}{2}\begin{bmatrix}
> > >             x^2/\sqrt{(x^2+\xi^2)} & x\xi/\sqrt{(x^2+\xi^2)}
> > >         \end{bmatrix}
> > >         \begin{bmatrix}
> > >             x^2/\sqrt{(x^2+\xi^2)}\\\\
> > >             x\xi/\sqrt{(x^2+\xi^2)}
> > >         \end{bmatrix}\cfrac{d^2f(x)}{dx^2}\\\\
> > >         &= \cfrac{x^2}{2}\cfrac{d^2f(x)}{dx^2}
> > >     \end{split}.
> > >     \end{equation}
> > >     Since the generators of the solution processes to both SDE (3) and SDE (4) are the same, the laws of their respective solution processes (i.e., the post-intervention processes) are also the same, even in this case where the signatures are different.
> > >
> > > This example illustrates an intriguing observation: for SDE models, the laws of the post-intervention processes are identifiable from the law of the observational process even in scenarios where the signature is not identifiable from the law of the observational process. One interpretation of this is that for SDEs, the laws of the post-intervention processes will be the same for all signatures which are compatible with the law of the observational process [16]. This observation also indicates that, when formulating a successful causality theory for SDEs, the relevant concept to consider is **the laws of post-intervention processes**, rather than the signatures, since the former is identifiable from the law of the observational process while the latter is not. This contrasts with the DAG-based case.
> > >
> > > Thank you again for your valuable comments. We believe that the quality of our work has been significantly improved thanks to your insightful comments.

---

> ### Comment · Reviewer_Huoq · 2023-08-13
> **Response to Authors**
>
> Thank you for your response. So, the graph can not be specified, because the SDE could presents the system that is non-Markovian (e.g., the system that contains self-feedback such as a cycle)?

---

> > ### Author Response · Authors · 2023-08-14
> > **Response to Reviewer Huoq**
> >
> > Thank you for your comments.  Yes, the graph can not be specified, because the SDE could present a system that is non-Markovian. In addition to the reason that the SDE system contains self-feedback such as a cycle. This is also because the error variables are dependent. Specifically, when the driving semimartingale $Z$ is a  Levy process, the error variables are independent across time but are not independent across coordinates.

---

> > > ### Comment · Reviewer_Huoq · 2023-08-14
> > > **Response from Authors**
> > >
> > > Thank you for the clarification!
> > >
> > > There is a recent effort for developing a causal inference framework for the system having self-feedback: https://arxiv.org/pdf/1611.06221.pdf. Could you please relate your work with this paper?

---

> > > > ### Author Response · Authors · 2023-08-15
> > > > **Response to Reviewer Huoq**
> > > >
> > > > Thank you for your comment. We have taken the time to carefully review the paper you referenced. Regrettably, we encountered challenges when attempting to establish a direct connection between their work and our proposed work. We would like to elucidate the reasons as follows:
> > > > 1. Firstly, their work focuses on the application of Structural Causal Models (SCMs), a framework typically tailored for scenarios involving a finite number of variables with no explicit time component (**static** in nature). Conversely, our work is based on Stochastic Differential Equations (SDEs), thereby entailing the development of causality concepts within a **continuous-time** framework.
> > > > 2. Their research is based on SCMs, wherein causal relationships among variables are expressed in the form of **deterministic**, functional raltionships. While our work is based on SDEs, dealing with causality for **stochastic** processes.
> > > > 3. Their work focuses on perfect interventions, e.g., $X_i = \xi_i$, where $\xi_i$ is a real value. However, our work deals with a more general  forms of interventions, defined as $X_t^l := \zeta(X_t^{-l})$, where $X^{-l}$ denotes the $(d-1)$-dimensional vector that results from the removal of the $l$-th coordinate of $X \in \mathbb{R}^d$ (see Definition 2.2. for more details).  In other words, an intervention of a coordinate of the state variable can be any Lipschitz function of the other $(d-1)$-dimensional coordinates, and the intervention also implicitly depends on time.
> > > >
> > > > In essence, these multifaceted discrepancies collectively contribute to the challenge in drawing a relationship between the work presented in the paper and our proposed research. Thank you for your understanding.

---

> > > > > ### Comment · Reviewer_Huoq · 2023-08-16
> > > > > **Response from Authors**
> > > > >
> > > > > Thank you for your thorough comparisons. I understood as follows. You have selected the generator-based approach to discuss causality in the SDE, as the SCM is not well-suited for discussing causality for non-Markovian and continuous settings.
> > > > >
> > > > > I think this claim makes sense. An SCM is known to be the sound and complete model for any counterfactual variable generating systems (ref. Pearl (2009) Chapter 7 or [Halpern (1998)](https://dl.acm.org/doi/pdf/10.5555/2074094.2074118)) when the variables are finite and no cycles occur. However, to my knowledge, causal models suitable for causal inference when these assumptions are violated (i.e., non-Markovian and continuous settings) have received less attention.
> > > > >
> > > > > Since the discussion with you addresses my concerns and questions, I will raise the score from 6 to 7. Thank you!

---

> > > > > > ### Author Response · Authors · 2023-08-16
> > > > > > **Response to Reviewer Huoq**
> > > > > >
> > > > > > Thank you for your acknowledgment and the increase in our score. Your recognition of the importance of our proposed theories is greatly appreciated. Once again, we extend our gratitude for your valuable response and support.

---

### Official Review · Reviewer_8vBF · 2023-07-11

**Soundness:** 3 good
**Presentation:** 2 fair
**Contribution:** 2 fair
**Rating:** 4
**Confidence:** 2

**Summary:**

This paper addresses the interventional identifiability of two classes of stochastic differential equations. Theoretical results are illustrated with simulation of identifiable and non-identifiable cases.

**Strengths:**

The paper is carefully written and the theoretical results appear sound, although I could not check the proofs in detail.

**Weaknesses:**

My main concern is whether the target audience for this paper is really NeurIPS. Considering myself one of relatively few in the NeurIPS community with some familiarity with both causality and (to a limited extent) with SDE, I have a hard time making sense of the contribution. Although the presentation of the framework of reference [16] for causality with SDE is interesting, it is quite challenging to get an intuition of the concrete cases for which this framework is relevant: the defined interventions are static (not time dependent) and I have a hard time to imagine in which application we need the heavy toolbox of SDEs to model static interventions, which moreover are constrained to always start from the exact same point in state space. The paper does not really help, neither with justifying this framework nor with providing some intuitions.

Moreover, the paper is written in a quite confusing way, at least for a non expert in SDEs: after introducing "standard" SDE based on Brownian motion in Eqs. (1) and (3), we are introduced in Lemma 2.1 and above with SDEs based on integrating Levy processes (and not being explained what is this). Then we move on to talking about Ito diffusion processes (Propostion 2.1) with an equation of the generator exhibited, although we have not being told at any time, even roughly, what a generator is for a stochastic process.

I can imagine the results may be relevant for a mathematical journal, but if the authors what to target the ML/causality community, there is a lot of work to do on the manuscript to keep only the essential mathematical concepts, explain them well with reduced formalism, and convey intuition of why this is relevant to the NeurIPS readership interested in causality and time series.

Additionally, if I risk myself on evaluating the importance/novelty of the key results of this work, they resemble standard results on characterizing the set of states reachable by some classes of Markov chains in general state space. I can imagine there are also fairly standard results for this question in the context of SDEs, and the rank constraint of Theorem 3.3 is not very surprising when considering the dimension of the subspace reachable from a fixed initial point. Given the whole state space is explored, again I can imagine identifying the generator boils down to exploiting standard results. I am left wondering whether this result really deserves a new paper that addresses it exclusively (meaning: without exhibiting any concrete problem where it is relevant and helpful).

**Questions:**

How the continuous time identifiability results relate to identifiability of equivalent discrete time models (Markov chains with additive/multiplicative noise in general state space)?

**Limitations:**

Limitations are addressed in the discussion. However, how restrictive is the interventional framework for concrete problems is not addressed (see "weaknesses" section).

---

> ### Author Rebuttal · Authors · 2023-08-10
>
> Thank you for your helpful comments.  We address your comments point by point as follows. We have also updated our paper according to your comments, and we believe that the quality has been significantly improved thanks to your insightful comments.
>
>
> **Answers to weaknesses:**
>
> >1. **W:** target audiance of this paper and interventions.
>
> **A:** With regard to the concern of our target audiance, please refer to our response to the third weakness, where we present the adjustments we have made to enhance the readability of this paper to the ML/causality community.
>
> Regarding interventions, we would like to clarify that although the interventions considered in this work are not explicitly dependent on time, they are not static since they implicitly depend on time  thorough the state variable $X_t$. Specifically, we define interventions in Definition 2.2, where they are described as $X_t^l := \zeta(X_t^{-l})$. Here, $X_t^l$ represents the $l$-th coordinate of the state variable $X\in \mathbb{R}^d$ at time $t$, and $X_t^{-l}$ denotes the $(d-1)$-dimensional vector resulting from the removal of the $l$-th coordinate of state variable at time $t$. In addition, in Lemma 2.1 we state that the intervention function $\zeta: \mathbb{R}^{d-1} \rightarrow \mathbb{R}$ only needs to be Lipschitz, which is a rather mild condition. In other words, an intervention of a coordinate of the state variable can be any Lipschitz function of the other $(d-1)$-dimensional coordinates, since each coordinate (e.g., $X_t^l$) depends on time, the intervention also implicitly depends on time.
>
> >2. **W:**  presentation of the paper.
>
> **A:** Thank you for bringing this matter to our attention. We have made necessary adjustments to enhance the presentation of our manuscript. Specifically, we have changed the title of Section 2 to "Background knowledge", which now consists of two subsections. In Subsection 2.1, now titled "Background knowledge of linear SDEs," includes:
>
> - The solutions of the two linear SDEs of interest (lines 64, 67-78).
> - Definition 2.1 (lines 79-89)
> - A gentle introduction to the generator of a stochastic process.
> - The general form (Eq. (10)) and its generator (lines 141-148).
>
> In Subsection 2.2, we have retained the (formerly Subsection 2.3).
>
> we have introduced a new subsection, 3.1, titled "Prerequisites", includes
> - Lemma 2.2 (lines 133-141).
> - Proposition 2.2  (lines 151-157).
>
> >3. **W:**  target audience
>
> **A:** Thank you for this comment, we would like to clarify that this paper is aiming to target the ML and causality community. We have made necessary adjustments to enhance the readability and accessibility of this paper to the ML/causality community. Includes:
> - In Introduction, added practical scenarios to motivate our proposed work.
> - In Section 4,  added a new subsection titled "Illustrative instances of causal inference for linear SDEs" to present two examples for both SDE (1) and SDE (3), respectively, to show how our proposed identifiability conditions can ensure reliable causal inference for SDE models.
> -  We have reduced unnecessary mathematical concepts and formulas, instead, we incorporated gentle introductions and relevant citations to elucidate these concepts, thereby ensuring a more reader-friendly presentation.
>
> >4. **W:** novelty
>
> **A:** Thank you for your comment. Since the linear SDEs with additive noise (SDE (1)) is a relatively simple model, the rank constraint of Theorem 3.3 may not be very surprising under our settings. However, to the best of our knowledge, there are no fairly standard results for this question under the same setting as ours. We would greatly appreciate it if you could share some relevant references.
> Moreover, we have derived the identifiability conditions for the linear SDEs with multiplicative noise (SDE (3)). Incorporating state variables into the diffusion coefficient significantly complicates the SDE model and the evolution of the state variables. As a result, the identifiability of this particular model remains significantly under-explored. Our work presents the first systematic study of the identifiability analysis of this particular model. We believe that our proposed identifiability conditions can effectively facilitate reliable causal inference for dynamical systems governed by linear SDEs.
>
> **Answers to questions:**
>
> >**Q:**  relation to discrete time models
>
> **A:** Thank you for your question. We would like to highlight that our proposed continuous-time identifiability conditions are considerably more relaxed when compared to the existing identifiability conditions based on discrete-time models. Specifically, we have demonstrated that our proposed identifiability conditions for both SDEs with additive or multiplicative noise are generic (lines 219-221 and lines 256-257). However, in the "Related Work" section, we have discussed that the present identifiability conditions based on discrete-time models tend to be restrictive.
>
> Furthermore, continuous-time models and discrete-time models represent two distinct approaches used to address different problems. Continuous-time models are employed to study systems that evolve continuously over time and are described by differential equations. While discrete-time models are designed to analyze systems that evolve in discrete steps and are described using difference equations. As a result, there is no direct correlation between our proposed continuous-time identifiability conditions and the current identifiability conditions of relevant discrete-time models.
>
>
> **Answers to limitations:**
>
> >**L:** interventions
>
> **A:** Thank you for your comment. Please refer to our answer regarding the concern related to interventions in the "weakness" section. In our explanation, we have elucidated that the definition of intervention is rather generic, which leads us to maintain our position that it is unnecessary to discuss the restriction of the interventional framework as a limitation.

---

> > ### Comment · Reviewer_8vBF · 2023-08-15
> > **Reply to rebuttal**
> >
> > Thank you for your reply.
> > However, at this step, I am not sure what tangible argument it brings to update my previous assessment.
> >
> > (1) Regarding the fact that interventions are not static, because they are state dependent. Point well taken, static may not be the best word to describe it, but that interventional setting remains very special: a time-invariant instantaneous relation between the state of intervened and unintervened variables. Do you have any practical scenario (in a real system), where we would consider such intervention (in the non-constant case)?
> >
> > (2-3) I appreciate the claim that the paper has been improved, but without seeing it, it is difficult to assess whether the improvement will be enough.
> >
> > (4) Regarding novelty, I understand the multiplicative noise case has some challenges, and I myself will have a hard time covering the huge SDE litterature to find related results. But that is my point: **why submit to NeurIPS a mathematical result on the identifiability of the generator of a specific SDE, with a very specific setting (starting from a single state), with no clear justification why this should be of special relevance to the ML-causality community?** This seems only related to causality through the fact that generator identifiability is a requirement for effect identifiability in a very special framework, which I am not aware has been used in the ML causality field before.
> >
> > For example, after a quick search, I could find a paper even providing confidence bounds on the estimation of infinitesimal generator for a fairly large family of models with unknown dynamics based on sampled trajectories. This is looks much stronger than identifiability.
> >
> > Data-driven estimation of infinitesimal generators of stochastic systems. Nejati et al. 2021 (Proceedings IFAC)
> >
> > I do appreciate mathematical contributions, but I doubt the novelty of that one can be evaluated by the NeurIPS community.
> >
> > (A) Regarding the connection between discrete and continuous time models, I tend to have a different view: both can address overlapping problems. There are further considerations, for example, continuous time settings are of course elegantly suited to irregularly sampled processes. It would have been good to better justify the practical use of this framework, given it has not really taken off yet (and it was introduced 10 years ago).

---

> > > ### Author Response · Authors · 2023-08-15
> > > **Reply to Reviewer 8vBF**
> > >
> > > Thank you for your comment, we address your comments as follows.
> > >
> > > (1) Regarding the interventions. Let us consider a few examples. In the medical realm, such as in diabetes management, insulin doses may be adjusted in response to blood glucose levels. Similarly, within the sphere of ecosystem management, interventions involving specific species often rely on interactions with other species to restore natural habitats, control invasive species, and promote biodiversity. It is important to highlight that our setting of interventions provides a more general interventional framework, except for constant interventions. We provide the potential to explore interventions in a more general and dynamic way.
> > >
> > > (2)-(3) As you may have known, regretfully, at this stage we are not able to upload our new revision. Should you possess any specific concerns, please feel free to communicate them with us. We are committed to addressing your concerns comprehensively within our response.
> > >
> > > (4) Regarding the topic of novelty, we would like to take a moment to provide further insight into our focus on deriving identifiability conditions for the generator of linear SDEs. The pursuit of identifying this generator stems from our deep commitment to ensuring reliable causal inference, a vital goal in the realm of causal inference for dynamic systems modelled by SDEs. Given the pervasive application of SDEs in modelling real-world data influenced by random fluctuations, their utilization within the ML community to represent stochastic dynamic systems is widespread. For instance, the emergence of neural SDE (NSDE) models has garnered substantial attention in recent years, sparking a surge of interest within the ML community. Nevertheless, it is important to acknowledge that the absence of comprehensive causal studies of NSDEs limits their applicability to forecasting tasks rather than robustly predicting the behavior of SDE systems under interventions. This underscores the pivotal role that causal analysis of SDEs plays and motivates our present work.
> > >
> > > While it is recognized that linear SDEs may not aptly capture the complexity of systems typically addressed in the machine learning community, the reasons we begin with linear SDEs are as follows. The causal studies for SDEs are still at the **early stage**, even for linear SDEs which have remained underexplored. It is worth emphasizing that the evolution from simpler linear cases to more intricate nonlinear instances is a common progression in research studies. And the study of the linear cases typically lays the foundation and paves the road for the study of the more intricate nonlinear instances. This journey is shared across various model classes, for instance, linear regression, though inadequate for representing intricate real-world systems, remains a foundational pillar of machine learning.
> > >
> > > We firmly believe that our investigation into the identifiability analysis of linear SDEs will furnish a solid cornerstone for causal studies within the ML framework involving SDEs. This, in turn, has the potential to kindle heightened research enthusiasm and concerted efforts directed towards the causal exploration of SDEs within the ML community. We hold the view that through persistent dedication to this research avenue, we will gradually unveil theories concerning causality and identifiability applicable to more intricate nonlinear systems. These insights, when integrated into real-world machine learning applications, stand to yield substantial benefits within the ML community.
> > >
> > > We would like to clarify that the involvement of many mathematical equations in our work is due to the inherent nature of SDEs. We are aiming to target the ML and causality community and we have made efforts to enhance the accessibility of our paper to the ML community. We firmly believe that the causal studies of SDEs can significantly contribute to the ML community. Nevertheless, we also recognize that fostering a deeper understanding and broader adoption of SDEs within the ML community will require both time and ongoing refinement of causal studies associated with SDEs.
> > >
> > > Regarding the mentioned paper titled “Data-driven estimation of infinitesimal generators of stochastic systems”, it is important to note that we focus on different goals. Their work focuses on the estimation of the generator, while our work focuses on uncovering conditions under which the generator is identifiable, such that facilitating reliable causal inference of the system.

---

> > > ### Author Response · Authors · 2023-08-15
> > > **Reply to Reviewer 8vBF**
> > >
> > > (A)	Regarding the connection between discrete and continuous time models. We agree that the continuous time settings are elegantly suitable for the study of irregularly sampled processes. Since irregularly sampled processes are ubiquitous in real-world data and are relatively underexplored. We believe that our study will benefit the study of irregularly sampled processes.
> > >
> > > However, we would like to clarify that, the reason why we can use continuous models to model the irregularly sampled processes (observations) is because here we assume the observations are generated from an underlying continuous model instead of a discrete model. What we would like to say is that it is important to avoid any confusion between the concept of a discrete model and that of discretely sampled observations.
> > >
> > > Thank you again for your valuable comments. We believe that the quality of our work has been significantly improved thanks to your insightful comments.

---

> > > ### Author Response · Authors · 2023-08-20
> > > **Reply to Reviewer 8vBF**
> > >
> > > Dear Reviewer 8vBF,
> > >
> > > We thank you again for the time and effort you have devoted to reviewing our manuscript. The discussion period will end soon, we will appreciate your feedback, and specifically, if you have further questions or concerns, we hope we will have the opportunity to address them.
> > >
> > > Once again, we extend our sincere gratitude for your valuable comments and we are looking forward to your feedback.
> > >
> > > Sincerely, authors of paper 5668

---

### Official Review · Reviewer_5yfh · 2023-07-17

**Soundness:** 3 good
**Presentation:** 2 fair
**Contribution:** 2 fair
**Rating:** 4
**Confidence:** 1

**Summary:**

This article derives relationships for identifying generators of linear SDEs in both the case of additive and multiplicative noise. It is written in a very classically statistical manor, with a great number of citations from the statistical research community.

**Strengths:**

This article identifies two apparently unresolved issues from stochastic differential equations and solves them.

I thank the authors for their thoughtful discussion in the rebuttal process.

**Weaknesses:**

This article assumes a high understanding of statistics, and unfortunately this is not my field. I do not have the expertise to review the proofs in this article. I think the article could do a better job at explaining the relevance of this topic to the broader machine learning community. Excluding the unpublished (arxiv) citations, it looks to me like there are only two published articles cited in the ML community, and both of those articles seem only tangentially related to the work in this article. The presentation of this article appears to be quite good for a pure-statistical research community.

- All of the proofs claimed in this article are in the appendices of their article. While I know this is standard practice, technically NeurIPS appendices are un-reviewed (and in practice they are criminally under-reviewed). In my own field, this has led to an increased number of incorrect results being published, cited, and propagated through the research community. Because reviewers like me cannot assess the validity of the results claimed in this article, I worry that this may also happen in statistics. For this reason, I am giving a low-confidence rejection. If I see that other reviewers make good points attesting to the validity of the theory, I will change my recommendation.
- Throughout the article there are many forward-references (e.g., on line 25, the reader has no idea what the matrix "A" means, in lines 31 and 32, the equations (1) and (3) are referenced without being introduced first, etc.) which make for a hard experience for the reader.
- Clerically, one could order the citations so that they appear in increasing order (e.g., in line 19, [4, 34, 45])
- Line 39 - 41: The argument for uniqueness is not clear from this statement.
- Line 69: I think you mean "are commutative"
- The sectioning in this article does not quite make sense to me. The pagraph "We know that both the SDE (1) and SDE (3) ..." which appears in Section 2.2, actually talks about both the content in Sections 2.1 and 2.2.
- Inconsistent citations throughout. For instance, some articles are cited using the lowercase "volume" while others do not use the abbreviation for a volume at all (e.g., 61(11): 1038-1044) -- this citation style should be consistent.
- The citation also contain many mis-capitalizations (odes vs ODEs, SIAM vs Siam), typos, duplicated authors [21], and clerical errors.
- There are a lot of citations of unpublished material; a small number of recent arxiv publications is fine in this community, but some are missing any references alltogether, e.g.,  [13] and [24].

**Questions:**

Why is this article relevant to the machine learning community?

**Limitations:**

This article is written for pure-statisticans and it is not really legible for many in the ML community.

---

> ### Author Rebuttal · Authors · 2023-08-10
>
> Thank you for your helpful comments.  We address your comments point by point as follows. We have also updated our paper according to your comments, and we believe that the quality has been significantly improved thanks to your insightful comments.
>
> **Answers to weaknesses:**
>
> >1. **W:**  Proofs concern.
>
> **A:** We acknowledge your concern regarding the correctness of the proofs presented in this article, considering that you may not possess the specialized expertise to thoroughly review them. We would like to assure you that all the proofs have been checked multiple times to ensure their accuracy. Additionally, we wish to highlight that the simulation results presented in Section 4 serve as a substantial validation of the proposed theoretical results.
>
> >2. **W:**  Regarding forward references.
>
> **A:** Thank you for bringing this matter to our attention. We have made the necessary adjustments to our manuscript. Specifically, we have relocated the concise introduction of the two linear SDEs (i.e., Eq.(1) and Eq.(3)) to the appropriate position of the introduction. In addition, we conducted a thorough review of the entire manuscript to ensure the elimination of any further instances of forward references.
>
> >3. **W:** Order the citations in increasing order
>
> **A:** Thank you for pointing this out. We have conducted a thorough review of the entire manuscript to ensure all multiple citations have been arranged in increasing order.
>
> >4. **W:** Line 39 - 41: The argument for uniqueness is not clear from this statement.
>
> **A:** Thank you for pointing this out. We have made the necessary adjustments to our manuscript. The updated version reads as follows: "For example, in the SDE (1), parameter $G$ cannot be uniquely identified since one can only identify $GG^{\top}$ based on the law of the observational process [16, 26]."
>
> >5. **W:** Line 69: I think you mean "are commutative"
>
>  **A:** Thank you for pointing this out. We have updated our manuscript accordingly.
>
> >6. **W:** The sectioning concern in Section 2.
>
> **A:** Thank you for bringing this matter to us. As we have mentioned that in order to get rid of the forward references, we have relocated the introduction of the two linear SDEs (i.e., Eq.(1) and Eq.(3)) to the  Introduction section. Here, within this section, we have included a new subsection titled "Background knowledge of linear SDEs" to introduce the solutions of the linear SDEs, as well as the paragraph you mentioned and beyond.
>
> >7. **W:** citation problems
>
> **A:** Thank you for bringing this matter to our attention. We have made a thorough review of all the citations and rectified the mentioned inconsistent issues and errors. Regarding those citations lacking proper references, we have added the relevant sources.
>
> **Answers to questions:**
>
> >**Q:** Why is this article relevant to the machine learning community?
>
> **A:** Thank you for your question. SDEs are a powerful mathematical tool for modeling dynamic systems subject to random fluctuations. These equations find wide-ranging applications in the modeling of real-world data, spanning fields such as finance, physics, biology, and engineering, among others. They are used in the machine learning field in various ways to model and analyze complex and dynamic systems affected by randomness and uncertainty. These applications include (1) time series modeling and forecasting; (2) generative models; and (3) causal inference for dynamic models affected by randomness.
>
> It is essential to acknowledge that our current understanding of the identifiability analysis of these models remains limited. This analysis is crucial for reliable causal inferences of these models and is the main motivation of our work. Through this work, we endeavor to pave the way for a comprehensive understanding of the identifiability of linear SDE models, ultimately enabling reliable causal inferences regarding system behavior under interventions, thus facilitating the aforementioned (1) and (3) applications of SDEs in the machine learning community.
>
> **Answers to limitations:**
>
> >**L:** This article is written for pure-statisticans and it is not really legible for many in the ML community.
>
> **A:** Thank you for your comment. we would like to clarify that this paper is aiming to target the ML and causality community. We have made necessary adjustments to enhance the readability and accessibility of this paper to the ML/causality community. We would like to summarize some of these improvements as follows:
> - We have made significant revisions to the original third paragraph in Introduction section. Specifically, we focused on conveying the intuitions and enhancing the clarity of why our proposed work can facilite reliable causal inference in dynamic models governed by linear SDEs, and how the machine learning community can benefit from our proposed theories.
> - In Section 4, we have modified the title to "Simulations and examples", accompanied by an additional subsection titled "Illustrative instances of causal inference for linear SDEs". Within this subsection, we present two examples for both SDE (1) and SDE (3), respectively, to show how our proposed identifiablity conditions can ensure reliable causal inference for SDE models. Specifically, we have shown how under our proposed identifiability conditions, "the law of the observational process $\xrightarrow{\text{identify}}$ the generator of the observational SDE $\xrightarrow{\text{identify}}$ the laws of post-intervention processes". By providing these examples afford readers a comprehensive understanding of how our proposed identifiability conditions are used for reliable causal inference.
> - Furthermore, we have reduced unnecessary mathematical concepts and formulas, instead we incorporated gentle introductions and relevant citations to elucidate these concepts, thereby ensuring a more reader-friendly presentation.

---

> > ### Comment · Reviewer_5yfh · 2023-08-16
> >
> > Thank you very much for your explanations. I have reviewed your comments, as well as those from other reviewers. I also found Reviewer b8xq 's comment about identifiability of NSDEs to also be particularly helpful in conveying the relevance to the ML community. I am now thoroughly convinced that this topic is relevant to the community.
> >
> > I was looking forward to reading the revised article, with hopes to verify that has been significantly improved for legibility by the broader ML community. However, I cannot find the revised version of the article under the Revisions history, and the current article PDF above appears to be identical to the one submitted. I am somewhat new to using OpenReview -- is it possible to view the revised article?
> >
> > Regarding the theoretical issues, I have seen that there are still unresolved theoretical issues raised by other referees (e.g., b8xq has not verified the validity of your statements about Theorem 3.4), so my score will remain the same for now.

---

> > > ### Comment · Area_Chair_ZC4y · 2023-08-16
> > >
> > > Thanks for reviewing for NeurIPS for the first time! Some guidelines: At this stage, only the rebuttal is visible - the authors are not able to upload a new revision until after the final decision. Please use the author's rebuttal as a guide and if you are unsure about the sufficiency of any points please reply to the authors and ask any clarifying questions or make any requests you wish.
> > >
> > > Best,
> > > the AC

---

> > > ### Author Response · Authors · 2023-08-16
> > > **Response to Reviewer 5yfh**
> > >
> > > Thank you for your comment and recognition of the pertinence of our work within the Machine Learning community.
> > >
> > > Regarding the revised article, we would like to draw your attention to the comment made by Area Chair ZC4y, who noted that, " At this stage, only the rebuttal is visible - the authors are not able to upload a new revision until after the final decision.” Consequently, we are unable to submit the new revision at this stage. Should you possess any specific concerns, please feel free to communicate them with us. We are committed to addressing your concerns comprehensively within our response.
> > >
> > > Regarding theoretical issues, we would like to reiterate that we have checked all the proofs multiple times and used simulations (Section 4) to ensure and illustrate their correctness. Addressing the inquiry concerning Theorem 3.4, raised by Reviewer b8xq, it is noteworthy that after reading our rebuttal, the reviewer did not raise any further concerns regarding this question. We interpret this as an indication that the question has been satisfactorily addressed and elucidated. However, to further ensure clarity, we intend to reach out to Reviewer b8xq in an endeavour to provide explicit clarifications on this matter. Hopefully, reviewer b8xq will respond to our comment and thus dispel your concern.
> > >
> > > Once again, we extend our gratitude for your valuable response.

---

> > > ### Author Response · Authors · 2023-08-20
> > > **Response to Reviewer 5yfh**
> > >
> > > Dear Reviewer 5yfh,
> > >
> > > We are writing to inform you that your theoretical concern regarding to Theorem 3.4, as raised by Reviewer b8xq, has been addressed. Following a more detailed discussion with Reviewer b8xq, we are pleased to announce that Reviewer b8xq now possesses a clearer understanding of our work and has expressed no further concerns regarding this matter. Should you have any more concerns, please feel free to communicate them with us.
> > >
> > > We extend our sincere gratitude for the time and effort you have devoted to reviewing our manuscript and we are looking forward to your feedback.
> > >
> > > Sincerely, authors of paper 5668

---

> > > > ### Comment · Reviewer_5yfh · 2023-08-21
> > > >
> > > > Dear Authors,
> > > >
> > > > Thank you very much for your dedicated time in submitting the article and your responses. I have reviewed the discussion with the Reviewer b8xq, and even though they no longer have questions, they still have not affirmed the validity of the proof. For this reason, I have kept my recommendation the same. However, to reflect that I am not really an expert in the area, I have lowered my confidence score for this article.

---

> > > > > ### Author Response · Authors · 2023-08-21
> > > > > **Response to Reviewer 5yfh**
> > > > >
> > > > > Dear Reviewer 5yfh,
> > > > >
> > > > > Thank you for your feedback. As we acknowledge that it is not within the reviewer's duty to affirm the validity of our proof, and consequently, we are unable to request reviewer b8xq to do so. However, we have made diligent efforts to address all of reviewer b8xq's concerns regarding this matter.
> > > > >
> > > > > We fully understand and respect your final decision and we thank you for lowering our confidence score. We genuinely appreciate the time and effort you have devoted to reviewing our manuscript.
> > > > >
> > > > > Thank you once again.
> > > > >
> > > > > Sincerely, authors of paper 5668

---

### Official Review · Reviewer_GdBe · 2023-07-21

**Soundness:** 2 fair
**Presentation:** 3 good
**Contribution:** 2 fair
**Rating:** 5
**Confidence:** 2

**Summary:**

This paper derives some sufficient conditions for identifying linear stochastic differential equations (SDEs). The validity of the identifiability conditions is clarified on synthetic simulation data.

**Strengths:**

- The identification problem of ordinary/stochastic differential equations has been a hot topic in machine learning community, but the condition for identification seems to be relatively overlooked. The paper could raise a good question about it.
- The paper is well-motivated and makes a technically solid contribution.

**Weaknesses:**

- The paper focuses on linear SDEs. They are very simple models and, to the best of my knowledge, are rarely applied to modern problems of system identification. Thus, the paper’s contribution is very limited and not impactful.
- The paper emphases its contribution to reliable causal inference, but the experiments don’t include causal inference task. How the derived sufficient conditions benefits the causal inference task remains to be clear.

**Questions:**

What are the situations where the derived conditions for identifying linear SDEs so useful in practice?

---

> ### Author Rebuttal · Authors · 2023-08-10
>
> Thank you for your helpful comments.  We address your comments point by point as follows. We have also updated our paper according to your comments, and we believe that the quality has been significantly improved thanks to your insightful comments.
>
> **Answers to weaknesses:**
>
> >1. **W:** The paper focuses on linear SDEs. They are very simple models and, to the best of my knowledge, are rarely applied to modern problems of system identification. Thus, the paper’s contribution is very limited and not impactful.
>
> **A:** Thank you for your comment. We would like to clarify that there are also some real-world applications used linear SDEs. For example, linear SDEs are wildly used in financial modeling for tasks like asset pricing, risk assessment, and portfolio optimization. Where they are used to model the evolution of financial variables, such as stock prices and interest rates. And linear SDEs are also used in genomic research, for instance, they are used for modeling the gene expression in the yeast microorganism Saccharomyces Cerevisiae [16], where one aims to identify the system such that making reliable causal inferences when interventions are made. For example, what genes to knock out to achieve optimal growth of an organism.
>
> Identifiability analysis of SDEs is essential for reliable causal inferences of dynamic models governed by SDEs. However, the inherent complexity arising from the stochastic nature of variable evolution poses challenges in addressing identifiability tasks. Consequently, the identifiability analysis for SDE systems is still at the **early stage**, even for linear SDEs (especially for multiplicative noise cases), remain understudied. This is one of our motivations for starting from the linear cases.
>
> Our other motivation is that the evolution from simpler linear cases to more intricate nonlinear instances is a common progression in research studies. This journey is shared across various model classes, including state space models, Ordinary Differential Equations (ODEs), among others. The foundation typically rests on comprehending linear cases first. Consequently, linear cases serve as a good starting point for identifiability analysis of SDEs.
>
> We believe that our proposed work not only serves to enhance reliable causal inferences for linear SDEs but also paves the way for a comprehensive identifiability analysis of more intricate SDE models. This advancement holds the potential to significantly elevate the accuracy of causal inferences within dynamic systems governed by SDEs.
>
> >2. **W:** The paper emphasizes its contribution to reliable causal inference, but the experiments don’t include causal inference task. How the derived sufficient conditions benefit the causal inference task remains to be clear.
>
> **A:** Thank you for bringing this matter to our attention. We have updated our manuscript and given two examples for both SDE (1) and SDE (3), respectively, to show how our proposed identifiability conditions can ensure reliable causal inference for SDE models. Specifically, we have shown how under our proposed identifiability conditions, "the law of the observational process $\xrightarrow{\text{identify}}$ the generator of the observational SDE $\xrightarrow{\text{identify}}$ the laws of post-intervention processes".
>
> Please note that since we can provide examples with analytic solutions, we do not conduct simulation experiments where the results may be affected by numerical errors. In addition, explicitly presenting the entire calculation process affords readers a comprehensive understanding of how our proposed identifiability conditions facilitate reliable causal inference for SDEs.
>
> In Section 4, we have modified the title to "Simulations and examples", accompanied by an additional subsection titled "Illustrative instances of causal inference for linear SDEs". Within this subsection, we present a concise introduction to these two examples. Due to constraints related to space, the detailed calculation processes are presented in the Appendix.
>
> **Answers to questions:**
> >**Q:** What are the situations where the derived conditions for identifying linear SDEs so useful in practice?
>
> **A:** Thank you for your question. We would like to clarify that an important application of our derived identifiability conditions is to ensure reliable causal inferences for dynamic models governed by linear SDEs. Specifically, for linear SDEs under our proposed identifiability conditions, laws of the post-intervention processes are identifiable from the law of the observational process. In other words, our derived conditions ensures the identification of intervention effects on linear SDE systems using observational data. Note that in cases where the SDE system lacks identifiability, observational data would only suffice for predictive tasks, rendering it unsuitable for observational causal inference tasks. Thus, when dealing with real-world phenomena governed by linear SDEs, the application of our proposed identifiability conditions guarantees reliable causal inferences within the systesm. The example of gene expression in the yeast microorganism Saccharomyces Cerevisiae, as presented in our response to weakness 1, serves as a good practical situation for the use of our derived conditions.

---

> > ### Author Response · Authors · 2023-08-13
> >
> > Thank you for your valuable comments. Due to the restriction of a "6000 characters limit" per rebuttal, we encountered limitations in our ability to expound comprehensively upon weakness 2. We would like to draw your attention to the additional comments appended to the rebuttal addressed to Reviewer eRyW, where we have shown the two examples added for clarifying how our proposed identifiability conditions can ensure reliable causal inference for SDE models. Thank you for your understanding.
> >
> > Thank you again for your valuable comments. We believe that the quality of our work has been significantly improved thanks to your insightful comments.

---

> > ### Comment · Reviewer_GdBe · 2023-08-13
> > **Replay**
> >
> > Thanks for the authors' clarification.
> >
> > I appreciate the real-world application examples of linear SDEs. I recommend the authors to add some detailed examples as possible, which include yeast microorganism Saccharomyces Cerevisiae and asset pricing, to motivate this study. I raised my score.

---

> > > ### Author Response · Authors · 2023-08-14
> > > **Response to reviewer GdBe**
> > >
> > > Thank you for your acknowledgment and the increase in our score. We are genuinely appreciative of your support. We will include the practical examples in more detail to motivate our study. Once again, we extend our gratitude for your valuable response.

---

### Official Review · Reviewer_b8xq · 2023-07-26

**Soundness:** 2 fair
**Presentation:** 3 good
**Contribution:** 3 good
**Rating:** 4
**Confidence:** 3

**Summary:**

The present work considers the problem of identifying the generators of autonomous linear
stochastic differential equations (SDEs) with an additive or multiplicative
noise term from the law of its solution process (e.g., in case of additive noise
this is a Gaussian process) with a given fixed initial state. The authors derive
generic sufficient conditions for this and use the knowledge gained to expand
the model interpretability from a geometric point of view. The applicability of
the theoretical results is shown in the context of causal inference using linear
SDEs and empirical evidence is provided by a (small) set of simulations.

**Strengths:**

In recent years, modeling random dynamics had great impact into deep learning by
proposing so called Neural SDE (NSDE) models. Since SDEs are ubiquitously used
for modelling real world data, research interest in NSDEs rapidly grew.
However, we still know very little about the often hidden systematics of these
models and in particular the design of problem-specific function classes is a
challenge without a deeper understanding of it. Therefore, I appreciate this
work's attempt to make a systematic contribution to solving the mystery behind
random dynamic modeling.

Further, the content is well organized, with very few exceptions, has no
spelling or grammatical flaws and is original to the best of my knowledge. Empirical evidence is accessed through
experiments on toy data. Code examples are included in the supplementary
material.

**Weaknesses:**

It is not the value of the approach I am concerning here, but the shortcomings
related to the clarity of the submission and the technical soundness which I
will address with examples in the following:

- W1: Although the use of SDEs has rapidly gained prominence in the NeurIPS
  contributor landscape in recent years, I would argue it still represents a
  rather subtopic, not easily accessible to every contributor. On the other hand, the generator theory for SDEs is more of
  an advanced topic. Hence, in the context of its central role in this paper, it is
  desirable to add a (gentle) introduction to the topic for the wider NeurIPS
  audience. To illustrate this deficiency with an example: In Sec. In Section
  2.3 the authors talk about SDEs generators without giving a definition, and
  later in Section 2.1 the reader discovers that it is an operator on function
  spaces.

- W2: Starting from l. 184 including Lemma 3.2 and the interpretation of
  identifiability conditions from a geometric perspective, i.e., the generator
  of the SDE (1) is identifiable from $x_0$ when not all the given vectors
  are confined to an $A$-invariant proper subspace of $\mathbb{R}^d$, I found
  difficult to access; similar in the case of multiplicative noise (l. 269ff.).
  Can you phrase the idea in an easier manner? I think what I'm saying
  is that the amount of technical detail presented diminishes the value of the
  result.

- W3: In my opinion, the presentation of the causal interpretation of SDEs in
  Section. 2.3 as a subsection to "Model and Problem Formulation" is confusing
  since this topic is more an example of the application of generator
  identification of SDEs than a contribution to the problem definition of the
  latter in general.

- W4: Identifying the solution process of autonomous linear SDEs with additive
  noise term as in Eq. (1) by Gaussian processes is necessary for the results in
  the present work, e.g., cf. Lemma 3.1. This statement is justified by the
  identification of generators by properties of the law of such Gaussian
  solution processes with a given fixed initial state. I am wondering if the
  authors also assume the solution process to be Gaussian in case of linear SDEs with multiplicative noise as in Eq.
  (3) (because the method of solving for m(t) and P(t) is involved in the proof
  of Theorem 3.4 in Sec. A.7 in the appendix) because this proves wrong in the general case; e.g., in case of a geometric
  Brownian motion which is a solution to a linear SDE with multiplicative noise
  term.

- W5: Simulations are only presented in a very limited scenario on (very) low
  dimensional toy datasets. In real world applications, e.g., modeling (multivariate) time
  series data, the state variable dimension $d$ is often significantly larger
  than 10 (cf. I. Silva et al. [52]). In order to create added value for such models, e.g., NSDEs, the
  approach has to be tested in this extended framework.

However, I would like to reiterate my appreciation for the authors' interest in
expanding the field of modelling random dynamics. I'm sure that by working on
the deficiencies mentioned, the work will gain in value
and will deliver an enriching contribution to the community.

**Questions:**

- l. 34f. Check wording.
- l. 63 What type of solution?
- l. 73 Can you provide a reference here?
- l. 75 Can you provide a definition for the notion of a stochastic process
  here?
- l. 82 What do you mean by: "..., we recover the notion of stochastic
  process,..." ?
- l. 85 Do you mean: "..., $S$-valued functions on $[0,\infty)$?
- l. 124 "... can be identifiable from their ...", check wording.
- l. 129 What is that gap you are mentioning here?
- l. 142 Eq. 10: how are $b$ and $\sigma$ defined?
- l. 147 Prop. 2.1: "... denotes the space of **continuous** functions ..."
- l. 185 Can you provide reference for this statement?
- l. 195 Lemma 3.2: How is $Q$ defined?

**Limitations:**

Limitations are addressed (see e.g. Chapter 6) and mainly concern the practical
verification of the identifiable conditions. From my perspective there are important aspects that need to be
discussed additionally, e.g., computational limitations in the case of system
dimension $d$, and Brownian motion dimension, $m$, greater 2. Further, the
authors consider the class of linear SDEs with either additive or multiplicative
noise in the autonomous setting (i.e., drift and diffusion do not explicitly
depend on time); can the presented results be transferred to the non-autonomous case?

No evidence was found that the authors addressed potential negative societal impact. Please make the effort to include this mandatory information in your work.

---

> ### Author Rebuttal · Authors · 2023-08-10
>
> Thank you for your helpful comments.  We address your comments point by point as follows. We have also updated our paper according to your comments, and we believe that the quality has been significantly improved thanks to your insightful comments.
>
> **Answers to weaknesses:**
>
> >**W1:** introduce generator
>
> **A:**  Thank you for bringing this matter to our attention. We have added an introduction to the generator of a stochastic process to our manuscript. Reads as follows: "The generator of a stochastic process $X_t$ can be defined as $(\mathcal{L}f)(x)= \lim_{s\rightarrow 0}\cfrac{\mathbb{E}[f(X_{t+s})-f(X_t)|X_t= x]}{s}$, where $f$ is a suitably regular function."
>
> >**W2:** geometric interpretation
>
> **A:** Thank you for your comment. In order to elucidate the geometric explanation of the condition, we would like to draw your attention to the two fundamental concepts (refer to lines 203-204) "$A$-invariant subspace" and "proper subspace".  In other words, an $A$-invariant proper subspace of $\mathbb{R}^d$ can be described as a non-empty subspace of the vector space $\mathbb{R}^d$ that remains unchanged under the transformation induced by matrix $A$ and is not the entire space itself.
>
> Moreover, it is crucial to recognize that this portion primarily constitutes an expression of the fundamental Jordan decomposition of matrix $A$.  We have made necessary adjustments to make the manuscript more reader-friendly.
>
> >**W3:** presentation of Section 2
>
> **A:**  We have made necessary adjustments to enhance the presentation of Section 2. Moreover, we have made necessary adjustments to various sentences and thoughtfully incorporated gentle introductions and relevant citations to elucidate the concepts presented throughout the paper, thereby ensuring a more coherent and reader-friendly presentation.
>
> >**W4:** Regarding the proof of Theorem 3.4
>
> **A:** Thank you for this comment. We would like to clarify that for linear SDEs with multiplicative noise, as depicted in Eq. (3), we did not assume the solution process to be Gaussian.
>
> >**W5:** simulation dimensions $d$ and NSDEs.
>
> **A:** Thank you for this comment. We would like to clarify that our proposed sufficient conditions for both linear SDEs with additive noise and linear SDEs with multiplicative noise are applicable to any dimension $d\geqslant2$.
>
> This work primarily focuses on establishing the foundational theory for the identifiability analysis of linear SDEs.  The simulations presented in this study are intended to validate the proposed theoretical results.  As for NSDEs, given their neural network-based structure and non-parametric nature, they differ from our model settings, and no identifiability guarantees currently exist. Consequently, NSDEs are predominantly employed for forecasting tasks. However, for SDEs satisfying our proposed identifiability conditions, not only perform well in forecasting tasks but also enable reliable predictions about the SDE system's behavior under **intervention**.
>
> **Answers to questions:**
> >1. **Q:** l. 34 wording.
>
> **A:** We have updated the sentence, reads as follows: "Traditional identifiability analysis of dynamic systems focuses on deriving conditions under which a unique set of parameters can be obtained from error-free observational data."
>
> >2. **Q:** l. 63 type of solution
>
> **A:** In the context of our study, the solution to the linear SDEs is the strong solution.
>
> >3. **Q:** l. 73 reference
>
> **A:** We have provided an inference [25].
>
> >4. **Q:** l. 75 notion of a stochastic process
>
> **A:** We have included the notion.
>
> >5. **Q:** l. 82 "$\ldots$, we recover the notion of stochastic process,$\ldots$"
>
> **A:** We would like to clarify that the mentioned sentence means that we obtain the definition of the law of a continuous stochastic process. We have updated our manuscript to the wording.
>
> >6. **Q:** l. 85 $S$-valued functions
>
> **A:** Here when $X$ is a continuous stochastic process, $S = C[0, \infty]$, in other words, one can say that a stochastic process is an $S$-valued random variable, where $S = C[0, \infty]$ stands for the space of all continuous, real-valued functions on $[0, \infty]$. But we cannot say $S$-valued functions.
>
> >7. **Q:** l. 124 wording
>
> **A:** We have updated the sentence
>
> >8. **Q:** l. 129  gap
>
> **A:** In the context of SDEs, the primary goal of causal inference is to uncover conditions under which: "the law of the observational process $\xrightarrow{\text{identify}}$ the generator of the observational SDE $\xrightarrow{\text{identify}}$ the laws of post-intervention processes". The latter aspect has been established in reference [16], while the former part represents the gap that we mention here and address in this paper.
>
> >9. **Q:** l. 142 Eq. 10: $b$ and $\sigma$
>
> **A:**  $b$ and $\sigma$ are locally Lipschitz continuous in the space variable $x$.
>
> >10. **Q:** l. 147 **continuous**
>
> **A:** We have updated our manuscript accordingly.
>
> >11. **Q:** l. 185 reference
>
> **A:**  We have added a reference ([41]).
>
> >12. **Q:** l. 195 $Q$
>
> **A:** In Lemma 3.2, $Q$ is defined in lines 189-193, such that matrix $A$ can be expressed as $A = Q\Lambda Q^{-1}$ through Jordan decomposition.
>
> >**Answers to limitations:**
>
> Thank you for this comment. We would like to reiterate that the goal of this work is to establish the foundational theory for the identifiability analysis of linear SDEs, rather than developing a specific parameter estimation method or algorithm for SDEs. Our proposed theories are applicable to any $d\geqslant 2$ and $m\geqslant 2$. Therefore, we maintain the stance that it is unnecessary to introduce this computational limitation, which are primarily of concern when proposing a new estimation algorithm.
>
> Regrettably, our proposed method cannot be readily extended to the non-autonomous case. Accordingly, we have updated our manuscript to reflect this constraint.
>
> Regarding the potential negative social impact, we have added relevant information.

---

> > ### Author Response · Authors · 2023-08-12
> > **Necessary elucidations of rebuttal addressed to Reviewer b8xq**
> >
> > Thank you for your valuable comments. Due to the restriction of a "6000 characters limit" per rebuttal, we encountered limitations in our ability to expound comprehensively upon certain aspects of our responses. Thus, we add essential elucidations within this comment. Thank you for your understanding.
> >
> > >Regarding W2 geometric interpretation
> >
> > We would like to give an example. For illustrative purposes, let us assume matrix $A$ has $d$ distinct real eigenvalues, allowing us to express the Jordan decomposition of $A$ as $A = Q \Lambda Q^{-1}$, Where $\Lambda$ represents a diagonal matrix with each entry corresponding to an eigenvalue of matrix $A$, denoted as $\lambda_k$. Each column of $Q$, denoted as $Q_k$, corresponds to the respective eigenvector of $\lambda_k$. Suppose we have a subspace $L_{-d}$ of $\mathbb{R}^d$, which is constructed as a linear span of the first $(d-1)$ columns of $Q$ (i.e., the first $(d-1)$ eigenvectors of $A$). In such a case, we classify $L_{-d}$ as an $A$-invariant **proper** subspace of $\mathbb{R}^d$.
> >
> > >Regarding W3 presentation of Section 2
> >
> > We would like to summarize the necessary adjustments we have made to Section 2 and Section 3 in the following:
> > We have changed the title of Section 2 to "Background Knowledge", which now only consists of two subsections. In Subsection 2.1, now titled "Background knowledge of linear SDEs," we have included the following components:
> > - The solutions of the two linear SDEs of interest (see lines 64, 67-78).
> > - Definition 2.1, which pertains to the law of a continuous stochastic process, along with the subsequent paragraph.
> > - A gentle introduction to the generator of a stochastic process.
> > -  A presentation of the general form (Eq. (10)) shared by both of our interested linear SDEs, whose solution is an It\^o diffusion process. Additionally, we have provided the explicit expression of the generator of an It\^o diffusion process (see lines 141-148).
> >
> > In Subsection 2.2, we have retained the original subsection titled "Causal Interpretation of SDEs" (formerly subsection 2.3).
> >
> > We have introduced a new subsection, 3.1, under "Main results" (Section 3), titled "Prerequisites." Within this subsection, we have included the following elements:
> > - Lemma 2.2 from the original version of the manuscript (see lines 133-141).
> > - Proposition 2.2 from the original version of the manuscript (see lines 151-157).
> >
> > >Regarding W4 the proof of Theorem 3.4
> >
> > We would like to clarify again that for linear SDEs with multiplicative noise, as depicted in Eq. (3), we did not assume the solution process to be Gaussian. To elucidate the rationale behind incorporating the first moment $m(t) = \mathbb{E}[X_t]$ and the second moment $P(t) = \mathbb{E}[X_t X_t^{\top}]$ in the proof of Theorem 3.4, we intend to provide a concise overview of the proof.
> >
> > 1. Let $\tilde{A}, \tilde{G}\_k \in \mathbb{R}^{d\times d}$ for all $k=1, \ldots, m$, such that $X(\cdot; x_0, A, \\{G_k\\}\_{k=1}^m) \overset{\text{law}}{=}X(\cdot; x_0, \tilde{A}, \\{\tilde{G}\_k\\}\_{k=1}^m)$,
> > we will show that under our identifiability condition, for all $x\in \mathbb{R}^d$, $(A, \sum_{k=1}^mG_kxx^{\top}G_k^{\top}) = (\tilde{A}, \sum_{k=1}^m\tilde{G}_kxx^{\top}\tilde{G}_k^{\top})$.
> >
> >     For simplicity of notation, in the following, we denote $A_1:= A$, $A_2:= \tilde{A}$, $G_{1,k}:= G_k$ and $G_{2,k}:= \tilde{G}\_k$.
> >
> > 2. Utilizing the fact that $m_1(t) = m_2(t)$ for all $0 \leqslant t < \infty$, we show that, under condition A1 stated in Theorem 3.4, $A_1 = A_2$, confirming the identifiability of matrix $A$.
> >
> > 3. Utilizing the fact that $P_1(t) = P_2(t)$ for all $0 \leqslant t < \infty$ and relying on the premise $A_1 = A_2$, we derive the result: $\sum_{k=1}^m G_{1,k} P(t) G_{1,k}^{\top} = \sum_{k=1}^m G_{2,k} P(t) G_{2,k}^{\top},\  \forall 0 \leqslant t < \infty$, where $P(t) = P_1(t) = P_2(t)$.
> >
> > 4. Recall our objective to establish $\sum_{k=1}^m G_{1,k} xx^{\top} G_{1,k}^{\top} = \sum_{k=1}^m G_{2,k} xx^{\top} G_{2,k}^{\top}$ for all $x\in \mathbb{R}^d$. Notably, both $xx^{\top}$ and $P(t)$ are $d\times d$ symmetric matrices. It follows that, when $P(t)$ with $0 \leqslant t < \infty$ cover all instances of $xx^{\top}$ for each $x\in \mathbb{R}^d$, the theorem is proved. Some calculation shows that condition A2 stated in Theorem 3.4 fulfills this requirement.
> >
> >   We hope this brief exposition can enhance your comprehension of the proof and, at the very least, offer you a rudimentary insight into its underlying principles.
> >
> > >Regarding questions 9 and 12
> >
> > We have made necessary adjustments to our manuscript to add notions of $b$ and $\sigma$, and have made $Q$ clear in Lemma 3.2.
> >
> > Thank you again for your valuable comments. We believe that the quality of our work has been significantly improved thanks to your insightful comments.

---

> > ### Comment · Reviewer_b8xq · 2023-08-15
> >
> > I thank the authors very much for their response and for the detailed discussion
> > of my questions and concerns.
> >
> > Regarding the authors comment: "Therefore, we maintain the stance that it
> >     is unnecessary to introduce this computational limitation, which are
> >     primarily of concern when proposing a new estimation algorithm."
> >     I disagree, I admit that purely theoretical questions should be answered
> >     without the primarily intend to concern computational limitations. But for
> >     completeness, if submitted works include simulations (as it is the case for this work) I expect some notes on computational limits; and don't get me
> >     wrong I am not talking about an extensive analysis here.
> >
> > My overall score relates mainly to the point, that this work needs a major revision. E.g., the given 6000 characters for authors' rebuttal do not suffice to clarify all raised weaknesses/questions/limitations.
> >
> > You have definitely clarified things, and I cherish the effort and your time. But I am sorry to say, that I stay with my score.
> >
> > Thank you again.

---

> > > ### Author Response · Authors · 2023-08-17
> > > **Response to Reviewer b8xq**
> > >
> > > Thank you for your comment.
> > >
> > > Regarding the computational limitations associated with the estimation method, as per your suggestion for enhancing the completeness of our paper, we have added pertinent explanations in our manuscript. The added content is as follows: “It is worth noting that in our simulations, we employed the MLE method to estimate the system parameters.  This necessitates the calculation of the transition pdf from one state to the successive state at each discrete temporal increment. Consequently, as the state dimension $d$ and Brownian motion dimension $m$ increase, the computational time is inevitably significantly increased, rendering the process quite time-consuming. To expedite parameter estimation for scenarios involving high dimensions, alternative estimation approaches are required. The development of a more efficient parameter estimation approach remains an important task in the realm of SDEs, representing a promising direction for our future research.”
> > >
> > > We understand that the 6000 characters constraint for our rebuttal hinders our clarification of your questions. As you may have known, regretfully, at this stage we are not able to upload our new revision. Should you possess any specific concerns, please feel free to communicate them with us. We are committed to addressing your concerns comprehensively within our response.
> > >
> > > Regarding the weakness of W4, we kindly request your assessment of our response to this matter. We seek your response on whether any further concerns persist regarding this aspect. The reason we make this request is because of the concern raised by Reviewer 5yfh, who would like to know the validation of our statements concerning Theorem 3.4 in response to weakness W4. We eagerly anticipate your feedback on this matter.
> > >
> > > Once again, we extend our gratitude for your valuable comments. We believe that the quality of our paper has been significantly improved thanks to your insightful comments.

---

> > > > ### Comment · Reviewer_b8xq · 2023-08-18
> > > >
> > > > Dear authors,
> > > >
> > > > thank you for reaching out again and addressing my concern. Regarding
> > > > computational limitations, this is pretty close to what I had in mind.
> > > >
> > > > Regarding your comment on the character limit, this is a conference review
> > > > process, which differs from, for example, a journal submission in the sense that
> > > > submissions that require major revision will be rejected. That is, as far as I
> > > > know, NeurIPS, ICML, ICLR, etc. do not intend to provide a more
> > > > comprehensive review style, I guess to keep the review manageable given the
> > > > large volume of submissions. I repeat, the 6000 character count should be enough
> > > > to address any concerns raised, otherwise I'll call it a major revision that
> > > > definitely needs revisiting.
> > > >
> > > > (ad Question 6): That is correct! What I had in mind is the following concern:
> > > > In Definition 2.1 your are introducing a random variable $X_t$ with domain $S$.
> > > > Subsequently you define a continuous stochastic process $X$ as family of such
> > > > $S$-valued random variables $X_t$. Then given a path $X(\omega)$ I assume this is
> > > > a continuous function $[0,\infty) \to S$. BUT now you say, $X(\omega) \in
> > > > \mathcal{C}[0,\infty)$, hence $S = \mathbb{R}^n$. Then why using $\mathbb{R}^n$
> > > > at the first place. I guess what you intended is (but which confused me while reading), to introduce a continuous
> > > > stochastic process as function valued random variable but then introducing $X_t$
> > > > is missing and the special structure of $\mathcal{F}$ in case of a stochastic
> > > > process is swept under the rug, in my opinion, since we need filtrations here.
> > > >
> > > > (ad Question 12): I am aware of, that you have (more or less) on the fly
> > > > introduced $Q$ in the previous text. Nevertheless, good practice (in my opinion)
> > > > involves to introduce $Q$ properly in Lemma 3.2 again.
> > > >
> > > > (ad Weakness 4)
> > > >
> > > > Here, my concern addresses the following point:
> > > >
> > > > > Let $\tilde{A}, \tilde{G}_k \in \mathbb{R}^{d \times d}$ for all $k =
> > > > > 1,\dots,m$, such that $X(\cdot; x_0, A, \{G_k\}_k=1^m)
> > > > > \stackrel{\text{law}}{=} X(\cdot; x_0, \tilde{A}, \{\tilde{G}_k\}_k=1^m)$,
> > > > > ....
> > > >
> > > > This is how proof of Theorem 3.4 starts.
> > > >
> > > > Taking a step back; Theorem 3.4 gives sufficient conditions on the identifiability of the generator of a linear SDE with multiplicative noise.
> > > >
> > > >  Preliminary and dealing with the
> > > > arguably simpler case of a linear SDE with additive noise, Theorem 3.3 gives
> > > > sufficient conditions on the identifiability of a generator in this case.
> > > > Identifiability in both cases
> > > > involves working with equality of stochastic processes in law. For linear stochastic
> > > > processes with additive noise, Lemma 3.1 provides necessary and sufficient
> > > > conditions that reduce equality in law, to equality of the first and second
> > > > moment functions of the involved processes. A look into the proof of Lemma 3.3
> > > > reveals, that the authors build on the fact, that every solution process to a
> > > > SDE (1) is a Gaussian process, and the law of the latter can be fully determined
> > > > by its first and second moment functions. Going further by analyzing the proof
> > > > of Theorem 3.3 reveals (cf. l. 640) the authors apply Lemma 3.1. Overall Gaussianity
> > > > is assumed implicitly in this case.
> > > >
> > > > Returning to the case of a linear SDE with multiplicative noise, i.e., SDEs of
> > > > type (3)
> > > > in combination with Theorem 3.4. Now see l. 747; why does $P_1(t) = P_2(t)$
> > > > follow by $X¹ \stackrel{\text{law}}{=} X²$. In the linear case we had to apply
> > > > Lemma 3.1 for something similar in case of SDEs of type (1). But even if I'm
> > > > slow here and you prove me wrong (my apologies in that case), I'm still
> > > > wondering how much Definition 3.2 really reduces the class of type (3) SDEs,
> > > > since we're talking about equality in law based on the first and
> > > > second moment functions of the solution process.
> > > >
> > > > I really hope that I was able to provide some clarity with these additional comments.

---

> > > > > ### Author Response · Authors · 2023-08-18
> > > > > **Response to Reviewer b8xq**
> > > > >
> > > > > Thank you for your insightful comments. We address the added concerns one by one in the following.
> > > > >
> > > > > >(Regarding ad Question 6)
> > > > >
> > > > > As stated in our rebuttal to Question 6, we say that when $X$ is a continuous stochastic process, $S = C[0,\infty]$ instead of $S= \mathbb{R}^n$, I guess "$\mathbb{R}^n$" is a typo in your comment? This definition is actually adapted from the definition in a classic stochastic calculus book [21] (with only minimal adjustments introduced in our definition). To increase readibility, we have updated the lines 82-85 as "When $X := \\{X_t;  0 \leqslant t < \infty\\}$ is a continuous stochastic process on $(\Omega, \mathcal{F}, \mathbb{P})$, and $S=C[0,\infty)$, such an $X$ can be regarded as a random variable on $(\Omega, \mathcal{F}, \mathbb{P})$ with values in $(C[0,\infty), \mathcal{B}(C[0,\infty)))$, and $P^X$ is called the law of $X$. Here $C[0,\infty)$ stands for the space of all continuous, real-valued functions on $[0, \infty]$." We hope this new version can dispel your confusion.
> > > > >
> > > > >
> > > > > >(Regarding ad Question 12)
> > > > >
> > > > > Thanks for pointing this out. As we stated in the "Necessary elucidations of rebuttal addressed to Reviewer b8xq", we have made Q clear in Lemma 3.2.
> > > > >
> > > > > >(Regarding ad Weakness 4)
> > > > >
> > > > > Firstly, it is important to know that for two stochastic processes $X^1$ and $X^2$, once $X^1 \overset{\text{law}}{=} X^2$, then $E[X_t^1]= E[X_t^2]$ and $P_1(t) = E[X_t^1 (X_t^1)^{\top}] = E[X_t^2 (X_t^2)^{\top}] = P_2(t)$ for all $0\leqslant t < \infty$, regardless of the associated SDE of the process. Actually, once $X^1 \overset{\text{law}}{=} X^2$, $E[f(X_t^1)]=E[f(X_t^2)]$ for any $0\leqslant t < \infty$ and $f \in C^{\infty}(\mathbb{R}^d)$. Intuitively, for two random variables, if their distributions are the same, then the first and second moment of these two random variables are the same.
> > > > >
> > > > > If you check the proof of Lemma3.1 (lines 574-577), you can see that the equality of mean and covariance are directly derived from the fact of the equality of law. However, the key of Lemma3.1 is the backward direction (lines 578-581), that is for a Gaussian process, the equality of mean and covariance can also determine the equality of law.
> > > > >
> > > > > The reason why when derive the identifiability condition of SDE (1) we apply Lemma3.1 is because, actually, we have proved that when $A$ has distinct eigenvalues, the proposed identifiability condition in Theorem 3.3 is not only sufficient but also **necessary** (proof of necessity: lines 655-708). Therefore, only when the process is Gaussian process, we can derive the necessity of the condition through Lemma3.1 (using the backword direction of Lemma3.1). That is why we apply Lemma3.1 in the proof of Theorem 3.3. While for the SDE (3), since the solution process is not a Gaussian process, we do not have similar properties as stated in Lemma 3.1, thus we can only derive a sufficient condition.
> > > > >
> > > > > The reason why we use $E[X_t]$ and $P(X_t)$ in the proof of Theorem 3.4 have been explained in the “Necessary elucidations of rebuttal addressed to Reviewer b8xq”, where we provided a concise overview of the proof.
> > > > >
> > > > > Definition 3.2 does not reduce the class of type (3) SDEs, since here our objective is to uncover conditions when the generator is identifiable from the law of the solution process of the SDE (3). In other words, when the laws of the solution processes of two type (3) SDEs are the same, we derived under what conditions the generators associated with these two type (3) SDEs are the same (i.e., identifiable).
> > > > >
> > > > > Once again, thank you for your valuable comments. Should you possess any more concerns, please feel free to communicate them with us.

---

> > > > > > ### Comment · Reviewer_b8xq · 2023-08-18
> > > > > >
> > > > > > Thank you for your clarification. That definitelly helped to better understand your work!
> > > > > >
> > > > > > (Weakness 4)
> > > > > >
> > > > > > - Did you include the necessary statement into Theorem 3.3? Its part of the proof but you will need $A$ to have distinct eigenvalues which is part of a subsequent comment, only.
> > > > > > - Section A.5 l. 640 is proof part _Sufficiency_ and includes Lemma 3.1. Then it should be deleted at this point.
> > > > > >
> > > > > > Thank you again.

---

> > > > > > > ### Author Response · Authors · 2023-08-18
> > > > > > > **Response to Reviewer b8xq**
> > > > > > >
> > > > > > > Thank you for your valuable comments.
> > > > > > >
> > > > > > > >(Weakness 4)
> > > > > > > >
> > > > > > > We did only present the necessary statement in the proof in the Appendix A.5 in our original manuscript. To explicitly show this in the Theorem 3.3, we have updated our manuscript (lines 216-219), reads as
> > > > > > >
> > > > > > > **"** **Themrem 3.3.** Let $x_0 \in \mathbb{R}^d$ be fixed. **Assuming that the matrix $A$ in the SDE (1) has $d$ distinct eigenvalues.** The generator of the SDE (1) is identifiable from $x_0$ if **and only if**
> > > > > > > \begin{equation}
> > > > > > >     rank([x_0|A x_0|\ldots|A^{d-1}x_0|H_{\cdot 1}|AH_{\cdot 1}|\ldots|A^{d-1}H_{\cdot 1}|\ldots|H_{\cdot d}|AH_{\cdot d}|\ldots|A^{d-1}H_{\cdot d}])=d\,,
> > > > > > > \end{equation}
> > > > > > > where $H:=GG^T$, and $H_{\cdot j}$ stands for the $j$-th column vector of matrix $H$, for all $j=1,\cdots, d$.
> > > > > > >
> > > > > > >
> > > > > > > The proof of Theorem 3.3 can be found in Appendix A.5. The condition in Theorem 3.3 is both sufficient and necessary when the matrix $A$ has distinct eigenvalues. It is worth noting that almost every $A \in \mathbb{R}^{d\times d}$ has $d$ distinct eigenvalues concerning the Lebesgue measure on $\mathbb{R}^{d\times d}$ [41]. Hence, this condition is both sufficient and necessary for almost every $A$ in $\mathbb{R}^{d\times d}$. However, in cases where $A$ has repetitive eigenvalues, this condition is solely sufficient and not necessary.**"**
> > > > > > >
> > > > > > > Thanks for pointing this out, the equality of this part has been significantly improved thanks to your insightful comments.
> > > > > > >
> > > > > > > Regarding the mentioned A.5 l.640, it is noteworthy that in this context, our focus shifts to the derivation of the covariance matrix, instead of the second moment of the state variable. As a result of this distinction, the process is rendered somewhat less straightforward (see lines 576-577 in the proof of Lemma 3.1). While, conventionally, the equivalence of law readily implies the equivalence of the second moment, such a direct inference is not immediately apparent in this scenario. We hold a preference for the retention of the specific sentence in l.640 ("by Lemma 3.1"). Your understanding in this matter is greatly appreciated.
> > > > > > >
> > > > > > > Once again, we extend our gratitude for your insightful and valuable comments. The quality of our work has been significantly improved thanks to your insightful comments. Should you possess any more concerns, please feel free to communicate them with us.

---

> > > > > > > ### Author Response · Authors · 2023-08-20
> > > > > > > **Response to Reviewer b8xq**
> > > > > > >
> > > > > > > Dear Reviewer b8xq,
> > > > > > >
> > > > > > > We thank you again for your time dedicated to reviewing our paper. The discussion period will end soon, we will appreciate your feedback, and if you have further questions or concerns, we hope we will have the opportunity to address them. Specifically, with regard to Weakness 4, we are keen to understand if any further concerns persist in this regard. We eagerly anticipate your feedback on this matter.
> > > > > > >
> > > > > > > Once again, we extend our gratitude for your valuable comments.
> > > > > > >
> > > > > > > Best regards, authors of paper 5668

---

> > > > > > > > ### Comment · Reviewer_b8xq · 2023-08-20
> > > > > > > >
> > > > > > > > Dear authors, thank you for all of your continued efforts to address concerns and questions related to my review. Nonetheless, I keep my score!
> > > > > > > > Because "...the goal of this work is to establish the foundational theory for the identifiability analysis of linear SDEs...", however, in sum a greater amount of weaknesses in the clarity of presentation were identified and require further examination after revision; according to the concept: trust is good, control is better.
> > > > > > > >
> > > > > > > > Sincerely
> > > > > > > > b8xq

---

> > > > > > > > > ### Author Response · Authors · 2023-08-20
> > > > > > > > > **Response to Reviewer b8xq**
> > > > > > > > >
> > > > > > > > > Dear Reviewer b8xq,
> > > > > > > > >
> > > > > > > > > Thank you for your feedback. We fully understand and respect your final decision. We wish to  extend our sincere gratitude for your valuable feedback and the time you dedicated to the thorough review and discussion of our paper.
> > > > > > > > >
> > > > > > > > > Thank you once again.
> > > > > > > > >
> > > > > > > > > Sincerely, authors of paper 5668

---

### Official Review · Reviewer_eRyW · 2023-07-26

**Soundness:** 3 good
**Presentation:** 3 good
**Contribution:** 3 good
**Rating:** 7
**Confidence:** 3

**Summary:**

As I am not an expert in causal inference nor in SDEs, I will start with summarising the goal of the paper how I understood by reading it and some references therein. This summary will certainly reveal my ignorance regarding this research area, but I prefer to give this transparency and hope my review will be weighted accordingly.
Ordinary and stochastic differential equations (SDEs) can be interpreted as modelling the influence among elements of a dynamical system. Therefore, they offer a tool for causal inference. For example, when making an intervention to a dynamical system, this intervention can be defined by a change to the corresponding SDE, resulting in “post-intervention” SDE. Previously, it was shown that if we can identify the generator of the original SDE, then we can use this result to obtain the distribution of the post-intervention SDE.

This paper derives sufficient conditions for the identifiability of the generator of linear SDEs with both additive and multiplicative noise. The authors conduct simulation experiments to assess the validity of their theoretical results.

**Strengths:**

The paper is generally well-written and clearly structured and, although technical, well readable.

The theoretical results which seem to be the major contribution of the paper are tested in an intuitive simulation study. The simulation experiments and results are described clearly and concisely.

**Weaknesses:**

The beginning of the introduction is good, but in the following paragraphs, the motivation for the importance of identifiability in SDE from a causal inference point of view is not well-explained.
I had to refer to [16] to find a clear explanation of how SDE can be used for causal inference.

SDE (1) is referenced in the introduction but appears only later in section 2.1.

The simulation experiment nicely illustrate the derived identifiability conditions. However, an example of how this can used for causal inference is not shown. Given that performing causal inference with SDE is one of the central motivations of this work, I think it would be essential to provide a concrete example of how this can be achieved using the contributions of this paper.

**Questions:**

You are using MLE to estimate the system parameters (let’s call them $\theta$) given discrete observational data $x$, i.e., you are obtaining a point for $\theta$. Given there is uncertainty in the system and due to limited amount of observational data, there should be uncertainty w.r.t. to the estimate of $\theta$ as well. Have you considered performing fully Bayesian inference over $\theta$ instead, i.e., obtaining $p(\theta | x)$ instead of performing $\hat{\theta}$ = MLE?
For example, you could use simulation-based inference (SBI) to obtain a approximate posterior from simulated data.

This would enable you add the uncertainty estimates to your analysis of the different “Identifiable” and “unidentifiable” cases reported in section 4.

**Limitations:**

The authors discuss the limitations of their work.

---

> ### Author Rebuttal · Authors · 2023-08-10
>
> Thank you for your helpful comments.  We address your comments point by point as follows. We have also updated our paper according to your comments, and we believe that the quality has been significantly improved thanks to your insightful comments.
>
> **Answers to weaknesses:**
> >1. **W:** The beginning of the introduction is good, but in the following paragraphs, the motivation for the importance of identifiability in SDE from a causal inference point of view is not well-explained. I had to refer to [16] to find a clear explanation of how SDE can be used for causal inference.
>
>  **A:** Thank you for bringing this matter to our attention. In order to enhance the clarity of the connection between the identifiability of the SDE's generator and causal inference, we have made the necessary adjustments to lines 41-46 of the manuscript. The updated version now reads as follows: "Secondly, the identifiability of an SDE's generator suffices for reliable causal inferences for this system. Note that, in the context of SDEs, the main task of causal analysis is to identify laws of the post-intervention processes from the law of the observational process. As proposed in [16], for an SDE satisfying specific criteria, laws of the post-intervention processes are identifiable from the generator of this SDE. Consequently, the intricate task of unraveling causality can be decomposed into two constituent components through the generator. This paper aims to uncover conditions under which the generator of a linear SDE attains identifiability from the law of the observational process. By establishing these identifiability conditions, we can effectively address the causality task for linear SDEs." The detailed interpretation of this connection, involving complex mathematical formulations and explanations, is presented in Section 2. Specifically, we elucidate the way of making an intervention on an SDE model and the arising SDE after intervention (i.e., the post-intervention SDE) in Definition 2.2 and the subsequent paragraph.
>
> >2. **W:** SDE (1) is referenced in the introduction but appears only later in section 2.1.
>
> **A:** Thank you for pointing this out. We have made the necessary adjustments to our manuscript to get rid of the forward references. Specifically, we have relocated the concise formulations of the two linear SDEs (i.e., SDE (1) and SDE (3)) to the appropriate position of the Introduction section.
>
> >3. **W:** The simulation experiment nicely illustrate the derived identifiability conditions. However, an example of how this can used for causal inference is not shown. Given that performing causal inference with SDE is one of the central motivations of this work, I think it would be essential to provide a concrete example of how this can be achieved using the contributions of this paper.
>
> **A:** Thank you for bringing this matter to our attention. We have updated our manuscript and given two examples for both SDE (1) and SDE (3), respectively, to show how our proposed identifiability conditions can ensure reliable causal inference for SDE models. Specifically, we have shown how under our proposed identifiability conditions, "the law of the observational process $\xrightarrow{\text{identify}}$ the generator of the observational SDE $\xrightarrow{\text{identify}}$ the laws of post-intervention processes".
> Please note that since we can provide examples with analytic solutions, we do not conduct simulation experiments where the results may be affected by numerical errors. In addition, explicitly presenting the entire calculation process affords readers a comprehensive understanding of how our proposed identifiability conditions facilitate reliable causal inference for SDEs.
> In Section 4, we have modified the title to "Simulations and examples", accompanied by an additional subsection titled "Illustrative instances of causal inference for linear SDEs". Within this subsection, we present a concise introduction to these two examples. Due to constraints related to space, the detailed calculation processes are presented in the Appendix.
>
> **Answers to questions:**
>
> Thank you for your question. As you know that the simulations we used in Section 4 were intended to validate our proposed theoretical results (i.e., the identifiability conditions established in Section 3) rather than propose a new estimation method. Our objective was to demonstrate that, when the proposed identifiability conditions are satisfied, the estimated parameters will converge to the true underlying parameters as the number of observations goes to infinity. For parameter estimation, we opted for the MLE method, as it is widely used for parameter estimation for SDEs and straightforward to implement. In addition, we find that the results obtained from using the MLE method well supported the validity of our theoretical findings.
>
> We truly appreciate your suggestion regarding the Bayesian inference method. In our future work, we plan to explore Bayesian inference for estimating system parameters in a finite sample setting. By doing so, we expect to obtain valuable uncertainty estimates for the estimated parameters, thus further enriching our analysis.

---

> > ### Author Response · Authors · 2023-08-12
> > **Necessary elucidations of rebuttal addressed to Reviewer eRyW**
> >
> > Thank you for your valuable comments. Due to the restriction of a "6000 characters limit" per rebuttal, we encountered limitations in our ability to expound comprehensively upon weakness 3. Thus, we add essential elucidations within this comment. Specifically, we show the two added examples for clarifying how our proposed identifiability conditions can ensure reliable causal inference for SDE models. Thank you for your understanding.
> >
> > 1. For SDE (1):
> >     Recall that SDE (1) is defined as
> >     \begin{equation*}
> >         dX_t = AX_tdt + GdW_t, \ \ \  X_0 = x_0,
> >     \end{equation*}
> >     where $0\leqslant t < \infty$, $A\in \mathbb{R}^{d\times d}$ and $G\in \mathbb{R}^{d\times m}$ are constant matrices, $W$ is an $m$-dimensional standard Brownian motion. Let $X(t; x_0, A, G)$ denote the solution process of SDE (1). Let $\tilde{A} \in \mathbb{R}^{d\times d}$ and $\tilde{G} \in \mathbb{R}^{d\times m}$ define the following SDE:
> >
> >     \begin{equation*}
> >         d\tilde{X}\_t = \tilde{A}\tilde{X}\_tdt + \tilde{G}dW\_t, \ \ \  \tilde{X}\_0 = x_0,
> >     \end{equation*}
> >     such that  $X(\cdot; x_0, A, G) \overset{\text{law}}{=} \tilde{X}(\cdot; x_0, \tilde{A}, \tilde{G})$. Then under our proposed identifiability condition stated in Theorem 3.3, we have shown that the generator of SDE (1) is identifiable, i.e., $(A, GG^{\top}) = (\tilde{A}, \tilde{G}\tilde{G}^{\top})$. Till now, we have shown that under our proposed identifiability conditions, "the law of the observational process  $\xrightarrow{\text{identity}}$ the generator of the observational SDE".
> >     Then we will show that the law of the post-intervention process is also identifiable. For notational simplicity, we consider intervention on the first coordinate, making the intervention $X_t^1 = \xi$ and $\tilde{X}\_t^1 = \xi$ for $0\leqslant t < \infty$. It will suffice to show equality of the laws of the non-intervened coordinates (i.e., $X_t^{(-1)}$ and $\tilde{X}\_t^{(-1)}$, note the superscripts do not denote reciprocals, but denote the $(d-1)$-coordinates without the first coordinate). Express the matrices of $A$ and $G$ in blocks
> >     \begin{equation*}
> >         A = \begin{bmatrix}
> >             A_{11} & A_{12}\\\\
> >             A_{21} & A_{22}
> >         \end{bmatrix},
> >     G =
> >     \begin{bmatrix}
> >         G_1 \\\\
> >         G_2
> >     \end{bmatrix},
> >     \end{equation*}
> >     where $A_{11} \in \mathbb{R}^{1\times 1}$, $A_{22} \in \mathbb{R}^{(d-1)\times (d-1)}$, $G_1 \in \mathbb{R}^{1\times m}$ and $G_2 \in \mathbb{R}^{(d-1) \times m}$. Also, consider corresponding expressions of matrices $\tilde{A}$ and $\tilde{G}$.
> >     By making intervention $X_t^1 = \xi$, one obtains the post-intervention process of the first SDE satisfies:
> >     \begin{equation*}
> >         dX_t^{(-1)} = (A_{21}\xi + A_{22}X_t^{(-1)})dt + G_2dW_t, X_0^{(-1)} = x_0^{(-1)}
> >     \end{equation*}
> >     which is a multivariate Ornstein-Uhlenbeck process, according to [Corollary 1, 1], this process is a Gaussian process, assuming $A_{22}$ is invertible, then the mean vector can be described as
> >     \begin{equation*}
> >         E[X_t^{(-1)}] = e^{A_{22}t}x_0^{(-1)} - (I - e^{A_{22}t})A_{22}^{-1}A_{21}\xi,
> >     \end{equation*}
> >     and based on [Theorem 2, 1], the cross-covariance can be described as
> >     \begin{equation*}
> >     \begin{split}
> >         V(X_{t+h}^{(-1)},X_t^{(-1)}) :&= \mathbb{E}\{(X_{t+h}^{(-1)}-\mathbb{E}[X_{t+h}^{(-1)}])(X_{t}^{(-1)}-\mathbb{E}[X_{t}^{(-1)}])^{\top}\}\\
> >         &= \int_0^t e^{A_{22}(t+h-s)} G_2 G_2^{\top}e^{A_{22}^{\top}(t-s)}ds
> >     \end{split}.
> >     \end{equation*}
> >     Similarly, one can obtain that the mean vector and cross-covariance of the law of the post-intervention process of the second SDE by making intervention $\tilde{X}\_t^1 = \xi$ satisfy:
> >     \begin{equation*}
> >         E[\tilde{X}\_t^{(-1)}] = e^{\tilde{A}\_{22}t}x_0^{(-1)} - (I - e^{\tilde{A}\_{22}t})\tilde{A}\_{22}^{-1}\tilde{A}\_{21}\xi,
> >     \end{equation*}
> >     and
> >     \begin{equation*}
> >     \begin{split}
> >         V(\tilde{X}\_{t+h}^{(-1)}, \tilde{X}\_{t}^{(-1)}) :&= \mathbb{E}\{(\tilde{X}\_{t+h}^{(-1)}-\mathbb{E}[\tilde{X}\_{t+h}^{(-1)}])(\tilde{X}\_{t}^{(-1)}-\mathbb{E}[\tilde{X}\_{t}^{(-1)}])^{\top}\}\\
> >         &= \int_0^t e^{\tilde{A}\_{22}(t+h-s)} \tilde{G}\_2 \tilde{G}\_2^{\top}e^{\tilde{A}\_{22}^{\top}(t-s)}ds
> >     \end{split}.
> >     \end{equation*}
> >     Then we will show that $E[X_t^{(-1)}] = E[\tilde{X}\_t^{(-1)}]$, and $V(X_{t+h}^{(-1)},X_t^{(-1)}) = V(\tilde{X}\_{t+h}^{(-1)}, \tilde{X}\_{t}^{(-1)})$ for all $0 \leqslant t, h <\infty$. Recall that we have shown $(A, GG^{\top}) = (\tilde{A}, \tilde{G}\tilde{G}^{\top})$, thus, $A_{22} = \tilde{A}\_{22}$ and $A_{21} = \tilde{A}\_{21}$, then it is readily checked that $E[X_t^{(-1)}] = E[\tilde{X}\_t^{(-1)}]$ for all $0 \leqslant t <\infty$.
> >
> > **The rest of this example is presented in the following comment**

---

> > > ### Author Response · Authors · 2023-08-12
> > >
> > > **following the previous comment**
> > >
> > > Since
> > >     \begin{equation*}
> > >         GG^{\top} = \begin{bmatrix}
> > >             G_1 G_1^{\top} & G_1 G_2^{\top}\\\\
> > >             G_2 G_1^{\top} & G_2 G_2^{\top}
> > >         \end{bmatrix} = \tilde{G}\tilde{G}^{\top},
> > >     \end{equation*}
> > >     thus, $G_2 G_2^{\top} = \tilde{G}\_2 \tilde{G}\_2^{\top}$, then it is readily checked that $V(X_{t+h}^{(-1)},X_t^{(-1)}) = V(\tilde{X}\_\{t+h}^{(-1)}, \tilde{X}\_{t}^{(-1)})$. Since both of these two post-intervention processes are Gaussian processes, according to Lemma 3.1, the laws of these two post-intervention processes are the same. That is, the law of the post-intervention process is identifiable.
> > >
> > >         reference:
> > >         [1] P. Vatiwutipong and N. Phewchean. Alternative way to derive the distribution of the
> > >         multivariate ornstein–uhlenbeck process. Advances in Difference Equations, 2019(1):1–7,
> > >         2019
> > >
> > >  3. For SDE (3):
> > >     Recall that SDE (3) is defined as
> > >     \begin{equation*}
> > >         dX_t = A X_tdt + \textstyle\sum_{k=1}^mG_kX_tdW_{k,t} , \ \ \ X_0 = x_0,
> > >     \end{equation*}
> > >     where $0\leqslant t<\infty$, $A, G_k \in \mathbb{R}^{d\times d}$ for $k = 1, \ldots, m$ are some constant matrices,  $W := \{W_t= [W_{1,t}, \ldots, W_{m,t}]^{\top}: 0\leqslant t<\infty\}$ is an $m$-dimensional standard Brownian motion. Let $X(t; x_0, A, \\{G_k\\}\_{k=1}^m)$ denote the solution process of SDE (3). Let $\tilde{A}, \tilde{G}\_k \in \mathbb{R}^{d\times d}$ for $k = 1, \ldots, m$ define the following SDE:
> > >     \begin{equation*}
> > >         d\tilde{X}\_t = \tilde{A}  \tilde{X}\_tdt + \textstyle\sum_{k=1}^m\tilde{G}\_k\tilde{X}\_tdW_{k,t} , \ \ \ \tilde{X}\_0 = x_0,
> > >     \end{equation*}
> > >     such that  $X(\cdot; x_0, A, \\{G_k\\}\_{k=1}^m) \overset{\text{law}}{=} \tilde{X}(\cdot; x_0, \tilde{A}, \\{\tilde{G}\_k\\}\_{k=1}^m)$. Then under our proposed identifiability condition stated in Theorem 3.4, we have shown that the generator of SDE (3) is identifiable, i.e., $(A, \textstyle\sum_{k=1}^mG_kxx^{\top}G_k^{\top}) = (\tilde{A}, \textstyle\sum_{k=1}^m\tilde{G}\_kxx^{\top}\tilde{G}\_k^{\top})$ for all $x\in \mathbb{R}^d$. Till now, we have shown that under our proposed identifiability conditions, "the law of the observational process  $\xrightarrow{\text{identity}}$ the generator of the observational SDE".
> > >     Then we aim to show that the law of the post-intervention process is also identifiable. For notational simplicity, we consider intervention on the first coordinate, making the intervention $X_t^1 = \xi$ and $\tilde{X}\_t^1 = \xi$ for $0\leqslant t < \infty$. It will suffice to show equality of the laws of the non-intervened coordinates (i.e., $X_t^{(-1)}$ and $\tilde{X}\_t^{(-1)}$, note the superscripts do not denote reciprocals, but denote the $(d-1)$-coordinates without the first coordinate). Express the matrices of $A$ and $G_k$ for $k=1, \ldots, m$ in blocks
> > >     \begin{equation*}
> > >         A = \begin{bmatrix}
> > >             A_{11} & A_{12}\\\\
> > >             A_{21} & A_{22}
> > >         \end{bmatrix},
> > >     G_k =
> > >     \begin{bmatrix}
> > >         G_{k,11}  & G_{k,12}\\\\
> > >         G_{k,21}  & G_{k,22}
> > >     \end{bmatrix},
> > >     \end{equation*}
> > >     where $A_{11}, G_{k,11} \in \mathbb{R}^{1\times 1}$, $A_{22}, G_{k,22} \in \mathbb{R}^{(d-1)\times (d-1)}$. Also consider corresponding expressions of matrices $\tilde{A}$ and $\tilde{G}\_k$ for $k=1, \ldots, m$.
> > >     By making intervention $X_t^1 = \xi$, one obtains the post-intervention process of the first SDE satisfies:
> > >     \begin{equation*}
> > >         dX_t^{(-1)} = (A_{21}\xi + A_{22}X_t^{(-1)})dt + \textstyle\sum_{k=1}^m (G_{k,21}\xi + G_{k,22} X_t^{(-1)})dW_{k,t}, X_0^{(-1)} = x_0^{(-1)}.
> > >     \end{equation*}
> > >     Since this post-intervention process is not a Gaussian process, one cannot explicitly show that the law of the post-intervention process is identifiable. Instead, we check the surrogate of the law of the post-intervention process, that is the first- and second-order moments of the post-intervention process $X_t^{(-1)}$. Which denote as $m(t)^{(-1)} = \mathbb{E}[X_t^{(-1)}]$ and $P(t)^{(-1)} = \mathbb{E}[X_t^{(-1)} (X_t^{(-1)})^{\top}]$ respectively. Then $m(t)^{(-1)}$ and $P(t)^{(-1)}$ satisfy the following ODE systems:
> > >     \begin{equation*}
> > >         \frac{dm(t)^{(-1)}}{dt} = A_{21}\xi + A_{22}m(t)^{(-1)},\ \ \ m(0)^{-1} = x_0^{(-1)},
> > >     \end{equation*}
> > >     and
> > >     \begin{equation*}
> > >     \begin{split}
> > >         \frac{dP(t)^{(-1)}}{dt} &= m(t)^{(-1)}\xi^{\top}A_{21}^{\top} + A_{21}\xi(m(t)^{(-1)})^{\top} + P(t)^{(-1)} A_{22}^{\top} + A_{22}P(t)^{(-1)} \\\\
> > >         &+ \textstyle\sum_{k=1}^m(G_{k,21}\xi\xi^{\top}G_{k,21}^{\top} + G_{k,22}m(t)^{(-1)}\xi^{\top}G_{k,21}^{\top} + G_{k,21}\xi(m(t)^{(-1)})^{\top}G_{k,22}^{\top} \\\\ &+G_{k,22}P(t)^{(-1)}G_{k,22}^{\top} ), \ \ \ P(0)^{(-1)} = x_0^{(-1)} (x_0^{(-1)})^{\top}.
> > >     \end{split}
> > >     \end{equation*}
> > >
> > > **The rest of this example is presented in the following comment**

---

> > > > ### Author Response · Authors · 2023-08-12
> > > >
> > > >  **following the previous comment**
> > > >
> > > > Similarly, one can obtain the ODE systems describing the $\tilde{m}(t)^{(-1)}$ and $\tilde{P}(t)^{(-1)}$. Then we will show that $m(t)^{(-1)} = \tilde{m}(t)^{(-1)}$ and $P(t)^{(-1)} = \tilde{P}(t)^{(-1)}$ for all $0\leqslant t < \infty$. Recall that we have shown $A = \tilde{A}$, thus $A_{21} = \tilde{A}\_{21}$ and $A_{22} = \tilde{A}\_{22}$, then it is readily checked that $m(t)^{(-1)} = \tilde{m}(t)^{(-1)}$. In the proof of Theorem 3.4, we have shown that $\textstyle\sum_{k=1}^mG_kP(t)G_k^{\top} = \textstyle\sum_{k=1}^m\tilde{G}\_kP(t)\tilde{G}\_k^{\top}$ for all $0\leqslant t < \infty$, where $P(t) = \mathbb{E}[X_t X_t^{\top}]$. Simple calculation shows that
> > > >     \begin{equation*}
> > > >        \sum_{k=1}^mG_kP(t)G_k^{\top} =\sum_{k=1}^m \Bigg( \begin{bmatrix}
> > > >            G_{k,11} & G_{k,12}\\\\
> > > >           G_{k,21} & G_{k,22}
> > > >        \end{bmatrix}
> > > >        \begin{bmatrix}
> > > >            \xi \xi^{\top} & \xi (m(t)^{(-1)})^{\top}\\\\
> > > >            m(t)^{(-1)}\xi^{\top}&  P(t)^{(-1)}
> > > >        \end{bmatrix}
> > > >        \begin{bmatrix}
> > > >            G_{k,11}^{\top} & G_{k,21}^{\top}\\\\
> > > >            G_{k,12}^{\top} & G_{k,22}^{\top}
> > > >        \end{bmatrix}\Bigg),
> > > >     \end{equation*}
> > > >     Then one can get that the (2,2)-th entry of the matrix $\textstyle\sum_{k=1}^m G_kP(t)G_k^{\top}$ is the same as the $\textstyle\sum_{k=1}^m (\ldots)$ part in the ODE corresponds to $P(t)^{(-1)}$, since $\textstyle\sum_{k=1}^mG_kP(t)G_k^{\top} = \textstyle\sum_{k=1}^m\tilde{G}\_kP(t)\tilde{G}\_k^{\top}$, then the $\textstyle\sum_{k=1}^m (\ldots)$ part in the ODEs correspond to both $P(t)^{(-1)}$ and $\tilde{P}(t)^{(-1)}$ are the same. Thus, it is readily checked that $P(t)^{(-1)} = \tilde{P}(t)^{(-1)}$ for all $0\leqslant t < \infty$. Though we cannot explicitly show that the law of the post-intervention process is identifiable, showing that the first- and second-order moments of the post-intervention process is identifiable can indicate the identification of the law of the post-intervention process to a considerable extent.
> > > >
> > > > Thank you again for your valuable comments. We believe that the quality of our work has been significantly improved thanks to your insightful comments.

---

### Decision · Program_Chairs · 2023-09-21

**Decision:**

Accept (poster)

**Comment:**

This work provides a theory of generator identification for linear SDEs, which has relevance to the blossoming theory for identifying causal models. The theoretical contribution is solid and relevant, but most reviewers found the discussion of the motivation and setting to be difficult to follow for an ML audience. That said, the authors have been extensively engaged in the rebuttal period and have promised specific, concrete changes to the manuscript that should satisfactorily address this concern. To quote one reviewer,

"Multiple reviewers, myself included, have raised concerns about how accessible this work is to the broader NeurIPS audience. In that vein, I'd like to underscore the importance of refining the introduction and background sections. It would be great to see these sections become more reader-friendly, providing a smoother entry point into the topic and shedding light on the motivating factors behind this research endeavor."

In this vein, it will be crucial for these edits (and others mentioned in the reviews below) to be included in the camera ready revision prior to publication.